# The brittle star genome illuminates the genetic basis of animal appendage regeneration

Elise Parey ●[1]✉, Olga Ortega-Martinez ●[2], Jérôme Delroisse ●[3], Laura Piovani ●[1], Anna Czarkwiani ●[1,10], David Dylus ●[1,11], Srishti Arya ●[1,12], Samuel Dupont[4,5], Michael Thorndyke[4,13], Tomas Larsson[6], Kerstin Johannesson ●[2], Katherine M. Buckley ●[7], Pedro Martinez ●[8,9], Paola Oliveri ●[1]✉ & Ferdinand Marlétaz ●[1]✉

Species within nearly all extant animal lineages are capable of regenerating body parts. However, it remains unclear whether the gene expression programme controlling regeneration is evolutionarily conserved. Brittle stars are a species-rich class of echinoderms with outstanding regenerative abilities, but investigations into the genetic bases of regeneration in this group have been hindered by the limited genomic resources. Here we report a chromosome-scale genome assembly for the brittle star *Amphiura filiformis*. We show that the brittle star genome is the most rearranged among echinoderms sequenced so far, featuring a reorganized Hox cluster reminiscent of the rearrangements observed in sea urchins. In addition, we performed an extensive profiling of gene expression during brittle star adult arm regeneration and identified sequential waves of gene expression governing wound healing, proliferation and differentiation. We conducted comparative transcriptomic analyses with other invertebrate and vertebrate models for appendage regeneration and uncovered hundreds of genes with conserved expression dynamics, particularly during the proliferative phase of regeneration. Our findings emphasize the crucial importance of echinoderms to detect long-range expression conservation between vertebrates and classical invertebrate regeneration model systems.

Brittle stars are by far the most speciose class of echinoderms; over 2,600 extant species occupy benthic marine habitats globally[1,2]. However, they remain poorly documented from a genomic standpoint, despite their broad interest to diverse fields including marine (palaeo) ecology, biodiversity monitoring, developmental biology and regenerative biology[2–9].

The echinoderm phylum encompasses five classes with a well-resolved phylogeny[10–13]: brittle stars, sea stars, sea urchins, sea cucumbers and sea lilies/feather stars. Genomics in this phylum began with the pioneering effort to sequence the genome of the purple sea urchin (*Strongylocentrotus purpuratus*)[14]. Analysis of this genome provided broad insights into the evolution of diverse traits and biological processes[15–17]. In recent years, the taxonomic sampling of echinoderm genomes has steadily expanded[18–24], enabling investigations into the evolution of new body plans and developmental strategies. However, given the deep evolutionary divergence of the five echinoderm classes (480–500 million years ago (Ma)), the lack of robust genomic resources for the brittle stars represents a problematic knowledge gap.

Adult echinoderms share a characteristic pentameral symmetry, which represents the most derived body plan among Bilateria[25]. Early analyses of sea urchin genomes unveiled local reorganizations within the Hox cluster, prompting speculation that they were associated with the evolution of this unique body plan[26–29]. However, the subsequent discovery of an intact Hox cluster in the crown-of-thorns sea star revealed that these rearrangements were not instrumental in the establishment of the pentameral symmetry[30,31]. These observations showcase the need to examine a more comprehensive sample of echinoderm whole genomes to accurately identify echinoderm-specific chromosomal rearrangements and subsequently investigate their functional implications.

Echinoderms exhibit extensive regenerative abilities. Species from each of the five classes are capable of varying levels of regeneration, including (larval) whole-body regeneration, appendage or organ regeneration[32]. Although species within nearly all major animal groups exhibit some regenerative capacity, it is not clear whether this trait is ancestral or independently acquired[33–35]. A comparative analysis of whole-body regeneration across a sea star larva, planarian worm and hydra has suggested that broadly conserved molecular pathways may mediate regeneration[36]. However, given the diversity of regenerative modes, additional comparative analyses of regenerating organisms are needed to fully understand the evolution of this complex process[33,34]. In particular, gene expression dynamics during regeneration have not been explicitly compared between invertebrates and vertebrates, partly because of the lack of gene expression profiling across comparable regenerating structures and difficulties in identifying orthologues among distant model systems. Echinoderms are more closely related to vertebrates than other classical invertebrate models of regeneration, hence providing a unique phylogenetic perspective. However, echinoderms remain largely underrepresented in transcriptomic assays of regeneration[5,37,38].

The brittle star *Amphiura filiformis* is one highly regenerative echinoderm species: fully differentiated arms regrow in a few weeks following amputation and over 90% of individuals sampled in the wild display signs of arm regeneration[39,40]. Consequently, *A. filiformis* is emerging as a powerful model for animal appendage regeneration, with a well-established morphological staging system[41–47]. Here we report a chromosome-scale genome assembly for the brittle star *A. filiformis*. This resource is crucial to accurately capture the brittle star gene repertoire and probe genome-wide gene expression patterns during regeneration. We investigate the complex history of karyotypes, Hox cluster and gene family evolution across echinoderms and reveal that *A. filiformis* displays the most rearranged echinoderm genome sequenced so far. Moreover, we report that *A. filiformis* extensive regenerative capacities correlate with significant expansions of genes involved in wound healing. Finally, we generate extensive transcriptomic data from regenerating brittle star arms, which we analyse in a comparative framework with previously generated datasets from the crustacean *Parhyale hawaiensis*[48] and the axolotl *Ambystoma mexicanum*[49], to illuminate common genetic mechanisms of animal appendage regeneration.

## A chromosome-scale genome assembly for *A. filiformis*

We sequenced and assembled the genome of the brittle star *A. filiformis* using high-coverage long nanopore reads assisted with proximity ligation data for scaffolding (Methods). The haploid assembly spans 1.57 Gb and contains 20 chromosome-size scaffolds (>60 Mb) that account for 93.5% of the assembly length (Extended Data Fig. 1, N50: 68.8 Mb (scaffolds equals to or longer than this value contain half the assembly)). We annotated a total of 30,267 protein-coding genes (92.7% complete BUSCO score; Methods, and Supplementary Tables 1 and 2), which is in line with the predicted gene complements of other echinoderms[18–20,23,50]. In addition, we generated manually curated lists for *A. filiformis* genes associated with immunity, stemness, signalling and neuronal function as well as transcription factors (Supplementary

Tables 3–5 and Methods). The *A. filiformis* genome represents to our knowledge the first high-quality and chromosome-scale genome assembly for the brittle star class (Supplementary Note 1) and fills an important gap in the echinoderm genomics landscape.

## The most rearranged genome among sequenced echinoderms

Chromosome evolution in echinoderms has primarily been investigated through the lens of sea urchin genomes. Sea urchins have globally preserved the ancestral bilaterian chromosomes[23,51,52]. However, they also underwent several chromosomal fusions whose origin cannot be established without examining more echinoderm genomes. To address this gap and document chromosome evolution across echinoderm lineages, we took advantage of chromosome-scale genomes released for sea stars, sea cucumbers and sea urchins[19,23,24] and our brittle star genome. Using these genomes and selected outgroups, we reconstructed the linkage groups present in their ancestor (Eleutherozoa linkage groups (ELGs), Fig. 1a).

Only one interchromosomal macrosyntenic rearrangement occurred in the 500 million years (Myr) of independent evolution between the spiny sea star (*Marthasterias glacialis*) and the black sea cucumber (*Holothuria leucospilota*)[19,24] (Fig. 1b and Methods). By contrast, the *A. filiformis* brittle star genome is extensively rearranged: only three chromosomes have a direct one-to-one orthology relationship with spiny sea star chromosomes (Fig. 1c). We reconstructed the ancestral ELGs on the basis of near-perfect conservation of macrosynteny between the spiny sea star and black sea cucumber and using outgroups to disentangle derived and ancestral chromosomal arrangements (Extended Data Fig. 2). We predicted that 23 ELGs were present in the eleutherozoan ancestor (Fig. 1d), descending from the 24 bilaterian linkage groups (BLGs)[52] through the fusion of the BLGs B2 and C2. The black sea cucumber maintained the 23 ancestral ELGs, a single chromosomal fusion took place in the spiny sea star lineage (interchromosomal rearrangement rate of 0.002 event per Myr), five fusions occurred in the sea urchin *Paracentrotus lividus* (0.01 event per Myr) and 26 interchromosomal rearrangements in the brittle star *A. filiformis* (0.052 event per Myr; Extended Data Fig. 3). These results indicate that sea cucumbers, sea stars and sea urchins have broadly conserved the ancestral bilaterian linkage groups, whereas the brittle star genome is highly reshuffled. Examination of additional sea star and sea urchin genomes suggests that these trends might extend to species within their respective classes[14,21,23,53–57] (Extended Data Fig. 3).

Among the four echinoderm genomes analysed, we find that repetitive elements coverage correlates as expected with genome size but not with rates of rearrangements. Repeat coverage is highest in the highly rearranged brittle star genome (1.57 Gb, repeat coverage 59.3%) and slowly evolving black sea cucumber *H. leucospilota* (1.31 Gb, 56.0%) compared with the sea urchin *P. lividus* (927 Mb, 49.2%) and spiny sea star *M. glacialis* (521 Mb, 47.6%). Repetitive elements accumulated more gradually in the slowly evolving sea star and sea cucumber genomes, compared with both the sea urchin and the brittle star which display recent bursts of repeat activity (Fig. 1e). Specifically, the brittle star genome is marked by a burst of repeat activity 10–15 Ma, consisting mostly of DNA transposons (peak of repeats with 2% divergence to consensus; Methods). We thus speculate that the evolutionary history of *A. filiformis* includes at least one period of genomic instability[58]. Together, these data highlight contrasting trends of chromosome evolution across echinoderm classes and indicate that *A. filiformis* is the most rearranged echinoderm genome among those sequenced so far.

## A locally rearranged Hox cluster

The organization of the Hox and ParaHox gene clusters has been documented in each class of echinoderms except for brittle stars[22,26,30,50,59]. To further explore the enigmatic evolution of these developmental homeobox gene clusters in echinoderms[31], we investigated the structure of

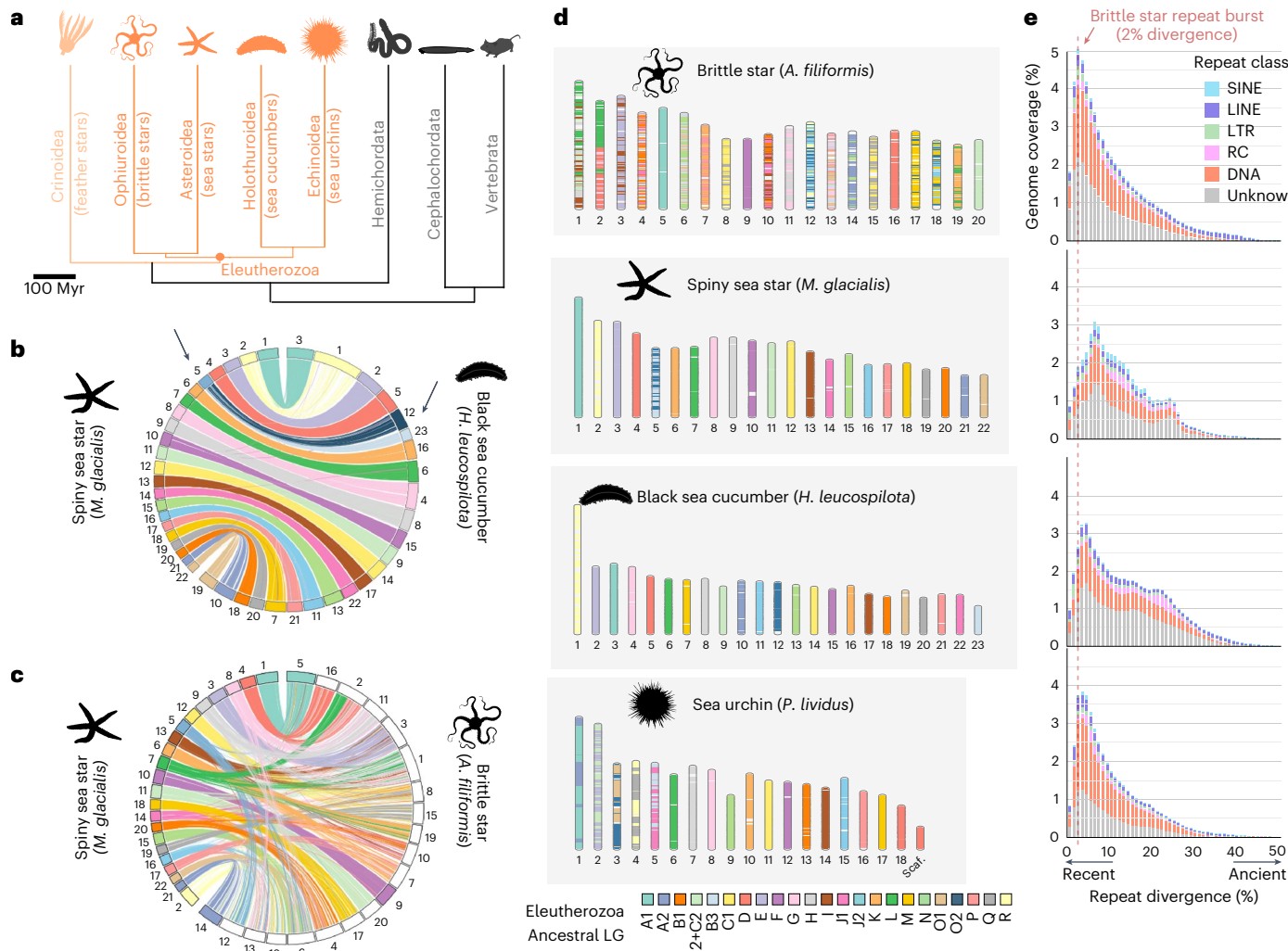

**Fig. 1 | Chromosome evolution in echinoderms. a**, Phylogenetic relationships of the five echinoderm classes (orange), with the position of the Eleutherozoa ancestor highlighted, and hemichordates and chordates as outgroups. Classes with available chromosome-scale genome assembly are shown in dark orange. Divergence times among echinoderms and with hemichordates were extracted from ref. 13, divergence with chordates from TimeTree[155]. **b**, Synteny comparison between the 22 chromosomes of spiny sea star and the 23 chromosomes of the black sea cucumber. The single macrosyntenic rearrangement between the two genomes is indicated with arrows. **c**, Synteny comparison between the 22 chromosomes of spiny sea star and the 20 chromosomes of brittle star. The three brittle star chromosomes with a one-to-one relationship with sea star chromosomes are shown with a colour matching its orthologous counterpart in spiny sea star (Fisher's exact test $P_{adj} < 10^{-5}$). **d**, Chromosome evolution in Eleutherozoa. We named the ancestral ELG using established naming conventions proposed for the 24 bilaterian ancestral linkage groups defined previously[23,52]. B2 + C2 corresponds to a fusion of bilaterian B2 and C2 present in the Eleutherozoa ancestor. **e**, Repeat landscapes for the brittle star and the three selected echinoderm genomes, with the *y* axis representing the genomic coverage and the *x* axis the CpG-corrected Kimura divergence to the repeat consensus. Species are presented in the same order as in **d**. The dashed red line indicates the repeat burst in the brittle star.

the *A. filiformis* Hox and ParaHox clusters. Notably, the *A. filiformis* Hox and ParaHox clusters both exhibit genomic rearrangements (Fig. 2, Extended Data Fig. 4 and Methods). Anterior Hox genes (*Hox1*, *Hox2* and *Hox3*) are inverted within the 3′ end of the cluster and *Hox8* was inverted and displaced between *Hox9/10* and *Hox11/13a*. Five repeat families are significantly expanded within the brittle star Hox cluster. The repeat family SINE/tRNA-Deu-L2 is significantly associated with breakpoint locations and may have contributed to the Hox1–Hox3 inversion through non-homologous repair (Benjamini–Hochberg (BH)-corrected permutation-based *P* < 0.05; Fig. 2b). Expanded repeats have an inferred divergence of 18–22% to their consensus, suggesting that they were active ~100 Ma (Methods). While brittle star Hox reorganization is distinct from the one observed in sea urchins, in both cases one of the breakpoints is located near *Hox4* (Fig. 2c). Moreover, the brittle star ParaHox cluster also underwent disruptions (Fig. 2d), such that *Gsx* was tandemly duplicated to generate two paralogues (protein identity: 74%) located a long distance

(>5 Mb) from *Xlox-Cdx*. Whereas *Xlox-Cdx* maintained close linkage in the brittle star, all three members of the ParaHox cluster are dispersed over their chromosome in sea urchins[59].

Hox expression throughout echinoderm embryogenesis, larval stages and metamorphosis remains largely enigmatic and spatio-temporal expression does not follow classical Hox collinearity rules[31,60]. We investigated Hox and ParaHox gene expression during brittle star development using previously published datasets[61–63] (Fig. 2e, Supplementary Table 1 and Methods). As in sea urchins[60], *Hox1* and *Hox3–Hox6* are expressed at very low levels in the brittle star embryos and pluteus larvae (normalized transcript per million transcripts (TPM) < 2), but *Hox7*, *Hox11/13a* and *Hox11/13b* are highly expressed. However, in the brittle star, *Hox2* is expressed early in embryogenesis, with maximal expression at 9 h post fertilization, whereas sea urchins *Hox2* is not expressed during early development[60,64]. Expression patterns of the brittle star ParaHox genes (Fig. 2e) match those observed in

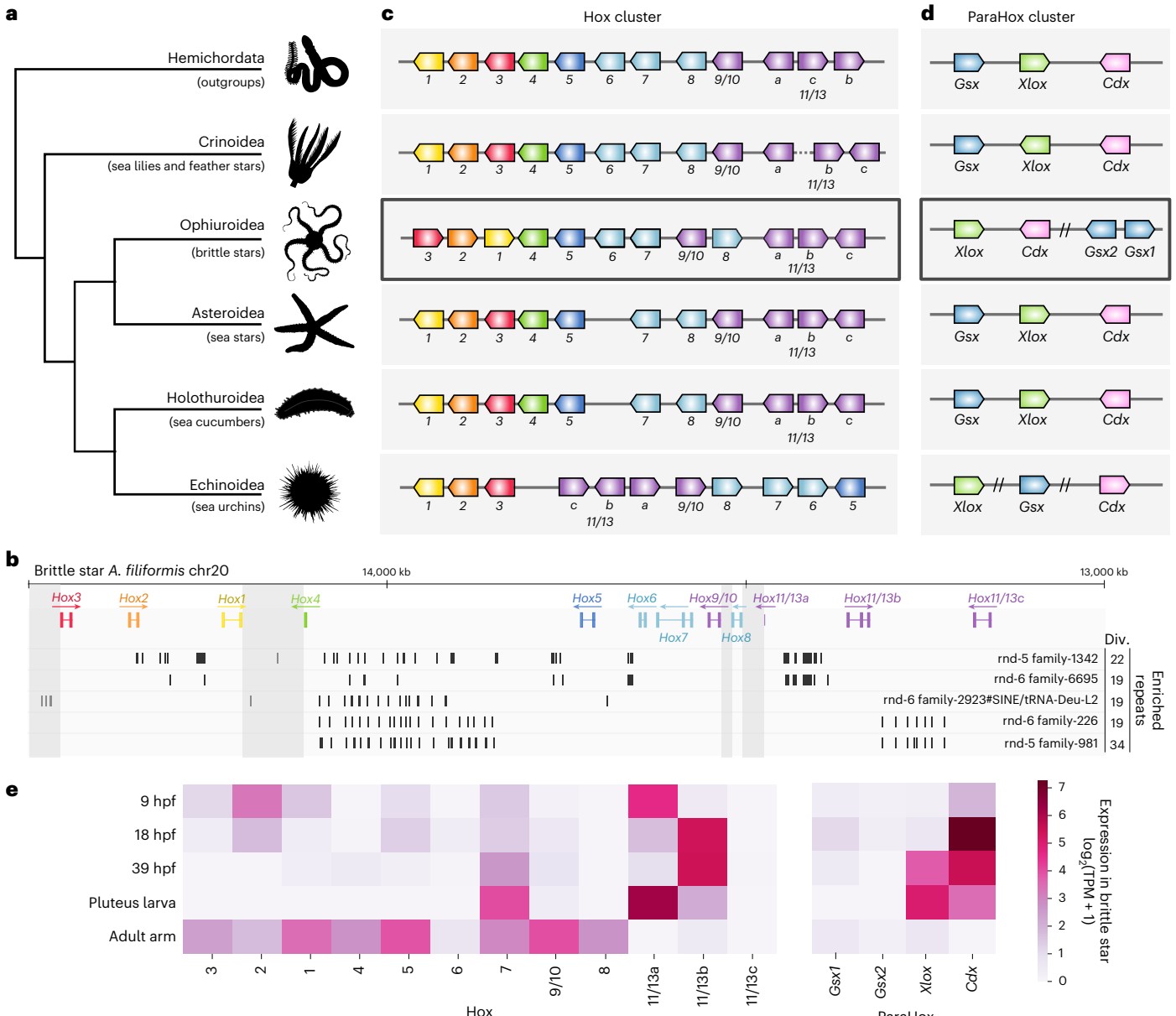

**Fig. 2 | Hox and ParaHox clusters organization across echinoderms.**
**a**, Phylogenetic relationships among the five classes of echinoderms, with hemichordates as the outgroup. **b**, Genomic organization of the brittle star *A. filiformis* Hox cluster. Significantly expanded repeats at the Hox cluster location are represented in their respective tracks below Hox genes, with the average sequence divergence to consensus indicated (div., %). Divergence to consensus is a proxy for repeat age, where higher divergence indicates older repeat insertions. Vertical grey rectangles indicate breakpoint locations. **c**, Schematic representation of Hox cluster organization across echinoderms and outgroups, based on organization reported in *Saccoglossus kowalevskii* and *Ptychodera flava*[156] for Hemichordata, feather star *Anneissia japonica*[50]

for Crinoidea, brittle star *A. filiformis* for Ophiuroidea, crown-of-thorns sea star *A. planci*[18,30] for Asteroidea, Japanese sea cucumber *Apostichopus japonicus*[22,66] for Holothuroidea and purple sea urchin *S. purpuratus*[26] for Echinoidea. **d**, ParaHox gene cluster organization, based on the same genomes as in **b**. Double slashes indicate non-consecutive genes, all separated by distances >5 Mb on the same chromosome or scaffold. **e**, Expression of Hox and ParaHox genes throughout 4 brittle star developmental time points and in the adult arm. hpf, hours post fertilization. Expression data from refs. [61–63] were normalized across samples using the TMM method[146] on the full set of brittle star genes, and shown as log₂(TPM + 1).

sea stars[59]. By contrast, dispersion of the ParaHox cluster in sea urchins is associated with the distinct temporal activation of *Gsx*, *Xlox* and *Cdx* during embryogenesis[65].

These results highlight intriguing parallels in the reorganization of developmental gene clusters and their expression patterns between brittle stars and sea urchins. Limited data are available on Hox gene expression in other echinoderm classes, but investigations in crinoids and sea cucumbers suggest that even in species with an intact Hox cluster, the anterior genes (*Hox1–Hox6*) exhibit low or no expression in early embryonic stages, whereas *Hox7* and *Hox11/13b* are

expressed[66–68]. Together, these suggest that only a subset of Hox genes have a role in echinoderm embryogenesis[69]. We therefore speculate that the relaxation of expression constraints on Hox genes during echinoderm embryogenesis may have allowed for the rearranged Hox cluster architectures seen in the sea urchin and brittle star lineages.

## Expansion of regeneration-related gene families

To assess the functional implications of gene complement evolution in echinoderms, we first documented the duplication history of *phb* and *luciferase* genes, known to be important for echinoderm larval

skeleton and bioluminescent abilities and extensively duplicated in *A. filiformis*[8,23,61,70] (Extended Data Fig. 5 and Supplementary Note 2). We next inferred gene family expansion and contraction events along echinoderm evolution (Fig. 3a and Methods). In contrast to other deuterostome lineages, which exhibit either extensive gene losses[71] or duplications[72], we found that echinoderms harbour relatively stable gene complements (790 expanded or contracted of 10,367 tested families). Several Gene Ontology (GO) terms are systematically found in the expanded and contracted families of brittle star and other echinoderms (Fig. 3b, Supplementary Tables 6 and 7 and Methods). This includes several GO terms linked to immune-related processes (for example, 'response to other organisms', 'leucocyte migration', 'cell recognition'), which encompass genes with elevated gene birth and death rates in animals (for example, Toll-like receptors)[73–75]. Some GO enrichments may reflect specific aspects of echinoderm biology. For instance, recurrent duplications of 'regeneration-related' genes may underlie the remarkable regenerative capacity of many echinoderms (Fig. 3b,c). In *A. filiformis*, members of these expanded gene families (Fig. 3c) are expressed during arm regeneration (Extended Data Fig. 6). In addition, genes within four of the seven regeneration-related expanded families (*plasminogen*, *carboxypeptidase B*, *coagulation factor* and *ficolin*) directly regulate coagulation and/or clotting in vertebrates[76] but may have a broader role in immune defence in echinoderms[77,78]. Moreover, the *ficolin* gene has also been implicated in the early stages of *A. filiformis* arm regeneration[79,80]. Duplications within the brittle star may have contributed to the evolution of a rapid and efficient wound closure process that is prerequisite to regeneration[80,81]. Finally, genes involved in keratan sulfate metabolism are overrepresented in both expanded and contracted gene families in the brittle star, with some members expressed in regeneration (Fig. 3b and Extended Data Fig. 6). Increased sulfated glycosaminoglycans production has been previously reported to be required for proper arm regeneration in *A. filiformis*[82]. We speculate that the evolution of brittle star efficient regeneration may have been accompanied by a specialization of glycosaminoglycan sulfate metabolism.

## Gene expression during brittle star arm regeneration

To gain insight into the transcriptional programmes that underlie brittle star arm regeneration, we profiled gene expression in seven representative regeneration stages following amputation and one non-regenerating control. Stages were selected on the basis of well-established morphological landmarks of brittle star arm regeneration[42] (Methods and Fig. 4a). Using soft-clustering, we classified genes into nine major temporal clusters (A1–A9) (Fig. 4a, Extended Data Fig. 6 and Methods). Functional enrichment analysis of genes within the co-expression clusters revealed three distinct phases of arm regeneration: (1) wound healing, (2) proliferation and (3) tissue differentiation (Fig. 4b and Extended Data Fig. 7). These results are consistent with morphological timelines of regeneration in the brittle star and other animals[33,38,42] but importantly capture the underlying genome-wide transcriptional programme. We corroborate the expression pattern of previously characterized brittle star regeneration genes and further report novel key candidates (Extended Data Fig. 6 and Supplementary Tables 8 and 9).

Early regeneration is marked by the expression of genes involved in wound response, including immunity/wound healing (clusters A1–A2), and cell migration/tissue protection (clusters A3–A4), which are enriched in immune and kinase genes, respectively (Fig. 4b,c and Supplementary Table 10). The regions surrounding transcription start sites (TSS) of genes within cluster A2 are enriched for transcription factor-binding motifs of NF-κB, a broadly conserved regulator of immune response (Fig. 4d). The early activation of NF-κB in the context of regeneration has been evidenced in vertebrates and hydra[83–85], and our findings suggest its implication in the brittle star regenerative response as well.

Wound healing is followed by cell proliferation (clusters A9 and A5–A7), as indicated by the overrepresentation of stemness genes and genes involved in cell proliferation, cell division and enhanced translational activity. Accordingly, binding motifs associated with several proliferation-related transcription factors are enriched around the TSS of genes from clusters A5 and A6. These transcription factors have not been previously investigated in the context of brittle star regeneration but are functionally well characterized in vertebrates. This includes NRF1 and p53, which have been implicated in vertebrates in regulating (stem) cell survival and proliferation[86,87], PRDM14 and YY1, which regulate pluripotency[88,89], and RORa, which controls inflammation by downregulating targets of NF-κB[90] and may thus have a role in the transition from wound response to proliferation (Fig. 4c,d). We also find enrichment of binding motifs corresponding to zinc-finger transcription factors that are involved in cell proliferation and pluripotency[91,92]. While we note that binding motif overrepresentation analyses are inherently biased towards more-studied vertebrate systems, transcription factor gene expression in the brittle star is globally consistent with reported motif enrichments (Extended Data Fig. 6). Cluster A9 encompasses genes expressed as early as 48 h post amputation (hpa) and active throughout regeneration, including translational regulators, cell division and vesicle transport genes (Fig. 4b), as well as genes involved in signalling pathways known to promote cell proliferation in vertebrates and fruit flies (VEGF, AKT, insulin-like and JAK-STAT pathways)[93–96] (Fig. 4c and Extended Data Fig. 6). The VEGF and AKT pathways have been previously implicated in brittle star regeneration[46]. Together, these data suggest that the signalling cascades that initiate cell proliferation are induced very early during brittle star regeneration (cluster A9); they are activated during the wound response phase and exhibit amplified expression during the peak of cell proliferation (stage 5; Fig. 4a). The early onset of proliferation (~48 hpa) is consistent with previous observations of cell proliferation and expression quantification of selected marker genes[42,44].

Finally, late regeneration is characterized by the expression of genes involved in differentiation, patterning and appendage morphogenesis, with a significant overrepresentation of transcription factors (cluster A8; Fig. 4b,c). This cluster includes two T-box transcription factors that are important for patterning in echinoderms (*tbx3-1* and *tbx3-2*) and two transcription factors with key roles in neurogenesis (*ngn1-like* and *hey1-like*)[97–99].

These data provide a genome-wide picture of the molecular pathways at play throughout brittle star arm regeneration and highlight three waves of gene expression that successively mobilize genes involved in wound response, cell proliferation and tissue differentiation. These general phases have been described in many regenerating animals, enabling investigations into the conservation of regeneration gene expression dynamics across species.

## Conserved gene expression during animal appendage regeneration

Several key genes and pathways have been repeatedly implicated in regeneration across animal lineages[33,38]. However, direct comparisons of temporal expression gene profiles throughout regeneration remain limited.

Using a genomic phylostratigraphy approach[100], we found that overall, brittle star arm regeneration is mediated by ancient genes (that is, metazoan or older) (Fig. 5a and Methods). The exception is the initial wound-healing phase, which is enriched in genes that are specific to the brittle star lineage. The observation that brittle star regeneration is mostly driven by ancient genes prompted us to investigate whether these genes are similarly involved in appendage regeneration across animals, and whether they are deployed in the same temporal order. We compared gene expression dynamics during appendage regeneration in *A. filiformis* with comparable datasets from the axolotl (*Ambystoma mexicanum*)[49] and the crustacean Parhyale (*Parhyale hawaiensis*)[48].

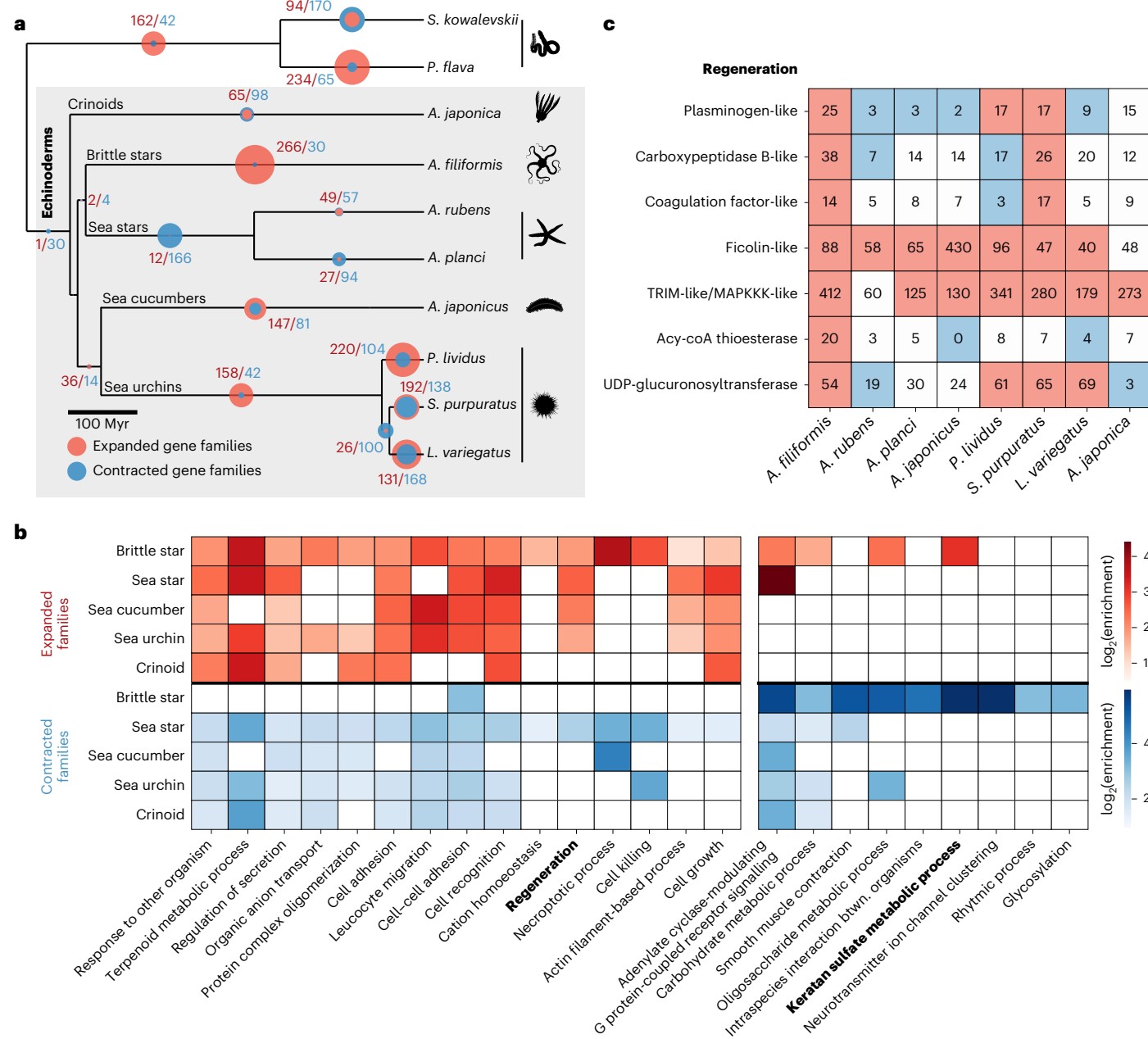

**Fig. 3 | Gene family evolution in echinoderms. a**, Number of significantly expanded (red) and contracted (blue) gene families throughout echinoderm evolution, from a total of 10,367 tested gene families (Methods). **b**, Gene Ontology (GO) functional enrichment tests (biological process) for expanded and contracted families in the different echinoderm classes. We selected the top 15 representative GO terms enriched in expanded brittle star gene families and 10 in contracted families (Methods). In the heat map, colours indicate GO terms significantly enriched in expanded or contracted families in other echinoderm classes (FDR < 0.05). Complete GO enrichment test results are provided in Supplementary Table 6, including *P* values, enrichment ratios, background and foreground gene families and genes. **c**, Gene copy number variation across echinoderms for regeneration gene families with significant expansion in *A. filiformis* (>1 brittle star gene in the family annotated with the GO term 'regeneration'). Gene families were named according to the *S. purpuratus* gene name. Red and blue colours denote significantly expanded and contracted families, respectively.

For this analysis, we defined nine major co-expression clusters during axolotl limb regeneration (Ax1–Ax9) (Extended Data Fig. 8 and Supplementary Tables 11 and 12) and used existing Parhyale clustering[48].

We used pairwise comparisons and permutation tests to reveal conserved co-expression clusters across species. Co-expression clusters were defined as conserved between two species when they used more shared genes than expected by chance (Fig. 5b and Methods). Among the nine co-expression clusters that mediate brittle star regeneration, five consist of genes that are also co-expressed during axolotl regeneration (926 genes), six clusters overlap with Parhyale (913 genes), and four clusters are consistent across the three species (154 genes) (Fig. 5b,c and Supplementary Tables 13 and 14). Expression comparisons between the more phylogenetically distant axolotl and Parhyale identify only two conserved co-expressed gene clusters (370 axolotl genes); this direct comparison is thus considerably less informative than comparisons that include the brittle star. Most genes with conserved expression patterns in the brittle star–axolotl comparison lack identifiable homologues in Parhyale, whereas genes with a conserved expression in the brittle star–Parhyale comparison exhibit a different expression pattern in the axolotl (Fig. 5c). This underscores

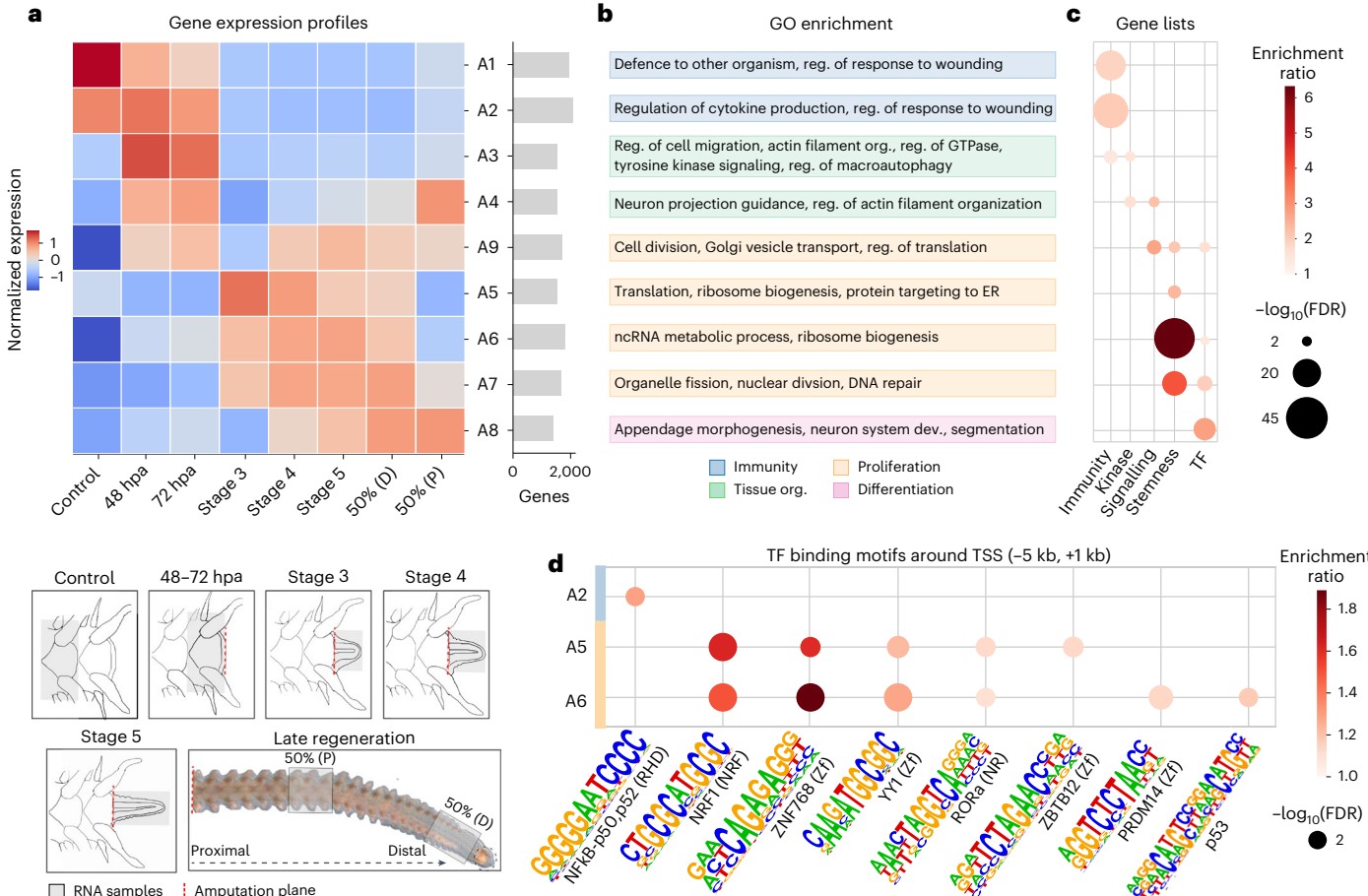

**Fig. 4 | Gene expression during brittle star arm regeneration. a,** Soft-clustering of gene expression profiles throughout regeneration time points, yielding 9 main temporal co-expression clusters (A1–A9) (Methods and Extended Data Fig. 6). Co-expression clusters are temporally ordered (from top to bottom) on the basis of their first expression time point. Barplots on the right indicate the number of genes assigned to each cluster. The RNA sampling procedure for each stage is illustrated at the bottom. Early stages are sampled at 48 and 72 hpa, when wound healing followed by regenerative bud formation occurs. Subsequent stages are defined by morphological landmarks: stage 3 corresponds to the appearance of the radial water canal and nerve (~6 dpa), stage 4 is the appearance of the first regenerated metameric units (~8 dpa), stage 5 corresponds to advanced arm extension and differentiation onset (~9 dpa), 50% stages correspond to when 50% of the regenerated arm has differentiated (~2–3 weeks post amputation) sampled at the distal (D, less differentiated) and proximal (P, more differentiated) ends[42,47]. **b,** GO enrichment for each co-expression cluster (Methods, see Extended Data Fig. 7 and Supplementary Table 10 for exhaustive GO results). **c,** Curated gene list enrichment for each co-expression cluster (hypergeometric test, Benjamini–Hochberg $P_{adj}$ < 0.05; Methods and Supplementary Table 2). **d,** Transcription factor (TF)-binding motifs enriched around the TSS (5 kb upstream to 1 kb downstream) of genes from co-expression clusters (hypergeometric test $P_{adj}$ < 0.05; Methods).

the relevance of using the brittle star to bridge comparisons across established regeneration models.

The broadly conserved co-expression clusters largely consist of genes expressed during the proliferative phase and, to a lesser extent, the initial wound-healing phase. By contrast, the genes that comprise clusters corresponding to tissue differentiation are distinct in each species, which is consistent with the fact that the regenerating appendages are not homologous across species. Notably, the conserved co-expression clusters are deployed in a consistent temporal sequence in each species (Fig. 5c). The only identified heterochrony concerns the matching of the axolotl cluster Ax3 (peak at 0–3 hpa) with brittle star cluster A5 (peak at 6 days post amputation (dpa)) (Figs. 5c and 4a, and Extended Data Fig. 8). Previous work suggested that similar co-expression gene modules are deployed during regeneration and development but are activated according to distinct temporal sequences[48]. We compared gene expression profiles during regeneration and development from the brittle star and Parhyale. The order in which co-expressed gene modules are activated is, as expected, more conserved within regeneration and within developmental datasets across species than between regeneration and development in

individual species (Extended Data Fig. 9). Together, these results broaden previous observations of distinct expression dynamics during development and regeneration, and document conserved gene expression modules recruited for animal appendage regeneration.

We further investigated the functions of brittle star genes with similar temporal expression profiles during regeneration in Parhyale and/or axolotl. Using a carefully selected background that accounts for homology detection and functional biases of different clusters (Methods), we found a significant overrepresentation of kinase and stemness genes and an underrepresentation of immune genes (gene list enrichment tests) (Fig. 4d and Extended Data Fig. 9). Moreover, these genes conserved in expression are enriched in general biological processes related to cell proliferation, such as translation, chromosome segregation, DNA replication and intracellular transport (GO enrichment tests; Fig. 4e). Among the conservative set of 154 genes with conserved expression profiles across the three species, only two transcription factors emerge (Supplementary Table 13): *Id2-like*, which activates regeneration-induced proliferation in mice[101] and *Wdhd1-like*, which regulates DNA replication[102]. We thus propose that *Id2* and *Wdhd1* may have a conserved role during animal regeneration. In addition, while several transcription

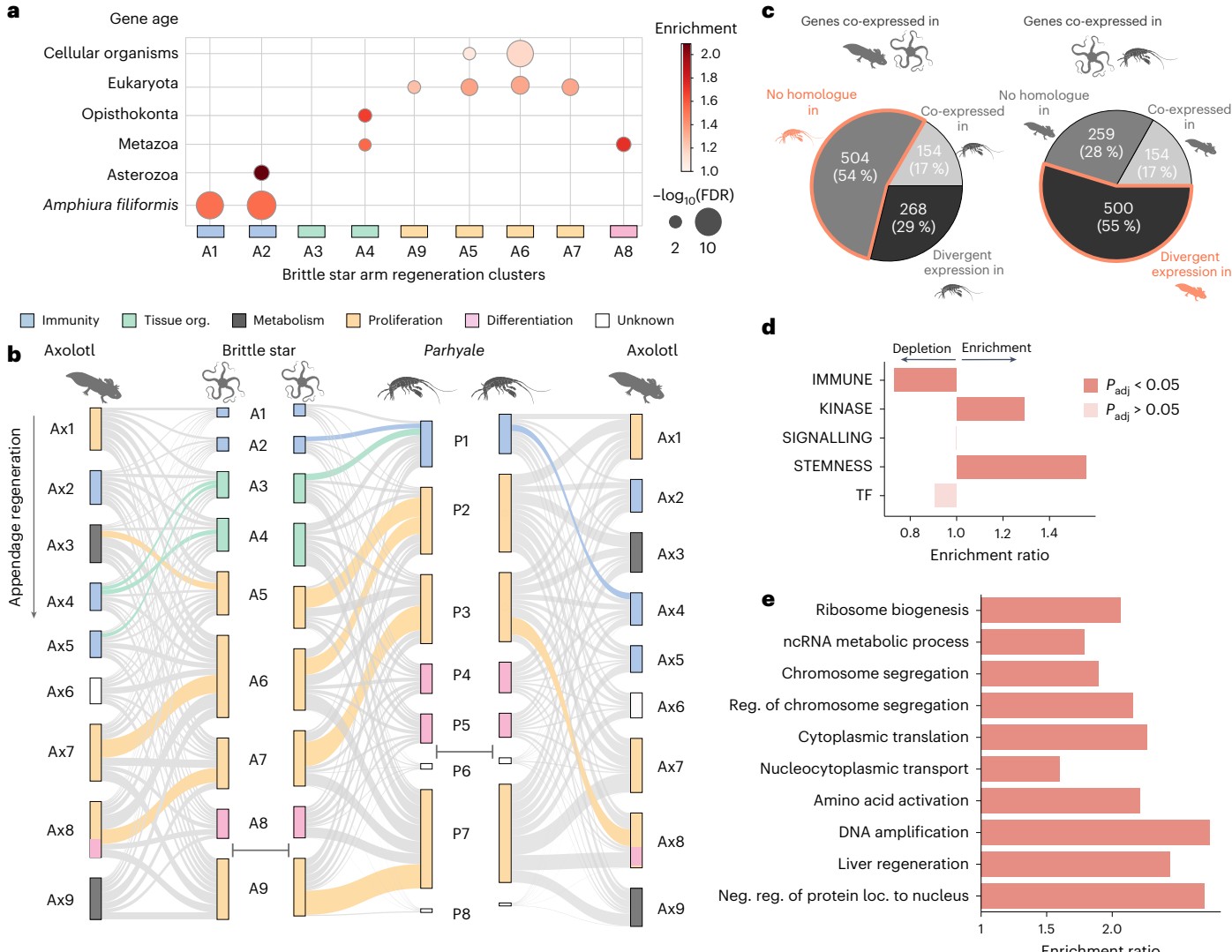

**Fig. 5 | Gene expression throughout appendage regeneration across animals. a**, Gene age enrichments for brittle star arm regeneration clusters (hypergeometric test, Benjamini–Hochberg $P_{adj} < 0.05$). Clusters are ordered by the time of expression onset. **b**, Comparison of co-expressed gene clusters deployed during appendage regeneration in axolotl, brittle star and Parhyale (left to right: axolotl vs brittle star, brittle star vs Parhyale, Parhyale vs axolotl). Clusters in Parhyale (clusters P1–P8) correspond to the clustering reported previously[48], but clusters were renamed to follow temporal activation and homogenize with respect to brittle star and axolotl clusters (Methods). Co-expression clusters in each species are shown in order of their temporal expression (from top to bottom), except for brittle star cluster A9 and Parhyale clusters P6–P8 which are expressed throughout several regeneration time points and shown at the bottom. Clusters are represented by vertical rectangles whose sizes are proportional to the number of homologous genes in the cluster,

and coloured according to enriched GO terms (Methods, Fig. 4 and Extended Data Fig. 8; see **a** for legend). Links between clusters of the two compared species indicate cluster membership of homologous genes, with coloured links indicating significant overlaps (permutation-based $P$ values with Benjamini–Hochberg correction <0.05; Methods). Credits for Parhyale silhouette: Collin Gross (CC BY 3.0). **c**, Most genes identified as co-expressed in the brittle star–*Parhyale* and brittle star–axolotl comparisons are not recovered in the direct *Parhyale*–axolotl comparison. Most genes co-expressed in the axolotl and brittle star have no identified homologues in Parhyale (54%, left pie chart). Genes co-expressed in Parhyale and the brittle star have a divergent expression in the axolotl, that is, they are not found in matched co-expression clusters (55%, right pie chart). **d**, Gene list enrichment and depletion tests performed for the set of brittle star genes with conserved temporal expression during animal regeneration (Methods). **e**, GO enrichment tests, as in **d**.

factor-binding motifs found in the vicinity of brittle star co-expressed genes are also overrepresented near Parhyale and axolotl co-expressed genes, only YY1 and NRF1 are present in corresponding co-expression clusters (Ax7–A6) (Extended Data Fig. 8), suggesting a possible conserved role for these transcription factors in regulating cell proliferation during regeneration in these distantly related organisms.

Finally, we find that two temporally matched gene expression clusters in brittle star and Parhyale regeneration include key genes involved in repressing transposable elements (that is, *Risc-like* (A2-P1) and *Ago2-like* (A9-P7)) (Supplementary Table 13). It has been proposed that transposon repression is important for proceeding from the immune

response phase to regeneration[103], by preserving genome integrity for cell proliferation and differentiation. In line with this hypothesis, we found a higher transcriptional activity of brittle star repetitive elements in the initial wound-response regeneration phase compared with the proliferative phase (Extended Data Fig. 10 and Methods).

## Expression in non-regenerative and regenerative responses

We have comprehensively characterized the genome-wide gene expression dynamics during brittle star arm regeneration. However, this does not allow us to directly interrogate the molecular drivers of

regenerative as opposed to non-regenerative wound-healing responses. To tackle this question, we performed explant experiments in which the arm is first amputated from the body (proximal cut) and subsequently amputated at the distal end (Fig. 6a). As in whole animals, explanted brittle star arms regenerate from the distal tip, whereas the proximal end undergoes a non-regenerative wound-healing response. To identify genes specifically involved in regeneration, we sampled distal, medial and proximal explant segments for RNA-seq experiments at 3 and 5 dpa (3 to 4 replicates, for a total of 20 samples; Fig. 6a and Supplementary Table 1).

We tested for differential expression of genes at the distal and proximal end compared to control medial segments (Methods and Fig. 6a). As expected, upregulated distal genes correspond to genes expressed during the proliferative phase of the brittle star arm regeneration time series, whereas upregulated genes in proximal segments correspond to early-response/wound closure genes (Fig. 6b). We identified more differentially expressed genes (DEGs) in the distal regenerating samples than in proximal non-regenerating samples (distal: 595 and 828 upregulated genes at 3 and 5 dpa respectively, 238 and 562 downregulated; proximal: 148 and 373 upregulated, 27 and 97 downregulated) (Fig. 6c). Most genes differentially expressed in proximal segments are also differentially expressed in distal segments (61% of the proximal DEGs are shared with distal), whereas distal genes are largely distal specific (82% of the distal DEGs are not shared with proximal) (Fig. 6c). This is consistent with the expected expression patterns, as wound closure is an integral part of regeneration. Altogether, we identify hundreds of differentially expressed candidate genes (Supplementary Table 2).

Notably, five genes display drastically opposite expression patterns in the wound-healing and regenerating segments (Fig. 6c) and are thus likely to contribute to distinct post-wounding outcomes. *Agrin-like-1* and *AFI33635* are significantly downregulated during wound healing but upregulated in regeneration. Agrin proteins are critical for neuromuscular junction development in vertebrate embryogenesis[104]. *AFI33635* is an uncharacterized brittle star gene with thyroglobulin and methyltransferase domains, putatively involved in regulating protease activity[105]. Conversely, the three genes *AW-SPI*, *AFI18858* and *Gdf8* are significantly upregulated during wound healing but downregulated in regeneration. *AW-SPI* is an antistasin/WAP-like serine protease inhibitor, with a possible role in immune defence[106]. *AFI18858* is a brittle star gene with a zf-Bbox domain and is a member of the expanded TRIM-like gene family, broadly involved in immune responses (Fig. 3c). Interestingly, the myostatin gene *Gdf8* is a member of the TGFβ signalling pathway that inhibits skeletal muscle growth and regeneration in mice[107,108]. Repression of *Gdf8* may similarly enable muscle regeneration in brittle stars. In summary, these five candidate genes might be tightly linked with the transition from wound healing to regeneration-induced cell proliferation, and some may have a conserved function in the brittle star and in vertebrates (*Agrin* and *Gdf8*).

## Discussion

The chromosome-scale genome of the brittle star *A. filiformis* represents a critical resource for the fields of evolutionary genomics, marine ecology and regenerative biology. Whereas previous studies of chromosome evolution in echinoderms were limited to sea urchins[23,51], our analyses revealed that the genomes of sea cucumbers and sea stars display even fewer rearrangements of the bilaterian ancestral chromosomal units than that of sea urchins. We showed that the 'Eleutherozoa Linkage Groups' descend from a single fusion of ancestral bilaterian linkages (B2 + C2). Chromosome-scale crinoid and hemichordate genomes will reveal whether this fusion is ancestral to Ambulacraria. Crucially, the fusion has not been observed in the genome of *Xenoturbella bocki* whose phylogenetic position is controversial, and thus cannot be used to support their proposed grouping with Ambulacraria[109,110]. The *A. filiformis* genome is highly rearranged: our analyses identified 26 interchromosomal rearrangements since the

Eleutherozoa ancestor. Additional brittle star genomes will reveal the precise timeline of chromosomal rearrangements and contributions of repeat expansion, chromatin architecture and population genetics dynamics to the rapid karyotype evolution in this group.

On a more local scale, we identified convergent rearrangements in the Hox clusters of sea urchins and the brittle star, which could be hallmarks of relaxed regulatory constraints within echinoderms. Hox genes, and in particular anterior Hox, show limited expression during echinoderm embryogenesis and are mostly expressed in adults[60,66–68,111]. We speculate that anterior and central/posterior Hox genes may belong to distinct chromatin compartments in echinoderms. Small-scale rearrangements may have occurred through elevated physical contacts at compartment boundaries (that is, around *Hox4*) and eventually become fixed owing to relaxed selection constraints on Hox expression. We revealed expansions of transposable elements in the brittle star Hox cluster ~100 Ma. If Hox cluster rearrangements co-occurred with the activation of repeats, distantly related brittle star species[11] may exhibit distinct Hox organizations.

The brittle star genome furthermore enables genetic characterization of the animal appendage regeneration process and remarkably allows the detection of long-range conservation of gene expression programmes. Incorporating the brittle star within a comparative transcriptomics framework extensively increased our ability to detect conserved co-expression modules between vertebrates (for example, axolotl) and arthropods (for example, Parhyale). We revealed that the proliferative phase of regeneration displays the highest expression conservation across these animals, suggesting that regeneration deploys an ancient, evolutionarily conserved proliferation machinery. These results are consistent with two alternative scenarios for the evolution of animal regeneration: (1) convergence, with the independent evolution of wound response programmes able to recruit the ancestral proliferative machinery or (2) homology, with an elevated divergence of wound response gene expression through diversifying selection, as typical for immune-related genes. The stronger conservation of gene expression during proliferation as opposed to the initial wound-healing response is consistent with the elevated turnover of immunity-related genes, broadly reported across animal lineages[73–75] and which we also demonstrate here in echinoderms. Our results, however, contrast with the only previous study to have explicitly interrogated the conservation of animal regeneration gene expression programmes, which revealed a higher conservation of early-response genes as opposed to the genes expressed during proliferation[36]. These discrepancies might be due to limited and asynchronous temporal sampling across species in previous comparisons[36], which is alleviated in our study through more comprehensive samplings of regeneration time points. Alternatively, they could reflect genuine biological differences between (larval) whole-body regeneration studied previously[36] and adult appendage regeneration. We nevertheless expect that future investigations into diverse regenerating animals with comprehensive temporal sampling will confirm the strong conservation of proliferation gene expression dynamics. Denser temporal samplings of early regeneration are necessary to confirm the limited conservation that we observe here but are currently technically challenging in the brittle star model. The conservation of proliferation ties in with a current hypothesis in the field that animal regeneration may recruit a homologous proliferating cell type[33,34], but this should also be further explored with single-cell sequencing techniques and additional comparative analyses.

Finally, in the brittle star *A. filiformis*, we identify notable expansions of gene families linked to regeneration-related processes and in particular, of homologues of vertebrate coagulation regulator genes, suggesting them as relevant candidates for follow-up in-depth functional characterizations. We also propose a conserved role for *Gdf8* during regeneration, as it is repressed during regenerative proliferation in both brittle stars and mice[107,108]. Our findings emphasize the importance of echinoderms as a powerful model for regeneration owing to

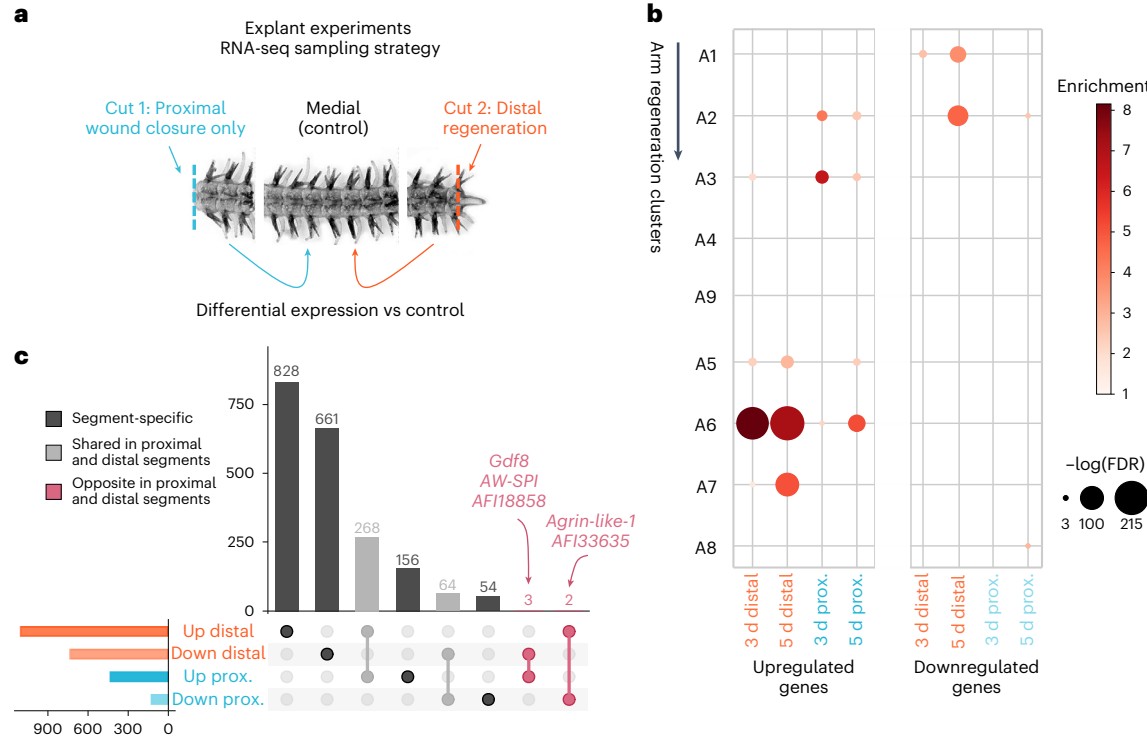

**Fig. 6 | Comparison of gene expression during wound closure and regeneration in brittle star explant experiments. a**, Experimental setup. Brittle star arms are amputated at the proximal (cut 1) and distal (cut 2) ends. Proximal, distal and medial (control) segments are sampled for RNA-seq at 3 and 5 dpa, using 3–4 replicates each (Supplementary Table 1). We identify DEGs in proximal (wound closure only, not followed by regeneration) segments and distal (regenerative) segments, compared to control medial segments. **b**, Comparison of DEGs from explant experiments with brittle star arm regeneration time-course clusters (Fig. 4; hypergeometric enrichment test, BH-corrected *P* < 0.05). **c**, Overlap between DEGs genes in distal and proximal segments. Bars in the UpSet plot are coloured to highlight (i) segment-specific DEGs, for DEGs unique to distal or proximal segments, (ii) shared proximal and distal segments, for DEGs shared between proximal and distal, and (iii) opposite proximal and distal segments, for DEGs upregulated in proximal and downregulated in distal (or vice-versa).

their unique regenerative capabilities and experimental amenability, but also to their phylogenetic position crucial for comparative analyses. The extensive genomic and transcriptomic resources we generated for the brittle star *A. filiformis* thus represent an entry point for future studies aiming to understand the evolutionary, molecular and genetic underpinnings of animal appendage regeneration, emergence of pentameral symmetry and remarkable diversity of morphologies and developmental strategies seen across echinoderm lineages.

## Methods

### Animal sampling
Adult *A. filiformis* were collected at 25–40 m depth from sediment in the Güllmarsfjord in the vicinity of Kristineberg Marine Station, Sweden, using a Petersen mud grab. Individuals were separated from the sediment by rinsing them with seawater, and then maintained in natural flowing seawater at 14 °C. Sperm was collected from a single individual by dissecting the gonads from the bursae.

### DNA extraction and sequencing
Sperm cells were concentrated by centrifugation, washed repeatedly and subsequently embedded in 2% low-melting agarose. Sperm cells were lysed in a solution of 1% SDS, 10 mM Tris (pH 8) and 100 mM EDTA and then resuspended in a solution of 0.2% *N*-laurylsarcosine, 2 mM Tris (pH 9) and 0.13 mM EDTA. High molecular weight DNA was released from the agarose blocks using β-agarase (NEB).

Long-read sequencing was performed on six Nanopore PromethION flowcells (v.R9.4.1). Several libraries were constructed using the ligation sequencing kit (Nanopore LSK109) using DNA sheared to different sizes using a megaRuptor (Diagenode) to optimize yield and contiguity. Bases were called from raw signal with Guppy (model 'dna_r9.4.1_450bps_hac_prom', v.2.3.5). A total of 160.56 Gb nanopore reads was acquired (~100× coverage). A library of 10× linked reads was generated using the Chromium system (10x Genomics) and sequencing on a Novaseq6000 SP lane in a 2 ×150 bp layout for a total of 246 M reads (86 Gb). Genome size was estimated to 1.33 Gb with a heterozygosity of 3.22% by counting *k*-mer (*k* = 31) in the short-read data using jellyfish2 (ref. 112) and fitted through a four-peak model using Genomescope2 (ref. 113).

### Genome assembly
We assembled Nanopore reads using flye (v.2.9-b1768)[114] assuming a coverage of 30× and a genome size of 3 Gb to account for the high level of heterozygosity. We obtained a diploid assembly of 2.86 Gb (N50: ~2.78 Mb), which was subsequently polished using Racon (v.1.5.0)[115] for two iterative rounds using the nanopore reads and then for another two rounds using the short-read Illumina reads that were aligned to the assembly using minimap2 (v.2.24-r1122)[116]. The flye assembly had *k*-mer completeness and QV base accuracy of 97% and 31.6 (that is, 0.000683556 error rate), as reported by Merqury (v.1.3)[117]. Structural accuracy was verified with Inspector (v.1.0.2)[118], revealing a read-to-contig mapping rate of 97% and a structural quality value QV of 26.88 (0.002 error rate). Haplotypes were then removed from the assembly using purge_dups (v.1.2.5)[119], with cut-offs visually adjusted from the coverage distribution on contigs. Correct haplotype removal was further verified by inspection of *k*-mer spectrum plots[117] (Extended Data Fig. 1a,b). The resulting assembly had a total length of 1.57 Gb, with

N50 and L50 (number of scaffolds containing half the genome assembly) of 3.2 Mb and 154, respectively, and 96.1% complete BUSCO score.

To scaffold this assembly, we built a Hi-C library from gonadal tissue using the Omni-C kit (Dovetail). Chromatin was fixed using paraformaldehyde and digested using a sequence-independent nuclease after re-ligation and biotinylation. A sequencing library was built from purified DNA and 225 M reads sequenced on a Novaseq X (~45× coverage). Hi-C reads were mapped to the polished haplopurged assembly using bwa mem (0.7.17-r1198-dirty) with options -5SP -T0, and alignments were further sanitized, sorted and duplications removed using pairtools (v.1.0.2)[120] with options '–walks-policy 5unique', '–max-inter-align-gap 30' and a minimum MAPQ of 40. We used YAHS (v.1.1a-r3)[121] to scaffold the genomic contigs using the Hi-C read alignment as input. We obtained 20 main chromosome-scale scaffolds totalling 1.47 Gb, corresponding to 93.5% of the total assembly length. The 20 chromosomes were strongly supported by the Hi-C contact map (Extended Data Fig. 1c) and also recovered with a perfect one-to-one match using an alternative assembly methodology (3D-DNA)[122]. The GC level of the final genomic sequence was 36.67% and the N50 was 68.86 Mb.

## Repeat annotation
We used RepeatModeler 2.0.2 to build a de novo repeat library for the brittle star genome and RepeatMasker 4.1.2-p to soft-mask the genome[123]. We used DeepTE[124] to classify repeats that could not be classified with the RepeatModeler homology-based classification. We retrained a DeepTE model to classify metazoan repeats into 5 classes, using a balanced dataset of 12,500 distinct repeats (2,500 repeats for each of the 5 classes) from different sources including repbase[125], Dfam[126] and homology-based classifications of repeats from 17 echinoderm and 2 hemichordate genomes (validation accuracy = 0.98 at the class probability threshold $P \geq 0.55$; Extended Data Fig. 1d). On a test set of 827 brittle star repeat families that were not included in the training set and where RepeatModeler homology-based predictions serve as ground truth, this retrained DeepTE model has higher accuracy than the default Metazoa model available in DeepTE (accuracy = 0.81 vs 0.67; Extended Data Fig. 1e). Divergence to consensus (kimura %) were computed and repeat landscapes plotted using the 'calcDivergence.pl' and 'createRepeatLansdscape.pl' scripts from RepeatMasker. The same methodology was applied to build repeat landscapes for *P. lividus*, *H. leucospilota* and *M. glacialis*. Repeat annotations are provided in dataset_s1 of ref. [127]. Repeat ages were estimated from divergence to consensus using a neutral substitution rate of $1.885 \times 10^{-9}$ per base pair per year for *A. filiformis*, which was estimated with phyloFit[128] from an alignment of 66,818 4-fold degenerate sites containing 17 echinoderm and 2 hemichordate genomes.

## RNA isolation, extraction and sequencing
**Arm regeneration RNA-seq in brittle star (time course in whole animals).** *Amphiura filiformis* individuals were obtained in the fjord close to the Kristineberg Center for Marine Research and Innovation, Sweden, at depths of 20–60 m. Samples of different regenerating stages were obtained as previously described[42] for early regeneration stages (48 hpa, 72 hpa, stages 3, 4 and 5) and as described[47] for 50% differentiation index stages (50% P and 50% D). Thirty regenerates from different individuals were used per stage. Dissection for RNA sampling was performed as follows (Fig. 4a): (1) for the non-regenerating control, we dissected one mature arm segment, (2) for 48 and 72 hpa samples, we dissected the last segment at the amputation site, (3) for stages 3 to 5, we dissected the regenerative tissues and (4) for 50% regenerates, we sampled several segments of proximal and distal tissues, excluding the differentiated distal cap structure. The collected regenerates were lysed in 10 volumes of RNA lysis buffer (RLT) (Qiagen) and total RNA extracted using RNAeasy micro RNA kit (Qiagen). RNA concentration and integrity were measured using Bioanalyzer (Agilent). Library preparation and paired-end sequencing was conducted by Novogene.

**Arm regeneration RNA-seq in brittle star severed arm experiments (explant).** We collected ~3,500 brittle stars with a 5–7 mm disc diameter. While animals were sedated in 3.5% w/w MgCl$_2$ in artificial seawater, two arms from each organism were amputated by pressing a scalpel blade into the intervertebral autotomy plane. We first sectioned the arms 0.5 cm from the disc (amputation 1, Fig. 6a) and then sectioned them again at the distal end (amputation 2, Fig. 6a). We thus produced explants (that is, severed brittle star arms) of 1 cm length with wound sites at the proximal and distal ends. Twenty samples (each sample consisting of a batch of 150–200 explants) were cultured in flow-through aquaria at 16 °C. Explants were sampled at 3 and 5 dpa, sedated in 3.5% w/w MgCl$_2$ in artificial seawater for 15 min and then dissected into three sections: proximal, medial and distal (Fig. 6a and Supplementary Table 1). Each explant section was flash frozen in liquid nitrogen and collected in batches of 150–200 pieces. Each batch was individually homogenized with glass pistils and RNA was extracted with the RiboPure kit (Applied Biosystems), following manufacturer protocol. RNA concentrations were measured using a QuBit 2.0 RNA fluorometric assay (Thermo Fisher) and RNA integrity was checked using 0.5% (w/v) agarose-MOPS-formaldehyde denaturing gel electrophoresis.

Complementary DNA (cDNA) libraries were prepared using the Illumina TruSeq v2 mRNA sample prep kit (Illumina), following a standard protocol. Briefly, mRNA was isolated with poly-A selection, followed by cDNA synthesis, Illumina standard index adapter ligation and a brief PCR reaction. Concentrations of the cDNA libraries were measured using a QuBit DNA high-sensitivity assay (Thermo Fisher) and fragment length distributions were assessed using an Agilent TapeStation with a D1000 tape (Agilent). cDNA libraries were multiplexed by equimolar pooling (5 or 6 samples per pool) and then sent to the Swedish National Genomics Infrastructure's SNP & SEQ platform in Uppsala for Illumina HiSeq 2500 sequencing (8 lanes; 126 bp paired-end sequencing; Illumina).

## Gene annotation
We annotated the brittle star genome using three types of evidence: (1) assembled transcriptomes from 18 samples, some published previously[46,61,62] and some newly generated (Supplementary Table 1), (2) similarity to proteins from 27 selected Metazoa and (3) ab initio predictions. We implemented a genome annotation pipeline[129] combining state-of-the-art tools. Implementation details are described in Supplementary Note 3. This annotation had a score of 92.7% complete BUSCO [C:92.7 (S:86.2%, D:6.5%), F:5.0%, M:2.3%, n:954][130] and a total of 4,974 unique PFAM domains[131], with 76% of genes (23,047) containing a PFAM domain. Annotation files are provided in dataset_s1 of ref. [127].

## Synteny comparisons and Eleutherozoa ALGs
For the sea urchin *P. lividus* and the black sea cucumber *H. leucospilota*, we used previously reported gene annotations[19,23]. We generated a draft homology-based annotation for the spiny sea star *M. glacialis*[24] with MetaEuk (6-a5d39d9)[132] using proteins of the sea urchin *S. purpuratus* (Spur_5.0, available in Ensembl Metazoa (v.56)[14], the crown-of-thorns sea star *Acanthaster planci* (OKI_Apl_1.0, available in Ensembl Metazoa (v.56)[30]) and the octopus sea star *P. borealis*[56]. One-to-one orthologous genes were identified by reciprocal best blast hit between pairs of compared genomes, using diamond[133]. We used Circos v.0.69.8 and circos-tools (0.23)[134] to plot synteny comparisons, with the bundle-links tool to group together neighbouring genes (maximum gap of 50 genes), filtering out bundles with fewer than 3 links. Chromosomes were ordered using the orderchr tool. The ancestral Eleutherozoa linkage groups were reconstructed on the basis of synteny comparisons between the spiny sea star *M. glacialis* and the black sea cucumber *H. leucospilota*, and with the amphioxus *B. floridae* and the scallop *P. maximus* genomes as well as previously defined bilaterian linkage groups (BLGs) (Extended Data Fig. 2). Only one macrosyntenic rearrangement occurred between the spiny sea star and the black sea cucumber: (a) spiny sea star chr5 maps to both sea cucumber chr12 and

chr23. Comparisons with outgroups and ancestral BLGs revealed that (a) corresponds to a derived fusion in the spiny sea star and that the black sea cucumber retained the ancestral state. Using this reconstruction, we annotated genes from matched orthologous chromosomes between sea stars and sea cucumbers with respect to their ancestral ELGs of origins and propagated annotations to orthologous genes in *P. lividus*, *A. filiformis* and other available chromosome-scale echinoderm genomes. Karyotypes were drawn with RIdeograms[135]: we painted genes on extant chromosomes using the ancestral chromosome colour when a significant number of genes were inferred to descend from an ancestral chromosome ($P < 10^{-5}$, Fisher exact tests corrected for multiple testing with the Benjamini–Hochberg procedure). Oxford grid plots in Extended Data Fig. 3 were plotted using the same statistical thresholds. ELG-related data files are provided in dataset_s2 of ref. 127.

### Hox and ParaHox genes identification

We identified Hox and ParaHox genes using sequence comparisons with other echinoderm and animal genomes and phylogenetic reconstruction. Detailed procedures are reported in Supplementary Note 3. Hox and ParaHox data files are provided in dataset_s3 of ref. 127.

### Gene families expansion and contraction

Gene phylogenies and history of duplication and losses for pmar1/phb and luciferase were reconstructed using RAXML-NG (v1.1)[136] and Treerecs[137] (Extended Data Fig. 5). We used broccoli[138] to group proteins of 28 selected Metazoa, 10 of which were Ambulacraria, into gene families. Out of the complete set of broccoli gene families, 10,367 originated before the last common ancestor of Ambulacraria (echinoderm and hemichordate outgroups). We used CAFE (v.5)[139] on the 10,367 families to identify significantly expanded and contracted gene families on each branch of the Ambulacraria phylogeny. To obtain a dated Ambulacraria phylogeny, we: (1) extracted 192 one-to-one orthologues from broccoli gene families, (2) built multiple sequence alignments for each orthologous group using MAFFT (v.7.475), (3) reconstructed a maximum likelihood phylogeny with RAxML-NG (v1.1)[136] using the concatenated alignment (LG + G4 + F model with 10 parsimony starting trees), (4) filtered out columns with over 15% gaps (47,520 retained sites) and (5) ran PhyloBayes (v.4.1b)[140] to obtain a time-calibrated tree, with the RAxML reconstructed tree as constrained topology and selected fossil calibrations extracted from the literature[13,141]. The chain was run for 4,166 samples and 3,500 were retained after burn-in to estimate the posterior distributions for node ages. We next ran CAFE in 2 steps: we estimated the lambda and alpha parameters of the 2-categories CAFE GAMMA model excluding the 128 gene families with the largest copy number differential and then ran CAFE on all families with these parameters fixed to test for significant contractions and expansions ($P < 0.05$). Fossil calibrations, dated species tree, gene families and CAFE output files are provided in dataset_s4 of ref. 127.

### Gene lists curation

We generated lists of immune, neuronal, signalling, kinase, transcription factors and stemness genes in *A. filiformis* (Supplementary Table 2) using a combination of PFAM domain annotation and lists of previously curated genes in echinoderms and other animal lineages. Further details of the procedure are provided in Supplementary Note 3.

### Gene Ontology and gene list enrichment tests

We used eggnog-mapper[142] to automatically annotate *A. filiformis* and *P. lividus* genes with GO terms from the Biological Process domain. The GO annotations were then transferred to the level of gene families. Specifically, for each family, we propagated all GO annotations associated with any *P. lividus* or *A. filiformis* genes as the complete set of GO annotations for this family. Hypergeometric tests for functional enrichments were then conducted with the enricher function of the ClusterProfiler R package[143], with custom foreground and background GO annotation

sets. For functional enrichment tests on expanded/contracted gene families (Fig. 3), tests were conducted at the level of gene families with expanded or contracted families as foreground and all gene families as background. For functional GO enrichment tests on regeneration co-expression clusters (Fig. 4), tests were conducted at the level of brittle star genes, using genes of a given cluster as foreground and genes of all clusters as background. We used false discovery rate (FDR) < 0.05 as significance threshold. Enrichment results were summarized with REVIGO[144]; we selected top ontology terms on the basis of REVIGO 'dispensability' score. Similarly, for gene list enrichment and depletion tests on the regeneration co-expression clusters (Fig. 4), we used the same foreground and background gene definitions as for the GO enrichment tests above. We performed hypergeometric tests with correction for multiple testing using the Benjamini–Hochberg procedure, with the same statistical threshold as for the GO enrichment tests (FDR < 0.05).

### Clustering of the arm regeneration expression series

Gene expression was quantified for all samples using the alignment-free method kallisto (v.0.48.0)[145]. We normalized TPM values across samples using the trimmed mean of m-values (TMM) method as implemented in edgeR[146,147] and used MFuzz (v.3.18)[148] to perform soft-clustering of genes on the basis of their standardized expression profiles across samples. We used the minimum centroid distance method to select the optimal number of clusters ($n = 19$; Extended Data Fig. 6). Major clusters were defined as all clusters with >1 enriched GO term and expression in >1 regenerating sample (Fig. 4 and Extended Data Fig. 6). Normalized gene expression tables are provided in dataset_s5 of ref. 127.

### Transcription factor-binding motif enrichment tests

We used HOMER (v.4.11)[149] to test for enriched transcription factor-binding motifs in the proximal regulatory domains (TSS + 5 kb upstream, +1 kb downstream) of genes of each regeneration cluster. We ran the findMotifsGenome.pl script from the HOMER suite, with −h to perform hypergeometric tests, contrasting proximal regulatory domains of genes from one expression cluster as foreground with proximal regulatory domains of genes from all clusters as background.

### Axolotl limb regeneration RNA-seq time course

Raw RNA-seq data for 12 limb regeneration time points from ref. 49 were downloaded from https://www.axolomics.org/?q=node/2. We used Trim Galore (https://github.com/FelixKrueger/TrimGalore) with default parameters to trim and quality filter raw sequencing reads via the Cutadapt tool[150]. Gene expression was quantified with kallisto (v.0.48.0)[145] using the set of annotated axolotl transcripts from the latest *Ambystoma mexicanum* assembly version (AmexG_v6.0-DD, available from https://www.axolotl-omics.org/assemblies ref. 151). We normalized TPM values across samples using the TMM method[146,147] and used MFuzz[148] to cluster genes according to their expression profile (Extended Data Fig. 7). Gene Ontology and transcription factor-binding sites (TFBS) motifs enrichment were performed as described in 'Gene Ontology and gene list enrichment tests' and 'Transcription factor-binding motif enrichment tests'. Normalized gene expression tables are provided in dataset_s5 of ref. 127.

### Parhyale limb regeneration RNA-seq time course

Parhyale leg regeneration expression data were previously processed and clustered into 8 co-expression gene groups using the same approach as we used for brittle star data[48]. We directly used the clustering reported previously[48] but renamed the clusters so that numbering follows temporal activation (P1 is R4 in the notation described previously[48], P2 is R1, P3 is R8, P4 is R2, P5 is R6, P6 is R3, P7 is R5 and P8 is R7).

### Comparison of gene expression dynamics

We used broccoli[138] to build homologous gene families encompassing genes of the brittle star *A. filiformis*, the axolotl *Ambystoma mexicanum* and *Parhyale hawaiensis*, as well as 8 echinoderms, 6 vertebrates, 7

 **1516**

ecdysozoans and 12 other animal genomes. We used these gene families to identify homologous genes and compare their expression profiles during appendage regeneration. We conducted pairwise comparisons, retaining all homologous gene families with >1 gene and <5 genes in each of the two compared species. This resulted in a total of 5,203 homologous groups retained for the axolotl (8,810 homologous genes)–brittle star (6,813 homologous genes) comparison, 3,137 for the brittle star (4,196)–Parhyale (3,617) comparison and 2,299 for the axolotl (3,903)–Parhyale (2,628) comparison (dataset_s5 of ref. 127). We next computed permutation-based $P$ values to test for the overrepresentation of homologous genes across co-expression clusters of the two compared species. Specifically, we generated, for each pairwise comparison, 10,000 randomizations of the gene labels of species 2, keeping clusters and orthologous gene family size constant to build a null distribution of the number of expected homologous genes shared by two clusters at random. Empirical $P$ values were computed from the null distribution and corrected for multiple testing using the Benjamini–Hochberg procedure. To investigate functional annotation of genes displaying co-expression across regeneration models as opposed to genes from the same clusters that do not show co-expression across species, we conducted gene list and GO enrichment tests as described in 'Gene Ontology and gene list enrichment tests' but using carefully selected background: we used as background all brittle star genes with a homologue in either Parhyale or axolotl (that is, whose expression conservation could be tested) and in a cluster with identified co-expressed genes in either Parhyale or axolotl (to test for the specificity of genes of a given cluster that show conservation vs those of the same cluster that do not).

### Differential analysis of repeats transcriptional activity

We tested for differentially expressed repetitive elements in early regeneration (immune phase: 48 hpa and 72 hpa samples) versus middle regeneration (proliferation: stage 3, stage 4, stage 5 samples), using our time course brittle star arm regeneration RNA-seq data. We used a conservative approach to first filter out highly duplicated genes which could have been captured in the set of repetitive elements called by RepeatModeler/RepeatMasker. We used diamond blastx[133] to search for homologies between repeat consensus and proteins in the swissprot database[152] and filtered out all 'Unknown' repeat families for which the consensus sequence had a strict match in swissprot ($e$-value cut-off $10^{-10}$), which did not correspond to transposon genes. We next used the SalmonTE pipeline[153] with default parameters on the full set of filtered repeat consensus ($n = 4,695$ repeat families), followed by differential analysis with DESeq2 (v.1.42.1)[154] on the estimated count values to test for differential transcriptional activity of repeats in the immune versus proliferation regeneration phases. We retained as differentially expressed the repetitive elements with an absolute $\log_2$ fold change >1, $P_{adj} < 0.001$.

### Differential gene expression in brittle star arm explants

Gene expression was quantified for all samples using kallisto[145]. Differential expression analyses were conducted with DESeq2 (ref. 154) on count values, contrasting distal replicates against medial replicates and proximal replicates against medial replicates for each time point. All genes with a $P_{adj} < 0.05$ and absolute $\log_2$ fold change >1 were retained as differentially expressed. Gene expression tables are provided in dataset_s5 of ref. 127.

### Reporting summary

Further information on research design is available in the Nature Portfolio Reporting Summary linked to this article.

## Data availability

Genome sequence and RNA-seq data have been deposited in NCBI SRA (Bioproject PRJNA1029566 and PRJNA1034116) and GEO (GSE246675). Supplemental datasets have been deposited in Zenodo[127] (see supplementary material for content details). These include the genome, gene and repeat annotations, processed gene expression tables and source data for the figures.

## Code availability

The code for the genome annotation workflow is publicly available[129].

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

## Acknowledgements

This work was supported by the Centre for Marine Evolutionary Biology at the University of Gothenburg (http://www.cemeb.science.gu.se/) and the IMAGO project led by Anders Blomberg. E.P. was supported by a Newton International Fellowship from the Royal Society (NIF\R1\222125). F.M. was supported by a Royal Society University Research Fellowship (URF\R1\191161) and a Japan Society for the Promotion of Science Kakenhi grant (JP 19K06620). L.P. was supported by the Leverhulme Trust Research Project (grant number RPG-2021-436 to F.M. supporting L.P.) and the BBSRC (grant BB/V01109X/1 to F.M. supporting L.P.). O.O.-M. was financially supported by CEMEB through grants from Swedish research councils Formas and VR (217-2008-1719) and from a VR grant to K.J. (253016979). J.D. was supported by an F.R.S.-FNRS research project (PDR, 40013965), previously held an F.R.S.-FNRS 'Chargé de recherche' fellowship (CR, 34761044), and also received financial support from an F.R.S.-FNRS research project (PDR, T.0169.20) and the Biosciences Research Institute of the University of Mons. P.M. visited the Department of Genetics, Evolution and Environment of UCL financed by a short-term scientific mission Grant of the EU COST Action MARISTEM (CA-16203). K.M.B. was supported by the National Science Foundation (NSF Award 2131297). P.O. visits to the Kristineberg Marine Station (Sweden) were supported by the KVA fund SL2015-0048 from the Royal Swedish Academy of Sciences and the EU FP7 Research Infrastructures Initiative ASSEMBLE (227799). P.O. was supported by the BBSRC (BB/W017865/1). The IAEA thanks the Government of the Principality of Monaco for the support provided to its Marine Environment Laboratories. We thank the staff at the Kristineberg Center for Marine Research and Innovation, especially U. Schwarz, for assistance during animal collection; A. Sabarí i Martí, E. Onal, L. Henke and W. Hart for assistance in conducting experiments. We acknowledge the Okinawa Institute of Science and Technology sequencing facility and its members for support, especially N. Arakaki and M. Kawamitsu. Finally, we thank J. Rast for helpful comments on the manuscript. The authors dedicate this manuscript to Michael Thorndyke and R. Andrew Cameron, who were both instrumental in the early stages of this project but, sadly, passed away before analyses were completed. Without their contributions, this work would not have been possible.

## Author contributions

O.O.-M., T.L., M.T., S.D. and K.J. initiated the genome sequencing project. O.O.-M., P.O., F.M. and E.P. designed the study. F.M. and P.O. generated sequencing data, and F.M. and E.P. assembled and annotated the final genome version. L.P. performed proximity ligation experiments. For transcriptomic analyses, O.O.-M. and T.L. designed the explant study and generated RNA-seq explant data, and A.C. collected regeneration time course samples. E.P. performed the computational analysis of synteny, gene family and comparative transcriptomics of regeneration with the support of F.M. D.D. and S.A. contributed to data analysis and visualization. O.O.-M., J.D., K.M.B., P.M. and P.O. curated the data and assisted with interpreting the results. E.P. wrote the manuscript with key contributions from F.M., P.M. and K.M.B. All authors commented on the manuscript and approved the final version.

## Competing interests

The authors declare no competing interests.

## Additional information

**Extended data** is available for this paper at https://doi.org/10.1038/s41559-024-02456-y.

**Correspondence and requests for materials** should be addressed to Elise Parey, Paola Oliveri or Ferdinand Marlétaz.

[1]Centre for Life's Origins and Evolution, Department of Genetics, Evolution and Environment, University College London, London, UK. [2]Tjärnö Marine Laboratory, Department of Marine Sciences, University of Gothenburg, Strömstad, Sweden. [3]Biology of Marine Organisms and Biomimetics Unit, Research Institute for Biosciences, University of Mons, Mons, Belgium. [4]Department of Biology and Environmental Science, University of Gothenburg, Kristineberg Marine Research Station, Fiskebäckskil, Sweden. [5]IAEA Marine Environment Laboratories, Radioecology Laboratory, Quai Antoine 1er, Monaco. [6]Department of Cell and Molecular Biology, National Bioinformatics Infrastructure Sweden, Science for Life Laboratory, Uppsala University, Uppsala, Sweden. [7]Department of Biological Sciences, Auburn University, Auburn, AL, USA. [8]Departament de Genètica, Microbiologia, i Estadística, Universitat de Barcelona, Barcelona, Spain. [9]Institut Català de Recerca i Estudis Avançats (ICREA), Barcelona, Spain. [10]Present address: Technische Universität Dresden, Center for Regenerative Therapies Dresden (CRTD), Dresden, Germany. [11]Present address: Roche Pharmaceutical Research and Early Development (pRED), Cardiovascular and Metabolism, Immunology, Infectious Disease, and Ophthalmology (CMI2O), F. Hoffmann-La Roche Ltd, Basel, Switzerland. [12]Present address: MRC Laboratory of Medical Sciences, Imperial College London, London, UK. [13]Deceased: Michael Thorndyke. ✉e-mail: e.parey@ucl.ac.uk; p.oliveri@ucl.ac.uk; f.marletaz@ucl.ac.uk

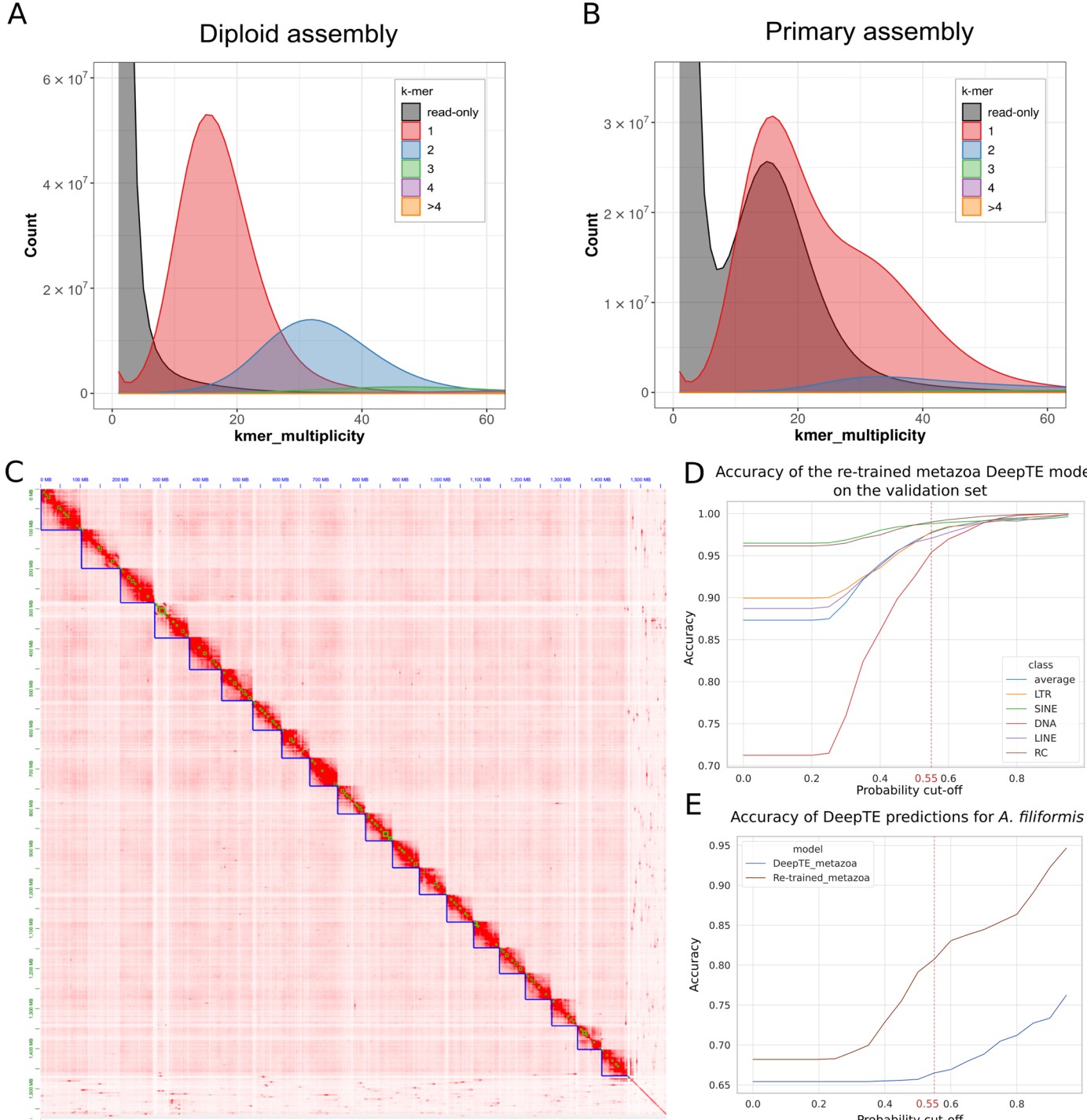

**Extended Data Fig. 1 | Genome assembly and repeat classification. A**. K-mer spectrum of the diploid assembly, that is before haplotype removal. Read-only k-mers (black curve) correspond to sequencing errors, and are not represented in the assembly. The 1-copy k-mer peak at 15X coverage (1n, red) corresponds to reads from heterozygous regions, whereas the 2-copy k-mer peak at 30X coverage (2n, blue) corresponds to homozygous regions. **B**. K-mer spectrum of the primary assembly, that is after haplotype removal. Following the collapse of haplotypes, half of the k-mers of the heterozygous peaks are accordingly not represented in the assembly anymore and homozygous regions are present as single copies only. **C**. Hi-C contact map showing the density of interactions between binned genomic regions in the proximity ligation data. The high contact regions are consistent with a 20 chromosome *A. filiformis* karyotype. **D**. Validation accuracy of a new DeepTE model[124], trained to classify repeats into 5 main classes: LTR, SINE, DNA, LINE and Rolling Circle (RC). The vertical dotted line corresponds to the calibrated 0.55 threshold that we used on the DeepTE scores to classify repetitive elements. **E**. Accuracy of the newly-trained and the default Metazoa DeepTE models on the test set of *A. filiformis* repeats. The accuracy of the new model is superior to the default model and can classify repeats into 5 as opposed to 3 classes (repeats of ClassI, ClassII and ClassIII).

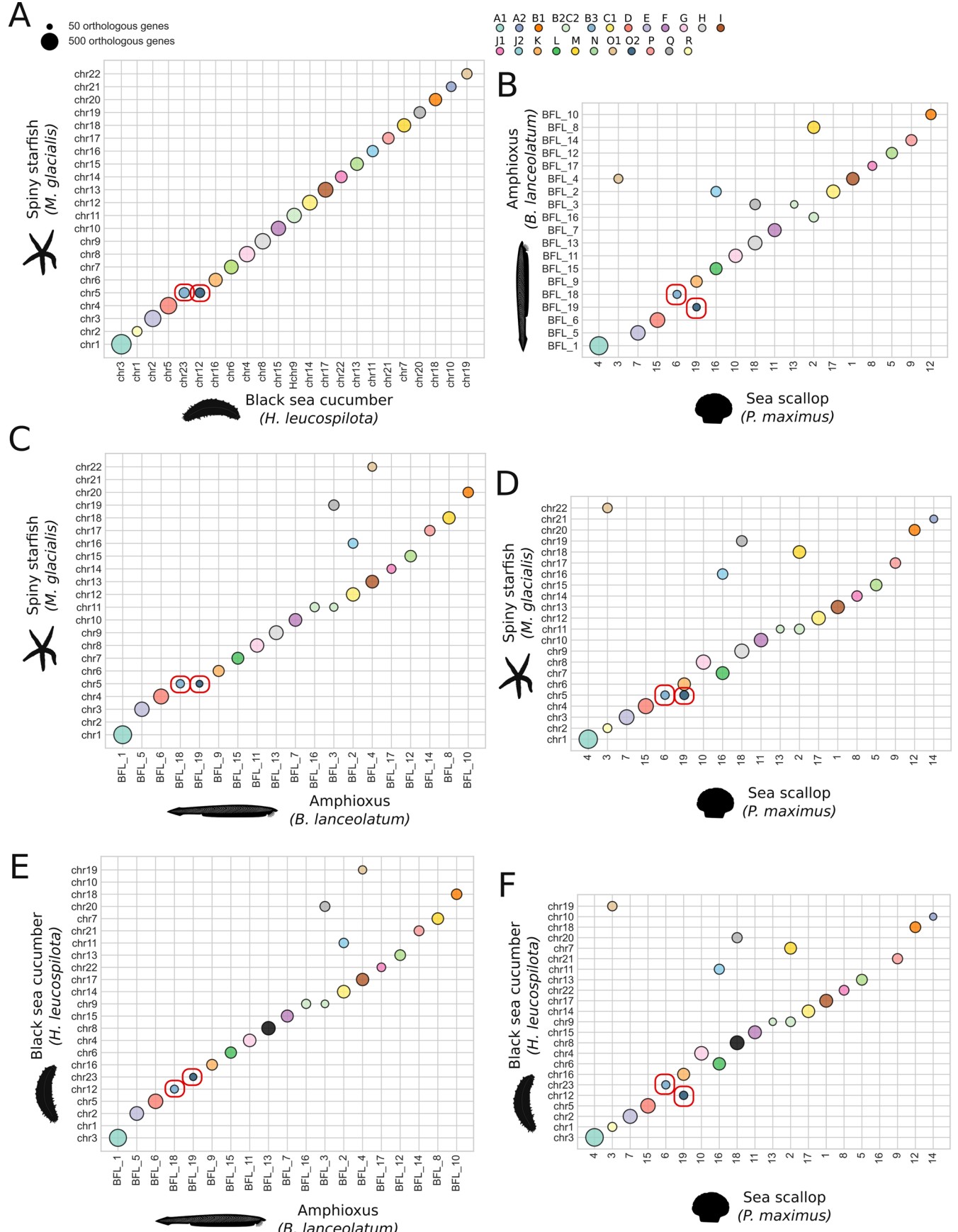

**Extended Data Fig. 2 | Reconstruction of the ancestral Euleterozoa linkage groups (ELG). A.** Synteny comparison between spiny starfish and black sea cucumber reveals one macrosyntenic rearrangement (red boxes). ELGs colours are indicated at the top and correspond to colours on Fig. 1. Pairwise synteny comparisons with Amphioxus and Sea Scallop are similarly displayed on **B.**, **C.**, **D.**, **E.** and **F.**, with red boxes highlighting that B3, and O2 are all on distinct chromosomes in Amphioxus and Sea Scallop, thus confirming that the sea star B3-O2 fusion is a sea star-specific derived rearrangement.

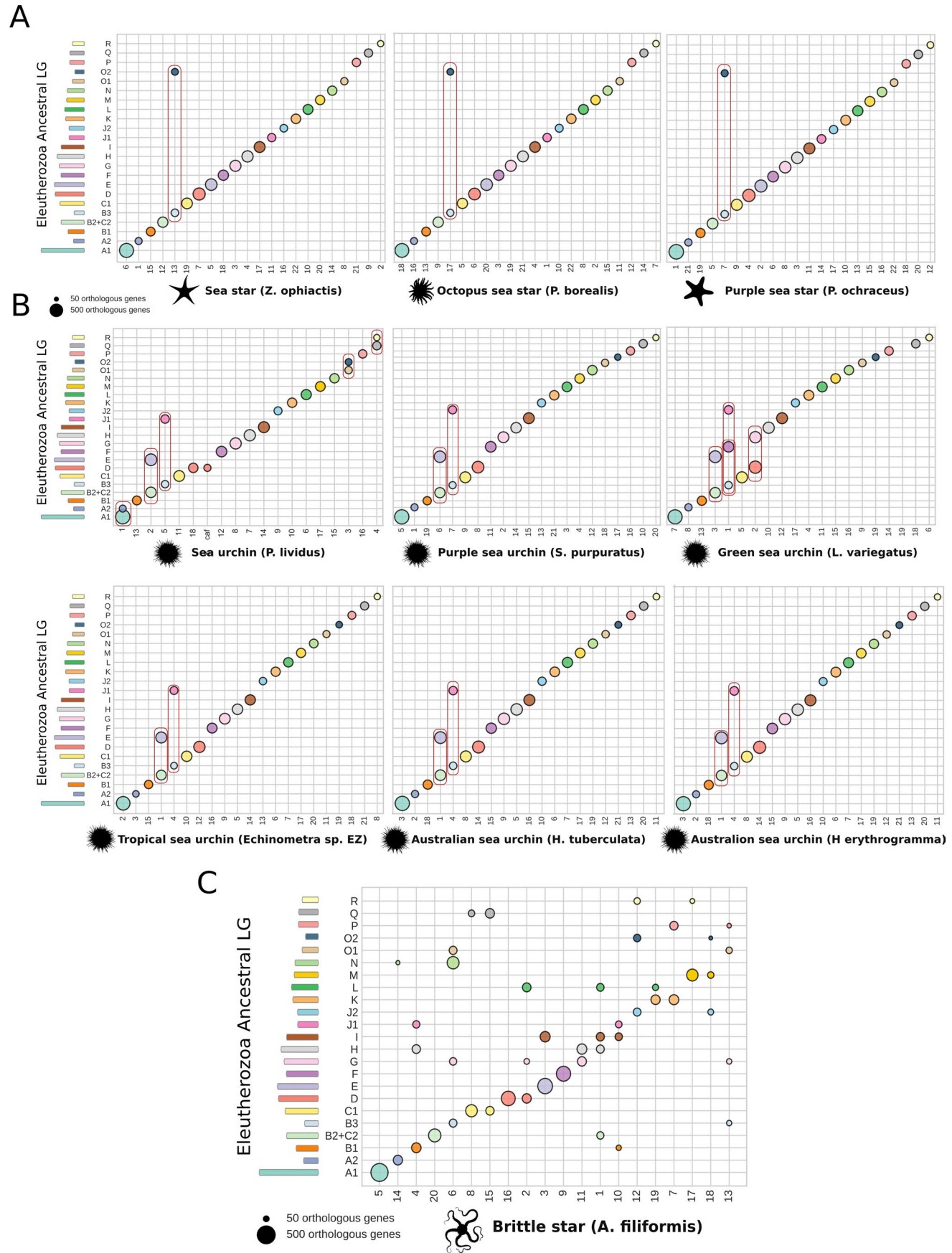

**Extended Data Fig. 3 | See next page for caption.**

**Extended Data Fig. 3 | Inter-chromosomal macrosyntenic rearrangements since the Eleutherozoa ancestor in sequenced echinoderms. A**. Synteny comparison between ELGs and available chromosome-scale sea star genomes [53,55,56]. All examined sea star genomes are marked by the single B3 + O2 fusion. **B**. Synteny comparison between ELGs and available chromosome-scale sea urchin genomes [14,20,21,23,57]. All examined sea urchin genomes are marked by the (B2 + C2) + E and B3 + J1 fusion. *L. variegatus* underwent the additional (B3 + J1) + J2 fusion and D + G. *P. lividus* underwent the additional A1 + A2, O1 + O2 and Q + R fusions (note that an additional fission of ELG D may have occurred if the large unplaced scaffold noted "Scaf." is not an assembly artefact.) **C**. Synteny comparison between ELGs and brittle star chromosomes reveals a total of 26 macrosyntenic inter-chromosomal rearrangements, in the most parsimonious scenario involving fusion, fission and translocation events. The rearrangements can be inferred from the oxford grid plot: 1 [fusion + mixing + fission] of 3 ELGs = 3 inter-chromosomal rearrangements (B3-G-O1), 3 [fusion + mixing + fission] of 2 ELGs = 6 inter-chromosomal rearrangements (B1-J1, C1-Q, J2-O2) and 17 translocations (A2-N, B1J1-H, B3GO1-N, D-G, DG-L, E-I, G-H, B2 + C2-H, B2 + C2H-I, B2 + C2HI-L, B1J1-I, J2O2-R, K-L, K-P, M-R, J2O2-M, B3GO1-P).

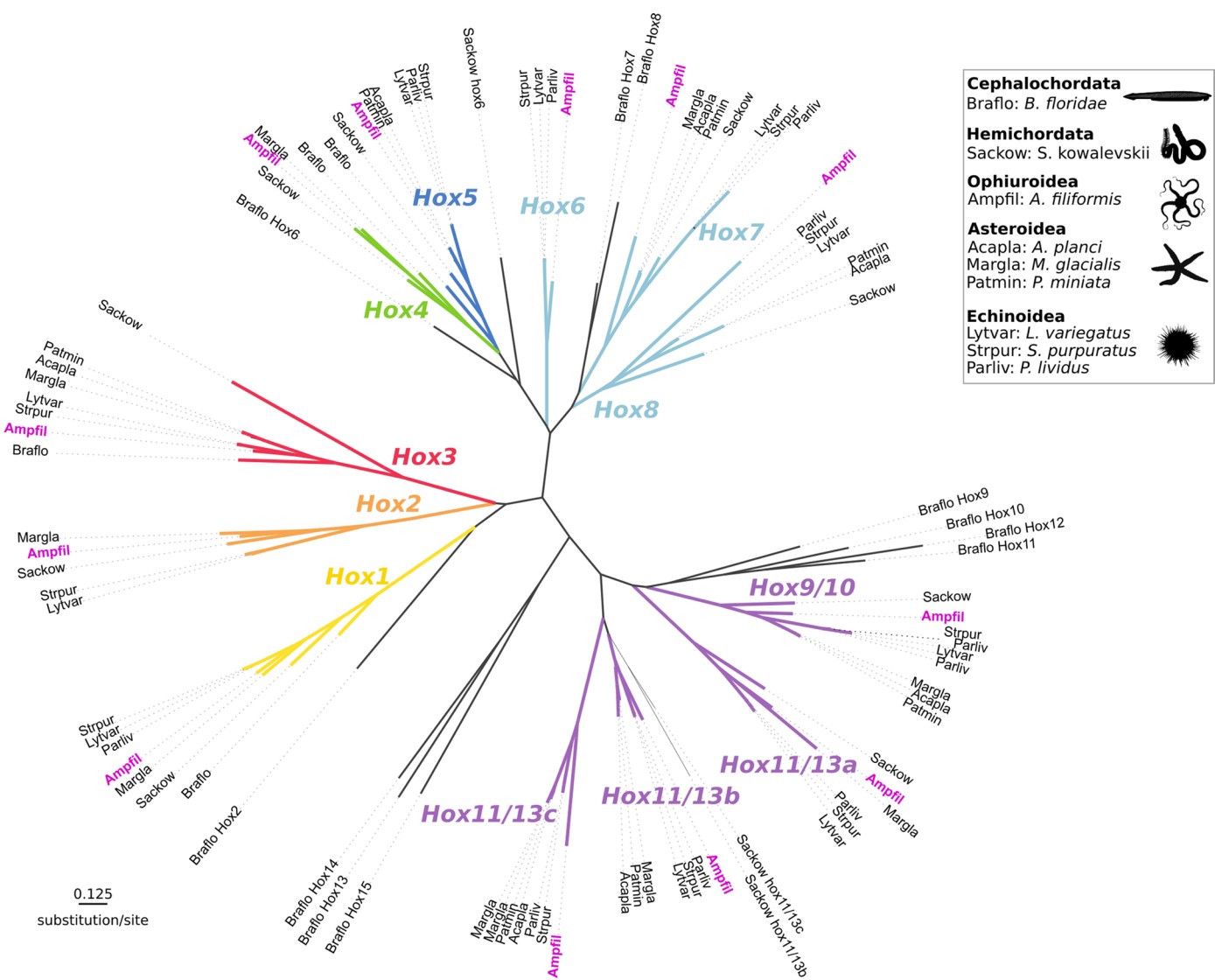

**Extended Data Fig. 4 | Molecular phylogeny of echinoderm Hox genes.** The phylogenetic tree is shown as an unrooted tree, with clades of Hox genes indicated with the same colours as in Fig. 2. The phylogenetic position of each identified Hox gene in the brittle star ("Ampfil") is highlighted in pink.

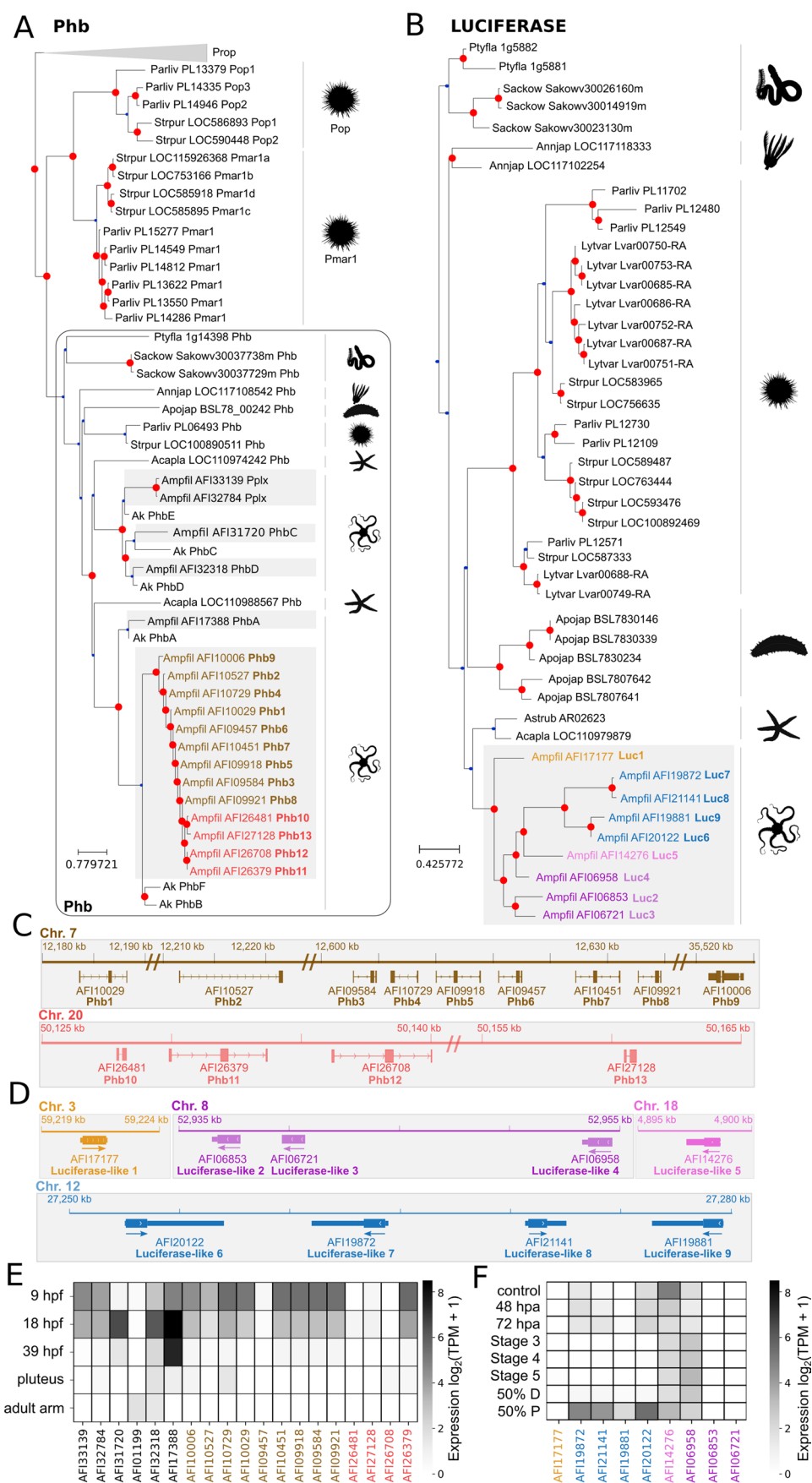

**Extended Data Fig. 5 | See next page for caption.**

**Extended Data Fig. 5 | Evolution of the p*mar1/phb* and *luciferase-like* genes by tandem duplications. A**. Molecular phylogeny of the *pmar1/phb* genes in echinoderms. The tree was reconstructed with RAxML-NG[136] (10 starting parsimony trees, 1000 bootstraps, LG + G4 + F model), lowly supported nodes (bootstrap < 60) were subsequently corrected with Treerecs[137] to maximise the parsimony of duplications and losses. Species are indicated by abbreviations (Ptyfla = *P. flava*, Sackow = *S. kowalevskii*, Annjap = *A. japonica*, Parliv = *P. lividus*, Strpur = *S. purpuratus*, Apojap = *A. japonicus*, Acapla = *A. planci*, Ak = *A. kochii*, Ampfil = *A. filiformis*). Inferred duplication nodes are shown in red. *pmar1/phb* full gene sequences were identified based on ref. 23,70 (Dataset_s4[127]). **B**. Phylogeny of luciferase genes in echinoderms, as in **A**. Luciferase-like genes were identified based on sequences from[8] (Dataset_s4[127]). **C**. Genomic location of tandem-duplicated *A. filiformis phb* genes. **D**. Genomic location of tandem-duplicated *A. filiformis* luciferase genes. **E**. *phb* expression throughout 4 brittle star developmental time points and in the adult arm, showing the early developmental expression of *phb* genes (hpf: hours post-fertilization). Expression across samples was normalised using the TMM method[146] on the full set of brittle star genes, and is shown as log2(TPM + 1). **F**. Luciferase-like gene expression during brittle star arm regeneration, showing that most luciferase-like genes are expressed in differentiated arms only: control arms and the latest regeneration time point (hpa: hours post-amputation, see Fig. 4 for staging details). Expression normalisation as in **E**.

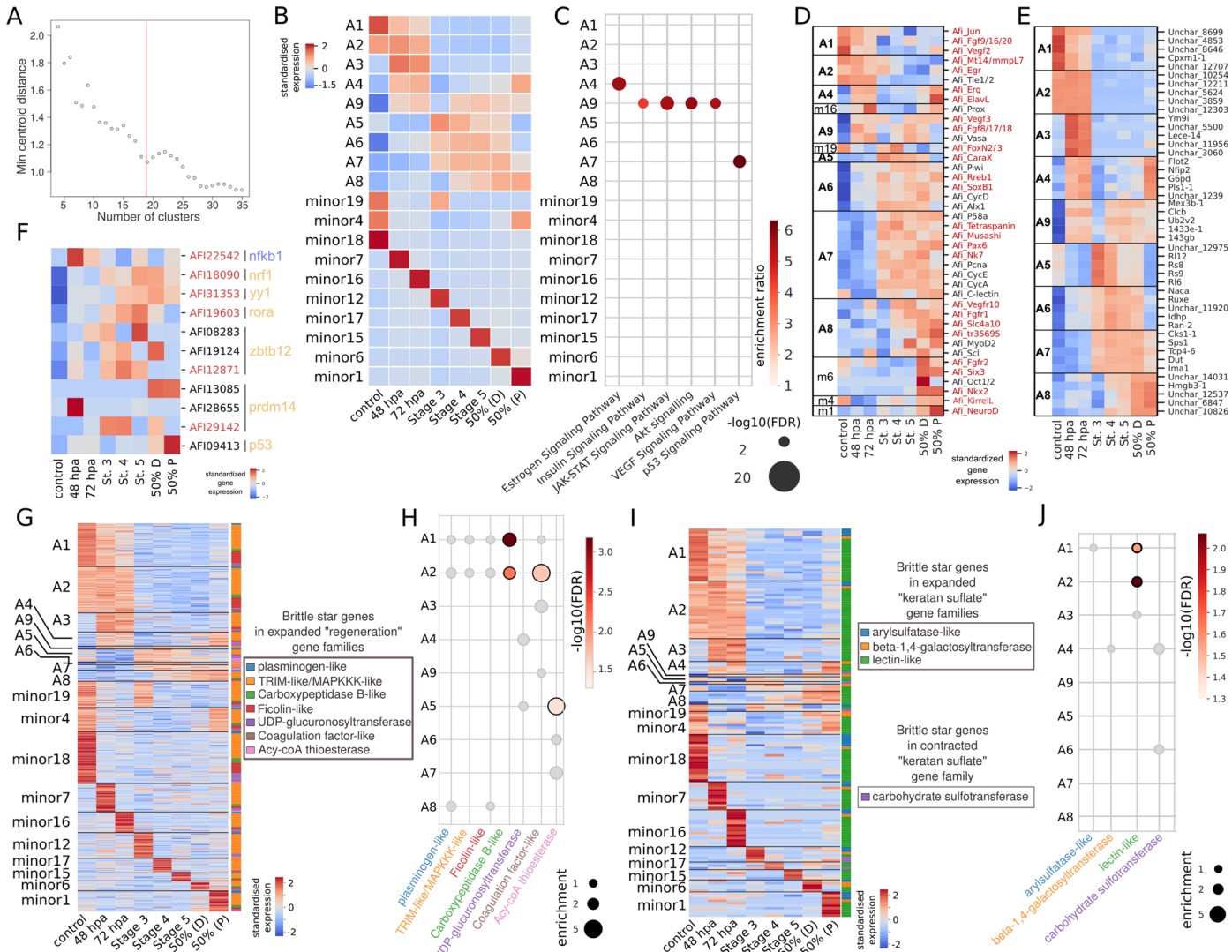

**Extended Data Fig. 6 | Clustering of gene expression during the brittle star arm regeneration. A.** Optimal number of clusters estimated using the centroid distance. After n = 19 clusters, there is no continuous decrease of the centroid distance. **B.** Normalised expression profiles (expression of the centroid) for each of the n = 19 clusters. Clusters with genes expressed over a single regeneration time point (or one regeneration point + control) were defined as minor clusters and not presented in the main text as these typically do not display significant enrichments and may be driven by noisy gene expression. **C.** Signalling pathways enrichment for each co-expression cluster (hypergeometric test, Benjamini-Hochberg adjusted p-values < 0.05, Methods). **D.** Expression of brittle star genes previously implicated in arm regeneration (gene names are from previous studies, see Supplementary Table 8). Co-expression clusters are shown on the left, gene names on the right, with red indicating availability of published *in situ* data. **E.** Expression of core genes in each co-expression cluster. Genes were filtered based on their cluster membership score (Supplementary Table 9, "acore" score) to retain the top 5% core genes in each cluster, and the five genes with the highest expression were selected for the heatmap. Gene names starting with 'Unchar' indicate genes without significant blast hits in the swissprot database. **F.** Expression of key TF genes during regeneration, as identified by binding motifs overrepresentation analysis (Fig. 4d). TF genes were identified by reciprocal blasts with mouse and swissprot blast hits; several

copies were reported where blast results were ambiguous. TF genes with consistent expression and binding motifs overrepresentation are shown in red. No homologue for ZNF268 could be identified in brittle star and the expression of the identified p53 homologue does not match motif enrichment results (but p53 pathway activation is consistent with p53 motif enrichments, see **C**). **G.** Expression throughout arm regeneration of genes in the expanded gene families annotated with the GO term 'regeneration' (see Fig. 3b,c). Gene family membership (correspondence with Fig. 3c) are indicated with colours on the right of the expression heatmap, clusters are shown on the left. **H.** Duplicated genes from expanded 'regeneration' gene families significantly associate with specific regeneration co-expression clusters (hypergeometric test, Benjamini-Hochberg adjusted p-values). Significant associations (FDR < 0.05) are presented in colour, non-significant enrichments (enrichment ratio > 1 but FDR > 0.05) in grey (Supplementary Table 7). **I.** Expression throughout arm regeneration of the brittle star genes in the expanded and contracted gene families annotated with the GO term 'keratan sulfate metabolism' (see Fig. 3b). Representation is as in G. Note that one identified contracted gene family contains no brittle star genes (*ST3GAL1-like*) and is thus absent from the figure. **J.** Genes from expanded and contracted keratan sulfate gene families are associated with specific regeneration clusters (Supplementary Table 7). Representation is as in H.

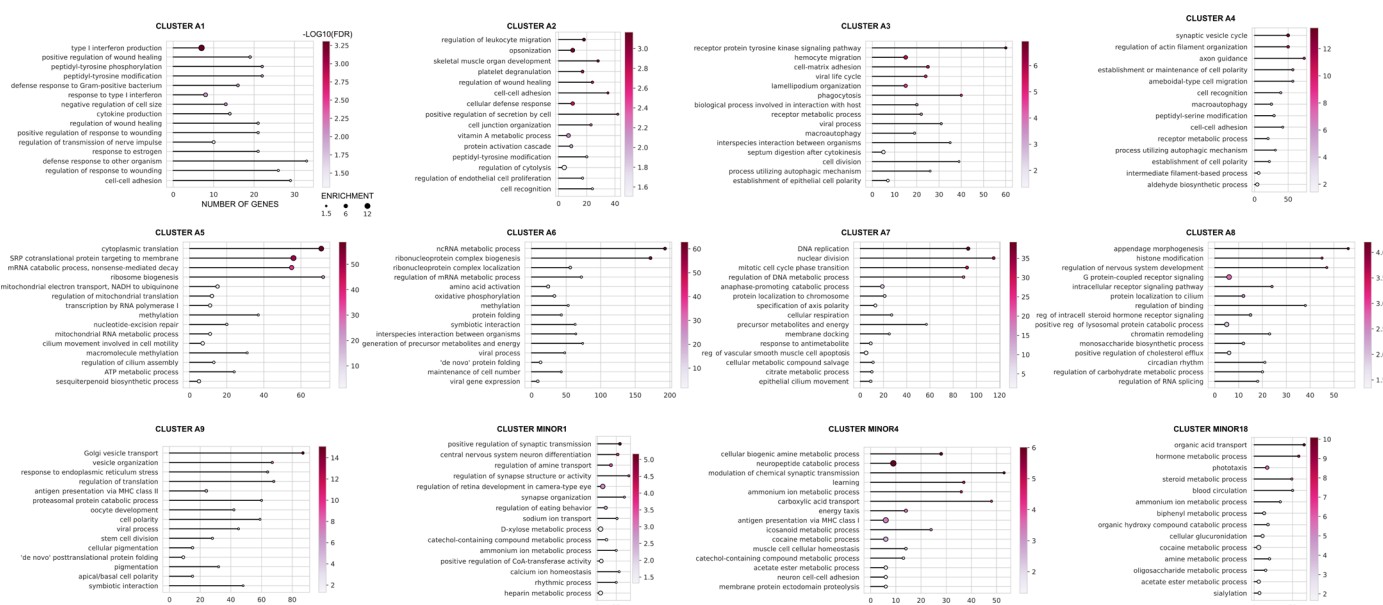

**Extended Data Fig. 7 | Gene ontology enrichment results for brittle star arm regeneration co-expression clusters.** GO enrichment tests were performed on each co-expression cluster and summarised using REVIGO (Methods). The complete list of enriched terms is presented in Supplementary Table 10.

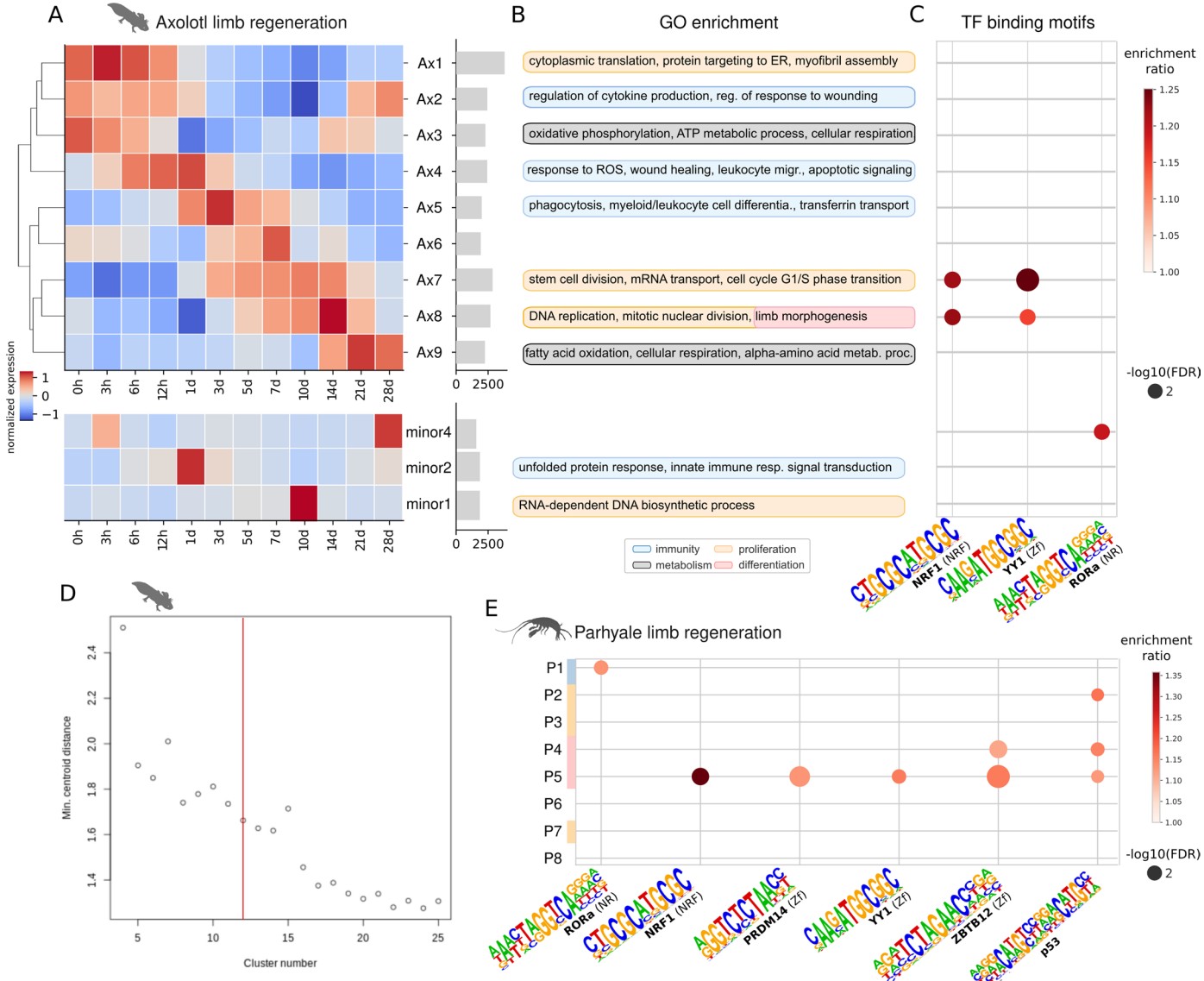

**Extended Data Fig. 8 | Clustering and functional enrichments for the axolotl and Parhyale limb regeneration gene expression time series. A.** Normalised expression profiles (expression of the centroid) for each of the n = 12 axolotl limb regeneration co-expression clusters. Raw expression data were re-processed from Stewart et al.[49] (Methods). Barplots on the right indicate the number of genes assigned to each cluster. Clusters with genes expressed over a single regeneration time point were defined as minor clusters and not presented in the main text as they may be driven by noisy gene expression. **B.** Gene ontology enrichment for each co-expression cluster (Methods, Supplementary Table 6). **C.** TF binding motifs enriched around the TSS of genes from axolotl co-expression clusters (hypergeometric test adjusted p-value < 0.05, Methods). Note that only TFBS motifs enriched in brittle star clusters are represented. **D.** Optimal number of clusters estimated using the centroid distance. We selected n = 12 clusters since further increase of the number of clusters does not result in a significant decrease of the centroid distance until n = 16, which, on the basis of functional enrichment tests, over-clusters the data. **E.** TF binding motifs enriched around the TSS of genes from Parhyale co-expression clusters as in **C**. Parhyale clusters were renamed from Sinigaglia et al.[48] as follows: P1 is R4 in the notation of Sinigaglia et al., P2 is R1, P3 is R8, P4 is R2, P5 is R6, P6 is R3, P7 is R5 and P8 is R7.

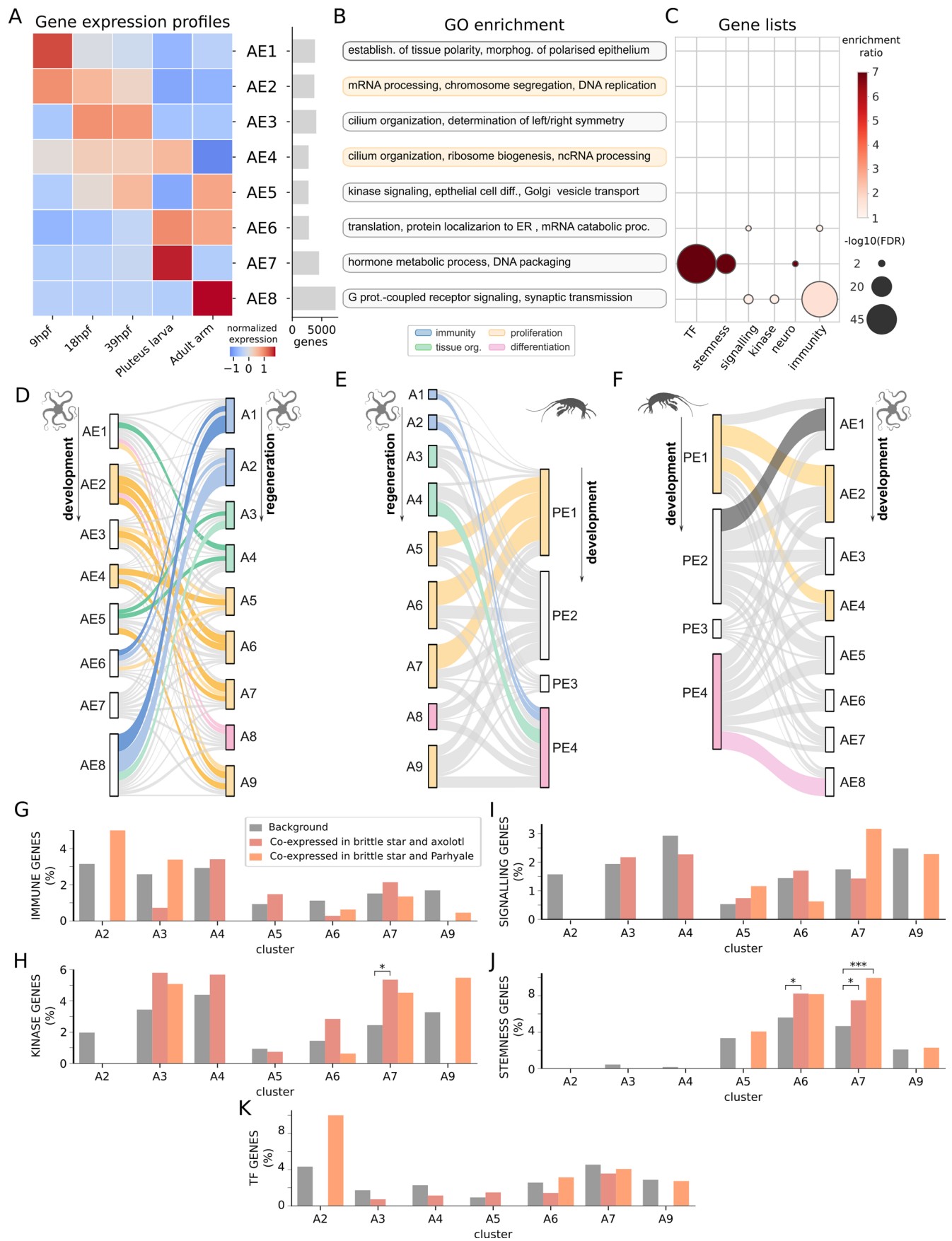

**A** Gene expression profiles

**B** GO enrichment

establish. of tissue polarity, morphog. of polarised epithelium

mRNA processing, chromosome segregation, DNA replication

cilium organization, determination of left/right symmetry

cilium organization, ribosome biogenesis, ncRNA processing

kinase signaling, epthelial cell diff., Golgi vesicle transport

translation, protein localizarion to ER , mRNA catabolic proc.

hormone metabolic process, DNA packaging

G prot.-coupled receptor signaling, synaptic transmission

immunity proliferation
tissue org. differentiation

**C** Gene lists

enrichment ratio

-log10(FDR)

normalized expression

**G** IMMUNE GENES (%)

Background
Co-expressed in brittle star and axolotl
Co-expressed in brittle star and Parhyale

**H** KINASE GENES (%)

**I** SIGNALLING GENES (%)

**J** STEMNESS GENES (%)

**K** TF GENES (%)

**Extended Data Fig. 9 | See next page for caption.**

**Extended Data Fig. 9 | Comparison of co-expression gene clusters during regeneration and development. A**. Clustering of the brittle star development time series. Normalised expression profiles for each of the n = 8 development co-expression clusters. Processing, clustering procedure and representation is as in (Fig. 4, Extended Data Fig. 8). RNA-seq source listed in Supplementary Table 1. **B**. Gene ontology enrichment for each co-expression cluster. **C**. Curated gene lists enrichment for each co-expression cluster (hypergeometric test, Benjamini-Hochberg adjusted p-values < 0.05). **D**. Comparison of co-expressed gene clusters deployed during embryonic development and appendage regeneration in the brittle star. Note that the embryonic development in brittle star does not produce appendages and is thus less informative than Parhyale development data. Clusters are represented by vertical rectangles whose sizes are proportional to the number of homologous genes in the cluster, and coloured according to enriched GO terms. Genes are linked across clusters, with coloured links indicating significant overlaps (hypergeometric test with the Benjamini-Hochberg correction <0.01, darker shades indicate p-values < 10-15). **E**. Comparison of co-expressed gene clusters deployed during appendage regeneration in the brittle star and leg development in Parhyale. Clusters in Parhyale (clusters PE1 to PE4) correspond to the clustering reported in Sinigaglia et al.[48], but clusters were renamed to follow temporal activation (PE1 corresponds to E2, PE2 to E4, PE3 to E1, PE4 to E3). Coloured links indicating significant overlaps (permutation-based over-representation p-values with Benjamini-Hochberg correction <0.05, Methods). **F**. Comparison of co-expressed gene clusters deployed during development in the brittle star and leg development in Parhyale, as in **E. G-K**. Gene list enrichment tests, for genes with a conserved expression profile during appendage regeneration, as in Fig. 5d, but sub-divided by cluster and species comparisons (hypergeometric tests, p-values corrected for multiple testing with the BH procedure, * p-values < 0.05, ** p-values < 0.01, *** p-values < 0.001).

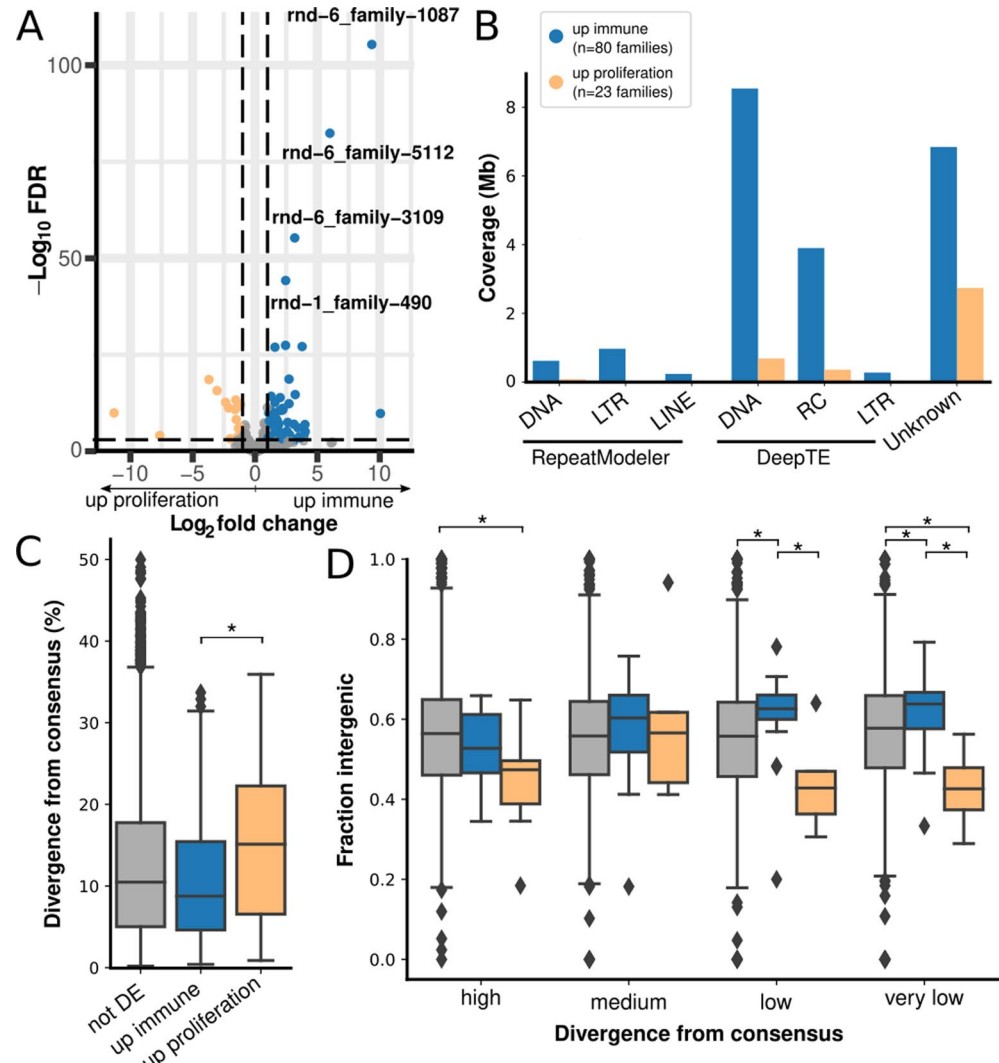

**Extended Data Fig. 10 | Differential transcriptional activity of repetitive elements in the immune and proliferative phases of brittle star arm regeneration. A**. Differentially expressed repetitive elements in early regeneration (immune phase: 48 hpa and 72 hpa samples) versus middle regeneration (proliferation: Stage3, Stage4, Stage5 samples). Coloured dots represent repeat families with significant up-expression in immune (blue) or proliferation phases (orange) (absolute log fold change > 1, FDR < 0.001, two-sided Wald test p-values corrected for multiple testing using the BH procedure, Methods). **B.** Immune up-expressed repeat families (n = 80) have a higher genomic coverage than proliferation up repeat families (n = 23), regardless of repeat class. Coverage is shown subdivided by repeat class, where classification was performed first using the homology-based approach of RepeatModeler, then with DeepTE for repeats that could not be classified by RepeatModeler (Methods). We note that the DeepTE classification has higher false positives than the RepeatModeler classification. **C.** Immune up-expressed repeat families have significantly lower divergence to their consensus (Kimura distance, Methods) than proliferation up-expressed repeat families (Mann–Whitney U test, one-sided p-value corrected for multiple testing with the BH procedure, * p-values < 0.05), indicating they are younger repeats with a higher potential to

still be active mobilisable transposable elements. Distribution details [minima, bottom whisker, q1, median, q3, top whiskers and maxima] are as follows: not DE [0, 0, 5.02, 10.48, 17.75, 36.81, 49.97], up immune [0.4, 0.4, 4.62, 8.76, 15.44, 31.43, 33.7], up proliferation [0.88, 0.88, 6.56, 15.12, 22.26, 35.93, 35.93]. **D**. Immune up-expressed repeat families with low divergence from their consensus have significantly higher fraction of intergenic repeat instances, suggesting up-expression is less likely to be a side-effect of host gene transcription. P-values and boxplot colours are as in **C** (grey = no significant differential expression, blue = up in immune, orange = up in proliferation). Repeat families were subdivided in 4 balanced categories based on their divergence to consensus (Kimura distance, d): d < 5.02 (very low), 5.02 < d < 10.48 (low), 10.48 < d < 17.73 (medium), 17.73 < d (high). Distribution details as in **C** are as follows (boxes from left to right): [0.0, 0.19, 0.46, 0.56, 0.64, 0.91, 1.0], [0.34, 0.34, 0.47, 0.53, 0.61, 0.66, 0.66], [0.18, 0.35, 0.39, 0.47, 0.5, 0.65, 0.65], [0.0, 0.2, 0.46, 0.56, 0.64, 0.9, 1.0], [0.18, 0.41, 0.52, 0.6, 0.66, 0.76, 0.76], [0.41, 0.41, 0.44, 0.57, 0.62, 0.62, 0.94], [0.0, 0.19, 0.46, 0.56, 0.64, 0.9, 1.0], [0.2, 0.57, 0.6, 0.63, 0.66, 0.71, 0.78], [0.31, 0.31, 0.36, 0.43, 0.47, 0.47, 0.64], [0.0, 0.22, 0.48, 0.58, 0.66, 0.91, 1.0], [0.33, 0.47, 0.58, 0.64, 0.67, 0.79, 0.79], [0.29, 0.29, 0.37, 0.43, 0.48, 0.56, 0.56].

# Reporting Summary

## Statistics

For all statistical analyses, confirm that the following items are present in the figure legend, table legend, main text, or Methods section.

| n/a | Confirmed | |
|---|---|---|
| ☐ | ☒ | The exact sample size (*n*) for each experimental group/condition, given as a discrete number and unit of measurement |
| ☐ | ☒ | A statement on whether measurements were taken from distinct samples or whether the same sample was measured repeatedly |
| ☐ | ☒ | The statistical test(s) used AND whether they are one- or two-sided *Only common tests should be described solely by name; describe more complex techniques in the Methods section.* |
| ☒ | ☐ | A description of all covariates tested |
| ☐ | ☒ | A description of any assumptions or corrections, such as tests of normality and adjustment for multiple comparisons |
| ☐ | ☒ | A full description of the statistical parameters including central tendency (e.g. means) or other basic estimates (e.g. regression coefficient) AND variation (e.g. standard deviation) or associated estimates of uncertainty (e.g. confidence intervals) |
| ☒ | ☐ | For null hypothesis testing, the test statistic (e.g. *F*, *t*, *r*) with confidence intervals, effect sizes, degrees of freedom and *P* value noted *Give P values as exact values whenever suitable.* |
| ☐ | ☒ | For Bayesian analysis, information on the choice of priors and Markov chain Monte Carlo settings |
| ☒ | ☐ | For hierarchical and complex designs, identification of the appropriate level for tests and full reporting of outcomes |
| ☒ | ☐ | Estimates of effect sizes (e.g. Cohen's *d*, Pearson's *r*), indicating how they were calculated |

*Our web collection on statistics for biologists contains articles on many of the points above.*

## Software and code

Policy information about availability of computer code

| Data collection | No data collection software |
|---|---|
| Data analysis | flye (v2.9-b1768), Racon (v1.5.0), minimap2 (v2.24-r1122), Merqury (v1.3), Inspector (v1.0.2), purge_dups (v1.2.5), bwa mem (0.7.17-r1198-dirty),  pairtools (v1.0.2) ,  YAHS (v1.1a-r3), 3D-DNA (v180922), RepeatModeler (2.0.2), RepeatMasker (4.1.2-p), DeepTE, MetaEuk (6-a5d39d9), Circos (v0.69.8), CAFE (v5), MAFFT (v7.475), RAxML-NG (v.1.1), PhyloBayes (v4.1b), eggnog-mapper (web version), ClusterProfiler (v4.10.1), kallisto (v0.48), edgeR (v4.0.16), MFuzz (v2.62.0), HOMER (v4.11), DESEq2 (v1.42.1) |

For manuscripts utilizing custom algorithms or software that are central to the research but not yet described in published literature, software must be made available to editors and reviewers. We strongly encourage code deposition in a community repository (e.g. GitHub). See the Nature Portfolio guidelines for submitting code & software for further information.

## Data

Policy information about availability of data

All manuscripts must include a data availability statement. This statement should provide the following information, where applicable:

- Accession codes, unique identifiers, or web links for publicly available datasets
- A description of any restrictions on data availability
- For clinical datasets or third party data, please ensure that the statement adheres to our policy

Genome sequence and RNA-seq data have been deposited in NCBI SRA (Bioproject PRJNA1029566). Supplemental datasets have been deposited in Zenodo (see

Supplemental Material for content details). These include the genome, gene and repeat annotations, processed gene expression tables and source data for the figures.

## Research involving human participants, their data, or biological material

Policy information about studies with human participants or human data. See also policy information about sex, gender (identity/presentation), and sexual orientation and race, ethnicity and racism.

| | |
|---|---|
| Reporting on sex and gender | *Use the terms sex (biological attribute) and gender (shaped by social and cultural circumstances) carefully in order to avoid confusing both terms. Indicate if findings apply to only one sex or gender; describe whether sex and gender were considered in study design; whether sex and/or gender was determined based on self-reporting or assigned and methods used. Provide in the source data disaggregated sex and gender data, where this information has been collected, and if consent has been obtained for sharing of individual-level data; provide overall numbers in this Reporting Summary. Please state if this information has not been collected. Report sex- and gender-based analyses where performed, justify reasons for lack of sex- and gender-based analysis.* |
| Reporting on race, ethnicity, or other socially relevant groupings | *Please specify the socially constructed or socially relevant categorization variable(s) used in your manuscript and explain why they were used. Please note that such variables should not be used as proxies for other socially constructed/relevant variables (for example, race or ethnicity should not be used as a proxy for socioeconomic status). Provide clear definitions of the relevant terms used, how they were provided (by the participants/respondents, the researchers, or third parties), and the method(s) used to classify people into the different categories (e.g. self-report, census or administrative data, social media data, etc.) Please provide details about how you controlled for confounding variables in your analyses.* |
| Population characteristics | *Describe the covariate-relevant population characteristics of the human research participants (e.g. age, genotypic information, past and current diagnosis and treatment categories). If you filled out the behavioural & social sciences study design questions and have nothing to add here, write "See above."* |
| Recruitment | *Describe how participants were recruited. Outline any potential self-selection bias or other biases that may be present and how these are likely to impact results.* |
| Ethics oversight | *Identify the organization(s) that approved the study protocol.* |

Note that full information on the approval of the study protocol must also be provided in the manuscript.

# Field-specific reporting

Please select the one below that is the best fit for your research. If you are not sure, read the appropriate sections before making your selection.

☒ Life sciences    ☐ Behavioural & social sciences    ☐ Ecological, evolutionary & environmental sciences

For a reference copy of the document with all sections, see nature.com/documents/nr-reporting-summary-flat.pdf

# Life sciences study design

All studies must disclose on these points even when the disclosure is negative.

| | |
|---|---|
| Sample size | No statistical methods were used to determine sample size. For explant experiments, 20 samples, each derived from 150-200 explants were used, and sample size was based on experimental capabilities and common practice in transcriptomic analyses. |
| Data exclusions | No data points were excluded |
| Replication | 3-4 replicates were generated for each condition |
| Randomization | No randomization was used |
| Blinding | Blinding was not relevant to our study. |

# Reporting for specific materials, systems and methods

We require information from authors about some types of materials, experimental systems and methods used in many studies. Here, indicate whether each material, system or method listed is relevant to your study. If you are not sure if a list item applies to your research, read the appropriate section before selecting a response.

## Materials & experimental systems

| n/a | Involved in the study |
|---|---|
| ☒ | ☐ Antibodies |
| ☒ | ☐ Eukaryotic cell lines |
| ☒ | ☐ Palaeontology and archaeology |
| ☐ | ☒ Animals and other organisms |
| ☒ | ☐ Clinical data |
| ☒ | ☐ Dual use research of concern |
| ☒ | ☐ Plants |

## Methods

| n/a | Involved in the study |
|---|---|
| ☒ | ☐ ChIP-seq |
| ☒ | ☐ Flow cytometry |
| ☒ | ☐ MRI-based neuroimaging |

# Animals and other research organisms

Policy information about <u>studies involving animals</u>; <u>ARRIVE guidelines</u> recommended for reporting animal research, and <u>Sex and Gender in Research</u>

| | |
|---|---|
| Laboratory animals | No laboratory animals were used. |
| Wild animals | All experiments were performed on adult brittle stars collected from the Gulmarsford. |
| Reporting on sex | The sex of the brittle stars used for experiments was not determined. The genome of a male was sequenced. |
| Field-collected samples | For experiments, brittle stars were maintained at the Kristineberg laboratory in sea water aquarium and amputations were performed as reported in the paper. |
| Ethics oversight | Brittle stars are invertebrates and not under any animal experimental regulation |

Note that full information on the approval of the study protocol must also be provided in the manuscript.

# Plants

| | |
|---|---|
| Seed stocks | *Report on the source of all seed stocks or other plant material used. If applicable, state the seed stock centre and catalogue number. If plant specimens were collected from the field, describe the collection location, date and sampling procedures.* |
| Novel plant genotypes | *Describe the methods by which all novel plant genotypes were produced. This includes those generated by transgenic approaches, gene editing, chemical/radiation-based mutagenesis and hybridization. For transgenic lines, describe the transformation method, the number of independent lines analyzed and the generation upon which experiments were performed. For gene-edited lines, describe the editor used, the endogenous sequence targeted for editing, the targeting guide RNA sequence (if applicable) and how the editor was applied.* |
| Authentication | *Describe any authentication procedures for each seed stock used or novel genotype generated. Describe any experiments used to assess the effect of a mutation and, where applicable, how potential secondary effects (e.g. second site T-DNA insertions, mosiacism, off-target gene editing) were examined.* |

