## [Peer review file · Nature Ecology & Evolution]

Peer Review Information

Journal: Nature Ecology & Evolution

Manuscript Title: The brittle star genome illuminates the genetic basis of animal appendage regeneration

Corresponding author name(s): Elise Parey, Paola Oliveri, Ferdinand Marlétaz

Editorial Notes:

Reviewer Comments & Decisions:

Decision Letter, initial version:

19th December 2023

Dear Dr Marlétaz,

Your Article, "The brittle star genome illuminates the genetic basis of animal appendage regeneration" has now been seen by three reviewers. You will see from their comments copied below that while they find your work of considerable potential interest, they have raised quite substantial concerns that must be addressed. In light of these comments, we cannot accept the manuscript for publication, but would be very interested in considering a revised version that addresses these serious concerns.

We hope you will find the reviewers' comments useful as you decide how to proceed. If you wish to submit a substantially revised manuscript, please bear in mind that we will be reluctant to approach the reviewers again in the absence of major revisions.

In particular, while we don't think that additional data such as single cell RNAseq or functional validation of candidates as mentioned by the first reviewer are necessary for publication, we are particularly concerned with their criticisms regarding genome quality and would need to see a sign off on this important issue.

If you choose to revise your manuscript taking into account all reviewer and editor comments, please highlight all changes in the manuscript text file in Microsoft Word format.

* Include a "Response to reviewers" document detailing, point-by-point, how you addressed each referee comment. If no action was taken to address a point, you must provide a compelling argument. This response will be sent back to the referees along with the revised manuscript.

* If you have not done so already we suggest that you begin to revise your manuscript so that it conforms to our Article format instructions at <http://www.nature.com/natecolevol/info/final->

2submission. Refer also to any guidelines provided in this letter.

[REDACTED]

If you wish to submit a suitably revised manuscript we would hope to receive it within 6 months. If you cannot send it within this time, please let us know. We will be happy to consider your revision so long as nothing similar has been accepted for publication at Nature Ecology & Evolution or published elsewhere.

Nature Ecology & Evolution is committed to improving transparency in authorship. As part of our efforts in this direction, we are now requesting that all authors identified as 'corresponding author' on published papers create and link their Open Researcher and Contributor Identifier (ORCID) with their account on the Manuscript Tracking System (MTS), prior to acceptance. This applies to primary research papers only. ORCID helps the scientific community achieve unambiguous attribution of all scholarly contributions. You can create and link your ORCID from the home page of the MTS by clicking on 'Modify my Springer Nature account'. For more information please visit please visit www.springernature.com/orcid.

Thank you for the opportunity to review your work.

[REDACTED]

Reviewer expertise:

Reviewer #1: genomics of marine invertebrates

Reviewer #2: evolution of regeneration, comparative analysis of genomics data

Reviewer #3: cnidarian regeneration, comparative analysis of genomics data

Reviewers' comments:

2Reviewer #1 (Remarks to the Author):

This paper makes the contribution to the field by providing plenty genomic and transcriptomic data on brittle stars. The genome assembly of *A. filiformis* complements genomic resources available for echinoderms. The authors have conducted comparative analysis of genome organization in brittle stars, sea stars, sea urchins, and sea cucumbers, offering insights into the evolution of gene clusters. Additionally, the study explores gene expression during arm regeneration in brittle stars, suggesting phases of gene expression linked to wound healing, proliferation, and differentiation. However, the paper falls short in several critical aspects. Much of the methodology and analysis are underdeveloped or insufficient. The claim of discovering most rearranged echinoderm genome and conserved gene expression during regeneration among brittle stars, *Parhyale*, and axolotl appears not solidly supported or likely overstated. Moreover, this study seems not significantly advance our current understanding of animal regeneration, as the field has witnessed numerous comprehensive studies on similar themes, rendering the current contributions less impactful.

Major issues:

1. The genome quality is the core of the whole paper, yet there're several spots make me concern about it. Firstly, the use of sperm cells for DNA sequencing and genome assembly is questionable, as recombination in these cells can lead to increased heterozygosity, rendering each sperm cell genetically unique. This variability could challenge the genome assembler to accurately assemble regions with high genetic variation, casting doubt on the paper's conclusion that "The brittle star exhibits the highest inter-chromosomal rearrangement rate amongst sequenced echinoderms". Secondly, the authors claimed they obtained a diploid assembly in Line 786, followed by the significant reduction of the assembled genome after haplotype removal (2.86 Gb to 1.57 Gb). It's unclear how sperm cells could produce such a diploid assembly with a high level of haplotypic duplication. Thirdly, to assure "*A. filiformis* is the most rearranged echinoderm genome among those sequenced to date" (Line 200), the authors should use different assemblers and sampling various individuals (with adult tissues preferred) to replicate these results. There is an insufficiency of genome assembly quality assessment (currently just BUSCO score) in this study, especially when the paper reports 20 chromosomes for *A. filiformis*, whereas other brittle stars are known to have 21 chromosomes (Lines 130-132). The claim of high genome rearrangement could also be caused by the problem of over-scaffolding. Therefore, these issues necessitate a thorough evaluation of the assembly process and its outcomes.
2. The manuscript would benefit significantly from a deeper exploration into the biological implications of Hox cluster rearrangements. The authors should investigate the spatio-temporal expression of Hox genes specifically in the context of brittle star embryonic development. Additionally, the adoption of more advanced techniques, such as single-cell RNA sequencing and/or spatiotemporal transcriptomics, is also recommended, as such analyses would enable a more comprehensive understanding of the developmental regulatory roles of these rearrangements.
3. The authors assert that "Tandem duplications of key genes contribute to brittle star larval skeleton and bioluminescence"(Line 291), yet they fail to provide functional analysis or evidence to support this claim. There is a possibility that these genes have acquired new functions in brittle stars. It remains to

3approve whether all or only specific copies of these paralogous genes are most crucial in these processes.

4. The discovery of co-expression clusters corresponding to three phases of arm regeneration is somewhat anticipated, which does not deviate from existing understanding and thus lacks the significance of novelty. The authors acknowledge that “These are consistent with morphological timelines of regeneration in the brittle star and other animal systems” (Line 381). A deeper understanding of the genetic mechanisms of regeneration would be beneficial, necessitating more histological and functional studies to validate the findings. Among the several thousand co-expressed genes during regeneration, which are key in each temporal cluster? Key genes could be identified by their hub status in co-expression networks and by their impact on the regeneration process when knocked down/out. Further inquiries should explore the signaling cascades of these key genes in brittle stars and their conservation in Parhyale and axolotl. It would be insightful to determine which cells express these genes during regeneration across these species, and whether they share cellular and spatial similarities.

5. While the study claimed the identification of conserved gene modules utilized by three species during regeneration, the evolutionary history behind this observation—whether it is ancestral origin or a product of convergent evolution—remain ambiguous. It is noteworthy that traits developed through convergent evolution often recurrently utilize similar gene sets. Notably, the number of genes shared among the three species is considerably fewer compared to those shared between any two species (154 genes versus 926 and 913 genes, respectively). This discrepancy suggests a potential under-representation of conserved genes across all three species. This study suffers from providing the strong evidence for distinguishing different hypotheses (ancestral origin or convergent evolution), partly due to the limited number of species included.

6. The explant experiments conducted in the study rely predominantly on bulk RNA-seq, which lacks the resolution necessary for drawing definitive conclusions. Furthermore, it is a pity that the manuscript did not include carefully designed functional validation for the five genes identified as having opposite expression patterns during the wound healing and regeneration processes. Such functional assays may provide novel mechanistic insights into the difference between wound healing and regeneration.

Minor issues:

1. The authors note that the *A. filiformis* genome is the most rearranged among echinoderms sequenced to date. It's important to explore the biological significance of identifying the 'most rearranged genome' and the evolutionary reasons for this. What are the DNA features around the rearrangement boundaries? Are they repeat-rich? What are the estimated ages of those boundary associated repeats?

2. The manuscript would benefit from further clarity regarding the brittle star genes that have expanded in relation to regeneration. Specifically, it is important to determine what proportion of these genes show significant regulation during the arm regeneration process. Additionally, what is the proportion of such genes in each co-expression cluster?

3. In Figure 3, the Ficolin-like and TRIM-like gene families show extensive expansion across echinoderms, whereas other regeneration-related gene families primarily expand in *A. filiformis* and *S. purpuratus*. Does this suggest a variation in regeneration capabilities among these species?
4. It is intriguing that in Figure 4, the non-regenerating control group exhibits the highest expression levels of wound-response genes (clusters A1 and A2), particularly since clusters A1 and A2 include many 'regeneration-related' expanded genes shown in Figure S6. This observation prompts further investigation into whether these genes are involved in other non-regenerative processes in the control group.
5. The author should examine the expression patterns of genes involved in keratan sulfate metabolism (Line 350) during regeneration, to determine if the variations in expanded and contracted genes indicate the 'specialization of glycosaminoglycan metabolism'.
6. It is confusing that in their previous papers (Czarkwiani et al. Development 2021, Piovani et al. BMC Biol 2021), the authors suggest FGF signaling and *alx1* genes play important roles during the regeneration process of brittle stars, which was neither discovered nor mentioned in this paper.
7. In Line 267, the authors reference Annunziata et al. (2014) to suggest that *Hox7* and *Hox11/13b* may play important roles in embryogenesis. However, it didn't mention *Hox7* at all in this cited paper. The correct citation should be Li et al. (Cell Discovery 2018).

Reviewer #2 (Remarks to the Author):

In this manuscript, the authors present a chromosome-level genome assembly of the brittle star *Amphiura filiformis*. These data enable the authors to compare various brittle star genomic features with other echinoderm species and make inferences about echinoderm evolution, as well as to combine these genomic data with transcriptomic datasets to compare the brittle star's appendage regeneration with other regenerative taxa. First, by using synteny analyses to compare the *A. filiformis* genome with other echinoderm members (sea urchin, sea star, and sea cucumber species), they find that the brittle star genome is the most rearranged at a chromosome level relative to the predicted ancestral chromosomal arrangement. They next demonstrate the power of these data to address comparative evolutionary questions by assessing finer-scale genomic features using this new assembly. First, they use *Hox* genes as a case study to investigate smaller-scale rearrangements in echinoderms, discovering that the brittle star *Hox* clusters are highly rearranged relative to other echinoderm species, and that these rearrangements seem independent of *Hox* rearrangements in other echinoderm members. They also identify tandem duplications in genes associated with key features of brittle star biology - larval skeletal and bioluminescence-related genes - and by identifying some of these genetic components suggest possible mechanisms for the evolution of these traits. Finally, they ask about gene family expansion and contraction in brittle stars relative to other echinoderms and find that many of these members are associated with regeneration-related GO terms, and in fact that many of these genes are expressed during brittle star regeneration based on

5previous transcriptomic datasets. They propose some of these identified changes could help explain the brittle star's high regenerative capacity.

Next the authors transition to investigating regeneration more deeply using both their genomic data as well as pre-existing and newly generated transcriptome data. They use soft clustering to identify sets of co-expressed genes, which they find correspond to known morphological features in brittle star regeneration that are also more broadly observed in additional regenerative taxa: wound-healing, proliferation, and differentiation. They then use transcriptomic time courses in appendage regeneration from axolotl and parhyale to find sets of co-expressed genes in those organisms, and then compare these clusters between species. They find that clusters enriched in proliferation genes are most similar, and suggest this may point to co-option of an ancient metazoan proliferation program across regenerative taxa. Additionally, they find that the general order of these types of processes in regeneration is similar across species. There is little correlation in gene membership between clusters associated with differentiation, as might be expected if these animals are all regenerating different tissues. There are few features at the gene level that are conserved across all three species, including enrichment of motifs near TSS of coexpressed brittle star genes; the few that are shared include two transcription factors that are related to proliferation in other systems and thus they suggest might be related to this potential shared proliferation program. Finally, the authors investigate differences in regenerative and non-regenerative wound-response in brittle stars by doing explant experiments, in which one portion of the limb heals the wound, whereas another portion undergoes regeneration. By comparing RNA expression between these sites, and across two timepoints, they identify an enrichment of proliferation-related genes in the regenerating vs non-regenerating datasets, many wound-healing genes that are shared, but also general wound response/wound healing genes that are specific to the regenerative sites.

This work demonstrates the power of the brittle star as a comparative model in multiple aspects of evolutionary/developmental biology. The genomic and transcriptomic datasets generate many hypotheses related to echinoderm body plan evolution and shared features of regeneration to be investigated further in the future. A clearer narrative connecting the various aspects of the paper would help readers unfamiliar with brittle stars navigate their many findings. The authors also present many possible mediators of regeneration that were identified in this study, but there are some important caveats to these conclusions that should be addressed, and it should be made clearer that these are important avenues for further investigation rather than definitive answers to what drives regeneration in the brittle star and other species. In particular, the authors are overinterpreting the proliferation gene data - all animals will need proliferating cells to be able to regenerate new tissue, and we already know that metazoans utilize very conserved machinery for controlling cell proliferation. Therefore, finding shared gene modules associated with proliferation would almost be a null hypothesis of sorts for this type of comparison. The analyses in this paper are still meaningful, but the authors should rewrite their interpretations of these results.

Major comments:

Narrative cohesiveness:

The paper seems divided into two major themes: using the genome to make comparisons between echinoderms, and then relying more heavily on transcriptomic data to make comparisons related to appendage regeneration between more distantly related taxa (brittle stars, parhyale, and axolotl). The

6rationale for linking these two themes within this same narrative, and how each section motivates the next, could be clearer. Similarly, how do the findings from the comparative regeneration work motivate the explant experiments comparing regenerative and non-regenerative wound healing in the brittle star? As written, these sections seem somewhat disjointed.

The title and framing of this study also centers the work around the new genome, but the regenerative portion of the manuscript and the conclusions that are drawn within it rely much more heavily upon the transcriptomic datasets. Making it more explicit how having the new genome makes it possible to complete and/or augments the analyses in this paper beyond what is possible only using transcriptomic data would help justify the title and framing of the work (for example, without the genome it was possible to look at gene expression, but motif enrichment analyses were not possible, and this was important in generating hypotheses regarding gene regulatory modules; or it was difficult to say without the genome if gene family expansions were actual gene duplications or just splice variants).

Clarification of previous knowledge:

In the regeneration portion of the manuscript there are many genes/pathways that the authors identify in their analyses and they note are previously known to have roles in regeneration or regeneration-related phenomena. It is often unclear whether this previous knowledge is related to a broad role across many taxa, was known to have that role in some other echinoderm (or echinoderms generally), or whether this was specifically something that had been previously identified in the brittle star – citations are provided, but it would be helpful for the statements to be more clear about this. Similarly, it isn't always clear when a feature of the transcriptomic data is a new finding or is corroborating past knowledge (is this something new and exciting? Or is it something that gives us confidence in the data and existing new avenues to pursue further?). Clarifying these points will help guide readers less familiar with brittle star/echinoderm literature, and also help place the comparative work with other species in the broader context of previously known shared features of regeneration. Related to this, expanding the schematic in Figure 4A depicting the transcriptomic sampling to include additional information corresponding to known features of regeneration relevant to the morphological stages that were previously identified would help readers unfamiliar with brittle star regeneration make more sense of the analyses that follow. For example, the schematic could include summarized results from EdU experiments in other studies, pointing out when the blastema-like outgrowth appears, and when differentiated structures begin appearing. Some of this information is in the text (ie, stage 5 is noted to be when peak proliferation occurs), and having it summarized in one place would help guide the reader through some of these findings.

Proliferation as the most similar feature across these regenerative taxa:

In comparing the transcriptomic datasets across three species, the authors identify proliferation as the most conserved feature of regeneration across taxa at a transcriptomic level, and propose that a common theme of regeneration might be an ancient metazoan proliferation program. There are a couple of discussion points we think should be addressed here. First, if proliferation is an important component of regeneration in all of these taxa, is it surprising that the genes involved in regenerative proliferation are similar between species, or is it simply that proliferative machinery in general is highly conserved across metazoa? Are there signatures of these regenerative proliferation-related co-expression modules that are distinct from proliferation in these organisms in other contexts?

The second point relates to the timepoints that are included in this study, and how these might allow

7for comparison of the wound-response/healing component of the regenerative program. Based on the work presented here and in Czarkwiani et al, 2016, it seems like the earliest time point sampled in this study (48 hpa) is nearing the end of the wound-healing phase and entering the proliferation phase. From other regenerative systems, it is known that many of the wound-induced signals that are required to launch regenerative responses occur very quickly following injury, often before any morphological changes are discernable in the organism. Has any attempt been made to look at, for example, a 6 or 12h time point? We do not necessarily suggest adding these times into the analysis if they do not exist, but rather addressing the limitations of the existing data to identify overlap in the earliest wound-induced networks of regeneration. In line with this, the expression profiles of genes in clusters A1 and A2, which are proposed to correspond to the wound healing process, seem to also have high expression in the control tissue - is this to be expected of genes involved in the wound response?

Because comparisons more broadly to regeneration are invoked, it might also be interesting to place these findings in the context of common themes in regeneration. What do these findings suggest about the evolution of regeneration, or where we should be looking if we want to better understand common regenerative themes? i.e., do these results point to each species independently evolving a wound response that can feed into a common proliferative process, which then again transitions to species-specific differentiation depending on what exactly the animal has to build? How would we expect this to relate to other types of regeneration, like whole body regeneration or different types of structural regeneration? These also often follow the general pattern of wound response, some degree of proliferation, and differentiation. If the argument is that appendage regeneration is not homologous between these species, yet we see the common proliferative response, then would we also expect to see this in other regenerative modes (would we expect this same proliferation program to be involved in the proliferation phase of whole body regenerators)?

Motif Analyses:

These binding motif enrichment experiments identify potential regulators of some of the regenerative processes described in this work. However, a total of 6kb surrounding a TSS is likely to harbor many potential binding sites, especially if the regions have not been narrowed down to those that are accessible at a particular time (and thus more likely to actually be involved in transcription factor binding/downstream gene expression at that time). Is there any evidence that transcription factors that might bind to the identified motifs are expressed at times consistent with them regulating the sets of coexpressed genes that showed enrichment of those motifs? For example, the YY1 motif is enriched at loci corresponding to transcripts in the A5 and A6 clusters; is there a gene corresponding to the YY1 transcription factor expressed at or before the onset of those genes at Stage 3? This could help strengthen the hypothesis that these factors are involved in the regenerative processes proposed.

Minor comments:

Lines 135-138: Why were these specific gene groups chosen for inclusion into manually curated lists? Were they previously known from brittle star biology? Important generally in body plan evolution and/or regeneration? Key pathways/processes that similar genomic studies tend to assess?

Lines 244-265 - In discussion about Hox colinearity and spatio-temporal expression, is there any knowledge about spatial expression in brittle stars, or more broadly other echinoderms?

Lines 293-322 ("Tandem duplications...") - are there other examples of tandem duplications found in the genome, or were these two groups used as a candidate case study based on previous echinoderm

knowledge?

Line 375 - It could help set up the findings in this section to state at the beginning that the stages were chosen to correspond to the known regenerative phases from previous regeneration studies; then the clusters confirmed this knowledge and allowed for further investigation (this also helps establish for readers less familiar with brittle star regeneration what was previously known).

Line 416 - It seems surprising that axial specification would be occurring so late in regeneration, is this consistent with previous reports in brittle stars?

Line 469 - Being clearer in the main text about the criteria for considering a cluster to be co-expressed would help to understand the analyses in this section - something like "clusters were considered co-expressed between two species if there were more shared genes than expected by random chance," then readers who want more details can read about those in the methods.

Line 671 - In the discussion, expanding the finding that proliferation seems conserved to talking about a common proliferative cell type seems like a stretch. This work doesn't address the cellular context of the proliferation program in these organisms or look more deeply at what it would mean for them to be homologous cell types.

Reviewer #3 (Remarks to the Author):

This manuscript focuses on the genomics and transcriptomics of the brittle star *Amphiura filiformis* with a focus on adult arm regeneration. This represents the first sequenced brittle star genome. It is a comprehensive study that is clear, well-written, and provides a broad evolutionary context by putting the findings in the context of other echinoderms as well as other invertebrates and vertebrates. For the appendage regeneration expression section, a major focus of the paper, they first characterize gene expression during three phases of arm regeneration and then compare co-expression clusters with both the crustacean *Parhyale* and with axolotl providing useful context on what aspects of the regeneration program are conserved across these three animals. They finish with explant RNAseq experiments to compare gene expression during wound closure and regeneration in their study organism.

This study will provide important resources for the community to include brittle stars in comparative analyses. For a genome manuscript, it was quite a pleasure to read, and highlights key features of the genome. The figures complement the text nicely and are clearly presented.

Below, I highlight a few areas that could use some attention and clarification in the manuscript:

1. In the section "Expanded gene families in echinoderms are enriched in regeneration-related Processes" the authors state that "recurrent duplications of "regeneration-related" genes may underlie the remarkable regenerative capacity of many echinoderm species. Notably, in *A. filiformis*, members of these expanded gene families (Figure 3C) are expressed during arm regeneration (Figure S6)." I would like the authors to clarify a few points in this section:

Figure 3A shows that there are 266 gene families which have expanded in *A. filiformis*. It would be

9helpful to know how many total "regeneration-related" genes/genes annotated with the GO term "regeneration" there are in the *A. filiformis* genome overall. Then, by comparison, how many of these "regeneration-related" genes are found in expanded gene families - only 7, correct? I appreciate Figure 3B but would like to know how many genes from each GO term shown are found in the genome versus how many are in expanded or contracted families. This could be done in a supplemental format or added to the figure.

From Figure 3C it looks like there are 7 gene families listed that are "regeneration-related" and expanded in *A. filiformis*, which differs from other echinoderms and which is quite interesting. It is not clear if the four gene families associated with coagulation and/or clotting in vertebrates are included under the GO term "regeneration" or if these were found independently?

Same question about the gene family associated with keratan sulfate metabolism. Is this gene family annotated with the GO term "regeneration" or was it found independently from the previous 2017 study on this topic?

For those who are not familiar with keratan sulfate metabolism, it might help to define it and explain that keratan sulfate is a glycosaminoglycan in the text. It might also help to list the gene family name (UDP-glucuronosyltransferase) in the text when talking about the specific gene family that has expanded that is related to keratan sulfate/glycosaminoglycan metabolism. Right now, the reader has to deduce that this is the specific gene that is related to this section of the text by looking at Figure 3C.

2. For the section "Gene expression during brittle star arm regeneration recapitulates major regeneration Phases" both transcription factors and evidence for "binding motifs associated with several TF" enriched around the TSS of genes expressed during regeneration are discussed. I see in the Methods how the transcription factor binding motif enrichments were identified with HOMER, but can you expand on how you identified transcription factors? In the Methods section under "Gene lists curation" I found this sentence "We generated a list of TFs based on the presence of DNA-binding PFAM domains." Can this be described in more detail? How exactly did you run the PFAM search? Did you use hmmscan or InterProScan or another method? Were default parameters and e-value cutoffs used? How did you curate the results for DNA-binding domains?

3. Is there a difference between "Gene lists enrichment tests" and "Gene ontology enrichment tests"? Were both performed with "the enricher function from the ClusterProfiler R package" as described in the Methods? In the manuscript this is unclear. Please clarify.

4. In the Explant experiments section, two of the five genes with the "drastically opposite expression patterns in the wound healing and regenerating segments" are discussed in the context of vertebrates/mice but it would be nice if the other three genes could be more clearly discussed regarding if their functions are known from vertebrates or not to support the final sentence of the section. Or perhaps you could add a bit to the Discussion on this part of the paper.

5. In the Methods for the "Arm regeneration RNA-seq in brittle star severed arm experiments (explant)" the sampling is described well except for explaining how the "43 groups of 150-200

10explants" relates to the final samples. How do these groups relate to the sampling done at two time points (3 days and 5 days) and three sections: proximal, medial and distal? In the end, I would like to know how many replicates were there for each time point and each section? So how many samples total were sequenced? I think this could be more clearly described and would give the reader more confidence to evaluate whether the experiment was designed well with enough replication.

Author Rebuttal to Initial commentsReviewer #1 (Remarks to the Author):

This paper makes the contribution to the field by providing plenty genomic and transcriptomic data on brittle stars. The genome assembly of *A. filiformis* complements genomic resources available for echinoderms. The authors have conducted comparative analysis of genome organization in brittle stars, sea stars, sea urchins, and sea cucumbers, offering insights into the evolution of gene clusters. Additionally, the study explores gene expression during arm regeneration in brittle stars, suggesting phases of gene expression linked to wound healing, proliferation, and differentiation. However, the paper falls short in several critical aspects. Much of the methodology and analysis are underdeveloped or insufficient. The claim of discovering most rearranged echinoderm genome and conserved gene expression during regeneration among brittle stars, Parhyale, and axolotl appears not solidly supported or likely overstated. Moreover, this study seems not significantly advance our current understanding of animal regeneration, as the field has witnessed numerous comprehensive studies on similar themes, rendering the current contributions less impactful.

We thank the reviewer for their honest feedback and comments. We respectfully disagree over their assessment of the methodology and impact of the presented work. Notably, the chromosome-scale genome assembly that we generated is based on state-of-the-art technology and methodology, and takes advantage of both high-coverage long-read nanopore sequencing and chromatin contacts HiC-seq data.

The synteny analysis builds upon a previously published ancestral chromosome reconstruction for the bilaterian ancestor (Simakov et al., 2022), and uses similar methods to extend synteny comparisons to all chromosome-scale genomes available for echinoderms.

With respect to regeneration, we contribute significant novel data (a total of 28 bulk RNA-seq samples) in an undersampled, yet highly regenerative group (Bideau et al., 2021). We moreover note that temporal sampling of gene expression dynamics during regeneration remains limited even in more studied regeneration systems, and that comparable datasets for appendage regeneration only exist for Parhyale and axolotl. Here, we conduct the first comparative analysis of temporal gene expression dynamics across vertebrate and invertebrate species, using standard comparative methodologies from the evo-devo field and reveal that proliferation is the most conserved phase of regeneration. No previous study has been as comprehensive before or has reported similar findings, we are thus confused about the reviewer's claim of compromised novelty.

Together, the presented work ranges from detailed genome evolution analyses to regeneration studies in whole animals and the development of explant cultures, which represent an important contribution to the field of echinoderm genomics. To substantiate the validity of our work, we provide a detailed point-by-point answer below, integrating additional analyses as suggested by the reviewer.

Major issues:

1. The genome quality is the core of the whole paper, yet there're several spots make me concern about it. Firstly, the use of sperm cells for DNA sequencing and genome assembly is questionable, as recombination in these cells can lead to increased heterozygosity, rendering each sperm cell genetically unique. This variability could challenge the genome

12

assembler to accurately assemble regions with high genetic variation, casting doubt on the paper's conclusion that "The brittle star exhibits the highest inter-chromosomal rearrangement rate amongst sequenced echinoderms". Secondly, the authors claimed they obtained a diploid assembly in Line 786, followed by the significant reduction of the assembled genome after haplotype removal (2.86 Gb to 1.57 Gb). It's unclear how sperm cells could produce such a diploid assembly with a high level of haplotypic duplication. Thirdly, to assure "A. filiformis is the most rearranged echinoderm genome among those sequenced to date" (Line 200), the authors should use different assemblers and sampling various individuals (with adult tissues preferred) to replicate these results. There is an insufficiency of genome assembly quality assessment (currently just BUSCO score) in this study, especially when the paper reports 20 chromosomes for A. filiformis, whereas other brittle stars are known to have 21 chromosomes (Lines 130-132). The claim of high genome rearrangement could also be caused by the problem of over-scaffolding. Therefore, these issues necessitate a thorough evaluation of the assembly process and its outcomes.

To address the reviewer's concerns regarding the quality of the genome assembly, we performed additional control analyses and included extra statistics regarding the assembly quality.

First, we produced k-mer spectrum plots (Rhie et al. Genome Biol. 2020, <https://doi.org/10.1186/s13059-020-02134-9>) that demonstrates correct haplotype removal (Figure S1A,B), with the effective collapse of homozygous and heterozygous regions from the diploid assembly that are represented as single copies (haploid) in the resulting primary assembly:

Figure S1. A. K-mer spectrum of the diploid assembly, i.e. before haplotype removal. Read-only k-mers (black curve) correspond to sequencing errors, and are not represented in the assembly. The 1-copy k-mer peak at 15X coverage (1n, red) corresponds to reads from heterozygous regions, whereas the 2-copy k-mer peak at 30X coverage (2n, blue) corresponds to homozygous regions. **B.** K-mer spectrum of the primary assembly, i.e. after haplotype removal. Following the collapse of haplotypes, half of the k-mers of the heterozygous peaks are accordingly not represented in the assembly anymore and homozygous regions are present as single copies only.

Second, we used an alternative assembly methodology (3D-DNA, Dudchenko et al., Science 2017, <https://doi.org/10.1126/science.aal3327>) to scaffold the genome and compare it to our assembly for validation. This comparison reveals a strong collinearity between the two

13

assemblies and a perfect 1-to-1 correspondence between the chromosomes assembled by the two methods (see the dotplot below for reference). We integrate this control in the Methods (see Genome assembly section): “The 20 chromosomes are strongly supported by the HiC-map (Figure S1) and recovered with a perfect 1-to-1 match using an alternative assembly methodology (3D-DNA, (Dudchenko et al. 2017)), demonstrating that the assembly is suitable to study chromosome evolution across echinoderms.”

Dotplot comparison of *Amphiura filiformis* genome assemblies generated using two different methodologies (YAHS and 3D-DNA). Chromosomes are highly contiguous, and perfectly 1-to-1.

Moreover, we outline that we did not only report high BUSCO scores (completeness of the assembly and annotation) as quality assessment metrics, we had also included (i) the Hi-C contact map (Figure S1C, supporting the 20 reported chromosomes) and (ii) contiguity metrics (N50 before and after scaffolding, fraction of the genome in chromosomes). Together, these are the statistics routinely reported by genome sequencing projects.

We further outline that our synteny analysis is performed at the macrosyntenic scale (inter-chromosomal rearrangements) and thus relies on the correct assignment of genes to chromosomes, not on gene order. For this reason, our results would not be impacted by very local regions made difficult to assemble by putative recombination hotspots: crossing-overs take place within a pair of homologous chromosomes, and therefore would not affect the distribution of genes across chromosomes. The effect is also likely to be more marginal than the reviewer’s suggests, as, for a given locus, the presence of both parental copies will

14

remain predominant (in most species, maximum of 1 or 2 crossing over per chromosome in each sperm cell). Consistently, the different assemblers do recover the same chromosomes. We note that karyotypic differences amongst brittle star species are expected (and reported in the cited cytogenetic studies), as is the case for any clade whose members diverged 250 Mya. This holds all the more true if accelerated karyotype evolution is a feature of this group.

Finally, we would like to emphasise that our genome assembly approach is state-of-the art. Sperm is considered a starting material of choice for high-molecular weight DNA extraction due to its elevated DNA content and, accordingly, several high-quality genome assemblies for echinoderms have been generated using sperm or gonadal tissue, as listed in the table below:

Species	Study	Tissue
Crown-of-thorn starfish (Acanthaster planci)	https://www.nature.com/articles/nature22033#Sec2	testes and sperm
Purple sea urchin (Strongylocentrotus purpuratus)	https://www.science.org/doi/10.1126/science.1133609	single male's sperm
Green sea urchin (Lytechinus variegatus)	https://academic.oup.com/gbe/article/12/7/1080/5841217#206041244	interpyramidal muscle of Aristotle's lantern, tube feet, and the ovarian tissue
Japanese sea cucumber (Apostichopus japonicus)	https://journals.plos.org/plosbiology/article?id=10.1371/journal.pbio.2003790#sec017	Muscle, gonad , and respiratory tree tissues
Painted urchin (Lytechinus pictus)	https://academic.oup.com/gbe/article/13/4/evab061/6189048	sperm
Green sea urchin (Lytechinus variegatus) Feather star (Anneissia japonica)	https://www.nature.com/articles/s42003-020-1091-1	Lytechinus variegatus : sperm Anneissia japonica : sperm from a single male
Rock boring urchin (Echinometra lucunter)	https://academic.oup.com/gbe/article/15/6/evad093/7190441#407299505	muscle and gonad tissues
Japanese common sea star (Asterias amurensis)	https://www.nature.com/articles/s41597-023-02688-w#Sec2	Fresh gonad tissue from a male
Blue bat star (Patiria pectinifera)	https://www.nature.com/articles/s41597-023-02508-1#Sec2	gonads
Common sea star (Asterias rubens)	https://sangerinstitute.blog/2018/10/04/25-genomes-the-common-starfish/	sperm

15

2. The manuscript would benefit significantly from a deeper exploration into the biological implications of Hox cluster rearrangements. The authors should investigate the spatio-temporal expression of Hox genes specifically in the context of brittle star embryonic development. Additionally, the adoption of more advanced techniques, such as single-cell RNA sequencing and/or spatiotemporal transcriptomics, is also recommended, as such analyses would enable a more comprehensive understanding of the developmental regulatory roles of these rearrangements.

While we agree with the reviewer that these would constitute highly interesting paths to follow, we feel that they are out of the scope of the current study. Establishing protocols to culture and sample embryos and larvae throughout development and up to the juvenile stage (young adult) in brittle stars is an active area of research. However, despite constant improvements, mortality rates in culture are still too high to obtain enough material for high-throughput sequencing assays (especially for late stages).

Here, we investigate the genomic architecture of the Hox cluster in brittle star and reveal small-scale rearrangements within the cluster. This observation suggests that Hox clusters are permissive to rearrangements in echinoderms, as sea urchins also underwent convergent rearrangements in the Hox cluster. *Hox* gene function throughout echinoderm development remains largely elusive, but the *Hox* genes as a cluster appears to be most crucial for later larval stages rather than embryogenesis. As explained above, later stages are however the most challenging to sample. We hope that the high-quality genomic resource that we provide here, and the presentation of a second echinoderm class with a rearranged cluster, will instigate future studies on this topic.

3. The authors assert that "Tandem duplications of key genes contribute to brittle star larval skeleton and bioluminescence"(Line 291), yet they fail to provide functional analysis or evidence to support this claim. There is a possibility that these genes have acquired new functions in brittle stars. It remains to approve whether all or only specific copies of these paralogous genes are most crucial in these processes.

Functional analyses of the reported tandemly duplicated genes involved in larval skeleton and bioluminescence definitely represent exciting avenues to explore in future work, but are technically challenging and out of the scope of the current study. Established protocols for gene knock-out/knock-down (i.e. Crispr, RNAi, transgenic lines) are currently unavailable for brittle stars and would need to be developed. Moreover, duplicated genes often carry redundant functions, and even in model systems where these experiments are possible, they are generally uninformative as they do not yield directly interpretable phenotypes.

We now highlight this point as a limitation of our study: "[...] and will benefit from future in-depth functional characterisation" (line 327), and have toned down our claim: "Tandem duplications of key genes likely contribute to brittle star larval skeleton and bioluminescence" (line 295).

4. The discovery of co-expression clusters corresponding to three phases of arm regeneration is somewhat anticipated, which does not deviate from existing understanding

16

and thus lacks the significance of novelty. The authors acknowledge that “These are consistent with morphological timelines of regeneration in the brittle star and other animal systems” (Line 381). A deeper understanding of the genetic mechanisms of regeneration would be beneficial, necessitating more histological and functional studies to validate the findings. Among the several thousand co-expressed genes during regeneration, which are key in each temporal cluster? Key genes could be identified by their hub status in co-expression networks and by their impact on the regeneration process when knocked down/out. Further inquiries should explore the signaling cascades of these key genes in brittle stars and their conservation in Parhyale and axolotl. It would be insightful to determine which cells express these genes during regeneration across these species, and whether they share cellular and spatial similarities.

We disagree that a comprehensive identification of co-expression modules governing brittle star arm regeneration lacks the significance of novelty. Histological studies of brittle star arm regeneration have been previously published (Czarkwiani et al., 2016, <https://doi.org/10.1186/s12983-016-0149-x>) and are hence not one of the aims of the current study. Previous studies also focused on functional characterisation of a handful relevant candidate genes or on establishing lists of differentially expressed genes in regeneration as opposed to controls. Here, we provide a detailed characterisation of gene expression dynamics throughout regeneration. While it is consistent that these results align with previous morphological observations, our goal here is to document the genome-wide transcriptional programme of regeneration, which was unknown in brittle stars. The co-expression gene modules further serve as a basis for several follow-up investigations in our manuscript, including identification of TF binding motifs, signalling pathway enrichments and cross-species comparisons of regeneration gene expression dynamics. Together, these results represent relevant entry points for future functional studies.

We have updated the text to better highlight the novelty of our work: “*These results are consistent with morphological timelines of regeneration in the brittle star and other animal systems (Bideau et al. 2021; Srivastava 2021; Czarkwiani et al. 2016), but importantly capture the underlying genome-wide transcriptional programme of brittle star arm regeneration.*” (lines 387-388).

To address the reviewer's point about key genes in each co-expression cluster, we computed cluster membership scores that reflect the correlation between the expression profile of a given gene and the mean cluster profile (Kumar and E Futschik 2007). In this way, genes that form the core of each co-expression module are characterised by elevated membership scores. We provide detailed results in **Table S4** and show the top 5 genes for each cluster in **Figure S6E**, highlighting them as novel key regeneration gene candidates in the brittle star. This list includes both genes with homologs in vertebrates and uncharacterised brittle star genes (with notably several uncharacterised genes in the early response clusters, in line with our conclusions from comparative analyses regarding the limited evolutionary conservation of these genes). These genes represent attractive targets for follow-up functional investigations and complement **Figure S6D** where we show expression patterns for known brittle star regeneration genes (as recommended by the reviewer as well, see below). We integrate these new results in the main text of the manuscript: “*We corroborate the expression pattern of previously characterised brittle star*

17

regeneration genes and further report novel key candidates (Figure S6, Table S4)." (line 388-390).

We also note that functional information is unavailable for the axolotl or Parhyale, and hence would not be comparatively interpretable.

Figure S6: E. Expression of core genes in each co-expression cluster. Genes were filtered based on their cluster membership score (Table S4, "acore" score, (Kumar and E Futschik 2007)) to retain the top 5% core genes in each cluster, and the five genes with the highest expression were selected for the heatmap. Gene names starting with 'Unchar' indicate genes without significant blast hits in the swissprot database (Boutet et al. 2007).

5. While the study claimed the identification of conserved gene modules utilized by three species during regeneration, the evolutionary history behind this observation—whether it is ancestral origin or a product of convergent evolution—remain ambiguous. It is noteworthy that traits developed through convergent evolution often recurrently utilize similar gene sets. Notably, the number of genes shared among the three species is considerably fewer compared to those shared between any two species (154 genes versus 926 and 913 genes, respectively). This discrepancy suggests a potential under-representation of conserved genes across all three species. This study suffers from providing the strong evidence for distinguishing different hypotheses (ancestral origin or convergent evolution), partly due to the limited number of species included.

18

Regarding the lack of conclusive statements regarding the evolutionary origin of regeneration, we agree that such discussion is important to include in the manuscript. We have reworked the discussion section to integrate a direct interpretation of our results in the context of regeneration evolution: *"These results are consistent with two alternative scenarios for the evolution of animal regeneration: (i) convergence, with the independent evolution of wound response programmes able to recruit the ancestral proliferative machinery or (ii) homology, with an elevated divergence of wound response gene expression through diversifying selection, as typical for immune-related genes."* (lines 689-693).

The manuscript includes an analysis to explain why only a few genes are conserved across all three species (**Figure 5B**): this is due to (i) elevated sequence divergence and/or gene complement evolution in Parhyale and (ii) elevated expression divergence in axolotl. These results underscore the relevance of the brittle star as a novel regeneration model, since it shows lower rates of gene complement and gene expression evolution, thus facilitating comparisons across species.

Finally, we also emphasise that our ability to compare with other regenerating model systems is inherently limited by the available datasets. We have opted to investigate datasets with extensive temporal gene expression sampling, as it allows for precise comparisons of regeneration expression dynamics (no other datasets for adult appendage regeneration with > 4 time points and an available genome assembly (Bideau et al., 2021)). These comprehensive datasets have been crucial to our analysis revealing a tight conservation of proliferation gene expression dynamics. We definitely agree that a wider taxonomic sampling of regenerating animals in future years will be a crucial next step, and have integrated this point in the discussion: *"We nevertheless expect that future investigations into diverse regenerating animals with comprehensive temporal sampling will confirm the strong conservation of proliferation gene expression dynamics and untangle the scenarios we propose for the evolution of animal regeneration."* (lines 705-708).

6. The explant experiments conducted in the study rely predominantly on bulk RNA-seq, which lacks the resolution necessary for drawing definitive conclusions. Furthermore, it is a pity that the manuscript did not include carefully designed functional validation for the five genes identified as having opposite expression patterns during the wound healing and regeneration processes. Such functional assays may provide novel mechanistic insights into the difference between wound healing and regeneration.

It is unclear to us why bulk RNA-seq would lack the resolution to answer the question we are tackling with this experiment. Here, we investigate global temporal expression differences between two spatially distinct arm regions: the distal (regeneration) and proximal (wound closure) ends, as opposed to controls, and at different time points. It is thus crucial to generate high-quality reproducible transcriptomic datasets that retain both this spatial and temporal resolution. We achieve this by (i) segmenting the arm to sample each region of interest separately, (ii) sampling at different time points and (iii) sampling several biological replicates. Single-cell sequencing of full explants would completely miss this spatial information, which is crucial in our experiment, whereas generating as many single-cell datasets as we generated bulk datasets (20) would be highly costly and technically challenging.

19

We are similarly looking forward to future work into functional validation of the five identified candidate genes. However, we reiterate that functional validations are out of the scope of this work, as they represent a tremendous technical challenge in a non-model genetic system such as brittle stars. Future research efforts will have to develop and optimise protocols for gene knock-out and/or knock-down in this species (i.e. Crispr, RNAi and/or transgenic lines). The significant overlap we report here between gene expression in explants and adult animals further validates our explant culture as a relevant system to study arm regeneration in brittle stars, hence establishing a more practicable framework for future functional studies. In addition with the availability of the *Amphiura filiformis* genome, we hope that these will spur such future developments.

Minor issues:

1. The authors note that the *A. filiformis* genome is the most rearranged among echinoderms sequenced to date. It's important to explore the biological significance of identifying the 'most rearranged genome' and the evolutionary reasons for this. What are the DNA features around the rearrangement boundaries? Are they repeat-rich? What are the estimated ages of those boundary associated repeats?

We thank the reviewer for their suggestion. The difficulty is that there are no rearrangement boundaries that can be identified at this level of divergence. Indeed, our ancestral reconstruction was performed at the macrosyntenic level (i.e. we estimate inter-chromosomal rearrangements, see lines 149-159), as the closest available genomes to the brittle star *Amphiura filiformis* are sea star genomes, and these two groups diverged 480 Mya. Divergence with sea stars is way too ancient to reconstruct the ancestral local gene order (microsynteny) necessary to pinpoint breakpoint locations. For this reason, we advocate in our manuscript for future sequencing efforts in the brittle star group, and now outline as well that these genomes will be necessary to pinpoint breakpoints location: "*A more comprehensive sampling of brittle star genomes will provide additional context toward establishing the precise timeline of chromosomal rearrangements and pinpointing breakpoint locations, to then investigate the relative contributions of repeat expansion, chromatin architecture and population genetics dynamics to the rapid karyotype evolution in this group.*" (line 661).

One exception is the Hox cluster, where gene order is in general highly constrained and for which we were able to identify breakpoints and perform the analyses suggested by the reviewer. We found significant enrichment for a SINE element at breakpoints and estimated that this transposable element was active ~100 Mya.

2. The manuscript would benefit from further clarity regarding the brittle star genes that have expanded in relation to regeneration. Specifically, it is important to determine what proportion of these genes show significant regulation during the arm regeneration process. Additionally, what is the proportion of such genes in each co-expression cluster?

20

To address the reviewer's concern we have added a supplementary table with the number of genes from the expanded "regeneration" family in each co-expression cluster (Table S3). We also conduct enrichment tests to evaluate their over-representation in specific co-

expression clusters (enrichment results and corrected p-values in **Table S3**). We integrate the over-representation test results in a new panel (**Figure S6H**), showing that duplicates are associated with specific regeneration clusters.

See the extra panel below, for reference:

Figure S6. H. Duplicated genes from expanded 'regeneration' gene families significantly associate with specific regeneration co-expression clusters (hypergeometric test, Benjamini-Hochberg adjusted p-values). Significant associations (FDR < 0.05) are presented in colour, non-significant enrichments (enrichment ratio > 1 but FDR > 0.05) in grey (**Table S3**).

3. In Figure 3, the Ficolin-like and TRIM-like gene families show extensive expansion across echinoderms, whereas other regeneration-related gene families primarily expand in *A. filiformis* and *S. purpuratus*. Does this suggest a variation in regeneration capabilities among these species?

Sea urchins are able to regenerate spines and tube feet, with *S. purpuratus* indeed reported to display rapid regeneration of their spines (Scholnick and Winslow 2020, <https://doi.org/10.1371/journal.pone.0228711>). However, and even though regeneration capabilities are highly variable across echinoderm species of the same class, sea urchins are usually regarded as the group with the weakest regeneration abilities, consistent with our global finding here: sea urchins expanded gene families have the lowest enrichment and highest p-values for the GO term "regeneration" (with the exceptions of the crinoid, but this might pertain to the annotation of the only crinoid genome available). We would therefore guard against interpreting the *S. purpuratus* data for the 7 families highlighted as relevant for

21

brittle stars, as these might not be a good reflection of echinoderm regeneration capacities as a whole.

4. It is intriguing that in Figure 4, the non-regenerating control group exhibits the highest expression levels of wound-response genes (clusters A1 and A2), particularly since clusters A1 and A2 include many 'regeneration-related' expanded genes shown in Figure S6. This observation prompts further investigation into whether these genes are involved in other non-regenerative processes in the control group.

We were similarly intrigued by the elevated expression of A1 and A2 genes in the control sample. A1 genes are progressively turned down as regeneration proceeds and A2 genes are slightly turned up at 48 hpa and silenced by Stage 3.

It is indeed very likely that genes from these clusters are also implicated in broad immunity-related processes, and are fine-tuned in the context of regeneration. Denser samplings of early regeneration time points might reveal a more precise picture of immune/wound healing processes at the onset of regeneration. We did attempt to generate RNA-seq data for earlier time points, but all resulting data failed our quality checks and were not retained for analysis, highlighting that early regenerates are challenging biological material to work with.

We now note this as a limitation of our study: "*Denser temporal samplings of early regeneration are necessary to confirm the limited conservation that we observe here, but are currently technically challenging in the brittle star model.*" (lines 708-710).

5. The author should examine the expression patterns of genes involved in keratan sulfate metabolism (Line 350) during regeneration, to determine if the variations in expanded and contracted genes indicate the 'specialization of glycosaminoglycan metabolism'.

We thank the reviewer for their recommendation. We have investigated the expression of these genes during regeneration and produced similar figures as those we generated for the "regeneration" GO term. As for the regeneration GO terms, gene members of these families are expressed during regeneration, and some preferentially associated with specific expression clusters (Figure S6). A corresponding supplementary table (Table S3) has accordingly been added. We now ensure that our interpretation of these patterns is carefully phrased as an hypothesis, rather than a strong claim.

These panels are now part of Figure S6, and shown below for reference:

22

11

ESS
: is

Figure S6. I. Expression throughout arm regeneration of the brittle star genes in the expanded and contracted gene families annotated with the GO term 'keratan sulfate metabolism' (see **Figure 3B**). Representation is as in G. Note that one identified contracted gene family contains no brittle star genes (ST3GAL1-like) and is thus absent from the figure. **J.** Genes from expanded and contracted keratan sulfate gene families are associated with specific regeneration clusters (**Table S3**). Representation is as in H.

6. It is confusing that in their previous papers (Czarkwiani et al. Development 2021, Piovani et al. BMC Biol 2021), the authors suggest FGF signaling and *alx1* genes play important roles during the regeneration process of brittle stars, which was neither discovered nor mentioned in this paper.

We have added an extra supplementary panel with a heatmap displaying the expression of genes that were previously investigated in the context of brittle star arm regeneration, including *fgf* and *alx1* genes (see below for figure). Expression patterns corroborate previous studies, and we integrate this observation in the main text of the manuscript: "We corroborate the expression pattern of previously characterised brittle star regeneration genes and report novel key candidates (**Figure S6, Table S4**)." (line 388).

While we completely agree that this data represents an interesting confirmation to integrate to the manuscript, we also point out that the strength of our current study is to investigate genome-wide temporal expression patterns in regeneration (~30,000 genes). The aims of our current study are hence distinct and complementary to previous investigations into the function of carefully selected candidate genes. In fact, the two studies cited by the reviewer focused on skeleton regeneration, and on specific genes that were known to be involved in skeleton development in sea urchin (~70 genes).

23

Figure S6: D. Expression of brittle star genes previously implicated in arm regeneration (Czarkwiani et al. 2022; Piovani et al. 2021; Czarkwiani et al. 2021). Co-expression clusters are shown on the left, gene names on the right, with red indicating availability of published *in situ* data (gene names from previous studies, see Table S4).

7. In Line 267, the authors reference Annunziata et al. (2014) to suggest that Hox7 and Hox11/13b may play important roles in embryogenesis. However, it didn't mention Hox7 at all in this cited paper. The correct citation should be Li et al. (Cell Discovery 2018).

We thank the reviewer for catching this error and have reorganised the references to better reflect the contribution of Li et al. Cell Discovery 2018.

Reviewer #2 (Remarks to the Author):

In this manuscript, the authors present a chromosome-level genome assembly of the brittle star *Amphiura filiformis*. These data enable the authors to compare various brittle star genomic features with other echinoderm species and make inferences about echinoderm evolution, as well as to combine these genomic data with transcriptomic datasets to compare the brittle star's appendage regeneration with other regenerative taxa. First, by using synteny analyses to compare the *A. Filiformis* genome with other echinoderm members (sea urchin, sea star, and sea cucumber species), they find that the brittle star genome is the most rearranged at a chromosome level relative to the predicted ancestral chromosomal arrangement. They next demonstrate the power of these data to address comparative evolutionary questions by assessing finer-scale genomic features using this new assembly. First, they use Hox genes as a case study to investigate smaller-scale rearrangements in echinoderms, discovering that the brittle star Hox clusters are highly rearranged relative to other echinoderm species, and that these rearrangements seem independent of Hox rearrangements in other echinoderm members. They also identify tandem duplications in genes associated with key features of brittle star biology - larval skeletal and bioluminescence-related genes - and by identifying some of these genetic components suggest possible mechanisms for the evolution of these traits. Finally, they ask about gene family expansion and contraction in brittle stars relative to other echinoderms and find that many of these members are associated with regeneration-related GO terms, and in fact that many of these genes are expressed during brittle star regeneration based on previous transcriptomic datasets. They propose some of these identified changes could help explain the brittle star's high regenerative capacity.

Next the authors transition to investigating regeneration more deeply using both their genomic data as well as pre-existing and newly generated transcriptome data. They use soft clustering to identify sets of co-expressed genes, which they find correspond to known morphological features in brittle star regeneration that are also more broadly observed in additional regenerative taxa: wound-healing, proliferation, and differentiation. They then use transcriptomic time courses in appendage regeneration from axolotl and parhyale to find sets of co-expressed genes in those organisms, and then compare these clusters between species. They find that clusters enriched in proliferation genes are most similar, and suggest this may point to co-option of an ancient metazoan proliferation program across regenerative taxa. Additionally, they find that the general order of these types of processes in regeneration is similar across species. There is little correlation in gene membership between clusters associated with differentiation, as might be expected if these animals are all regenerating different tissues. There are few features at the gene level that are conserved across all three species, including enrichment of motifs near TSS of coexpressed brittle star genes; the few that are shared include two transcription factors that are related to proliferation in other systems and thus they suggest might be related to this potential shared proliferation program. Finally, the authors investigate differences in regenerative and non-regenerative wound-response in brittle stars by doing explant experiments, in which one portion of the limb heals the wound, whereas another portion undergoes regeneration. By comparing RNA expression between these sites, and across two timepoints, they identify an enrichment of proliferation-related genes in the regenerating vs non-regenerating datasets, many wound-healing genes that are shared, but also general wound response/wound healing genes that are specific to the regenerative sites.

25

This work demonstrates the power of the brittle star as a comparative model in multiple aspects of evolutionary/developmental biology. The genomic and transcriptomic datasets generate many hypotheses related to echinoderm body plan evolution and shared features of regeneration to be investigated further in the future. A clearer narrative connecting the various aspects of the paper would help readers unfamiliar with brittle stars navigate their many findings. The authors also present many possible mediators of regeneration that were identified in this study, but there are some important caveats to these conclusions that should be addressed, and it should be made clearer that these are important avenues for further investigation rather than definitive answers to what drives regeneration in the brittle star and other species. In particular, the authors are overinterpreting the proliferation gene data - all animals will need proliferating cells to be able to regenerate new tissue, and we already know that metazoans utilize very conserved machinery for controlling cell proliferation. Therefore, finding shared gene modules associated with proliferation would almost be a null hypothesis of sorts for this type of comparison. The analyses in this paper are still meaningful, but the authors should rewrite their interpretations of these results.

We thank the reviewer for their generous feedback and insightful comments. We have reworked the text to integrate the reviewer's suggestions and conducted additional analyses to address the reviewer's concerns. We concur about the reviewer's comments regarding the conservation of animal proliferation machinery genes, and have clarified the text in that respect. However we outline that demonstration of its strongly conserved temporal expression dynamics in regeneration was previously elusive. We have carefully followed the reviewer's suggestions to phrase our conclusions in a way that makes it clearer that our study paves the way for future functional and comparative investigations of animal regeneration (see below).

Major comments:

Narrative cohesiveness:

The paper seems divided into two major themes: using the genome to make comparisons between echinoderms, and then relying more heavily on transcriptomic data to make comparisons related to appendage regeneration between more distantly related taxa (brittle stars, parhyale, and axolotl). The rationale for linking these two themes within this same narrative, and how each section motivates the next, could be clearer. Similarly, how do the findings from the comparative regeneration work motivate the explant experiments comparing regenerative and non-regenerative wound healing in the brittle star? As written, these sections seem somewhat disjointed.

The title and framing of this study also centers the work around the new genome, but the regenerative portion of the manuscript and the conclusions that are drawn within it rely much more heavily upon the transcriptomic datasets. Making it more explicit how having the new genome makes it possible to complete and/or augments the analyses in this paper beyond what is possible only using transcriptomic data would help justify the title and framing of the work (for example, without the genome it was possible to look at gene expression, but motif enrichment analyses were not possible, and this was important in generating hypotheses regarding gene regulatory modules; or it was difficult to say without the genome if gene family expansions were actual gene duplications or just splice variants).

26

We thank the reviewer for their suggestion and have outlined in the introduction how the availability of a high-quality genome assembly is crucial to address the questions we tackle in our manuscript: *“Here, we report a chromosome-scale genome assembly for the brittle star *A. filiformis*. This unique, high-quality genomic resource is crucial to accurately capture the brittle star gene repertoire and provides a rigorous framework to probe genome-wide gene expression patterns during regeneration.”* (lines 115-117). We also add a sentence to motivate the explant experiments, where we investigate differences between regenerative and non-regenerative wound healing: *“We have comprehensively characterised the genome-wide gene expression dynamics during brittle star arm regeneration and assessed its evolutionary conservation in other animal systems. However, these frameworks do not allow to directly interrogate the molecular drivers of regenerative as opposed to non-regenerative wound healing processes. To define what differentiates [...]”* (lines 577-580).

Clarification of previous knowledge:

In the regeneration portion of the manuscript there are many genes/pathways that the authors identify in their analyses and they note are previously known to have roles in regeneration or regeneration-related phenomena. It is often unclear whether this previous knowledge is related to a broad role across many taxa, was known to have that role in some other echinoderm (or echinoderms generally), or whether this was specifically something that had been previously identified in the brittle star – citations are provided, but it would be helpful for the statements to be more clear about this. Similarly, it isn't always clear when a feature of the transcriptomic data is a new finding or is corroborating past knowledge (is this something new and exciting? Or is it something that gives us confidence in the data and existing new avenues to pursue further?). Clarifying these points will help guide readers less familiar with brittle star/echinoderm literature, and also help place the comparative work with other species in the broader context of previously known shared features of regeneration.

Related to this, expanding the schematic in Figure 4A depicting the transcriptomic sampling to include additional information corresponding to known features of regeneration relevant to the morphological stages that were previously identified would help readers unfamiliar with brittle star regeneration make more sense of the analyses that follow. For example, the schematic could include summarized results from EdU experiments in other studies, pointing out when the blastema-like outgrowth appears, and when differentiated structures begin appearing. Some of this information is in the text (ie, stage 5 is noted to be when peak proliferation occurs), and having it summarized in one place would help guide the reader through some of these findings.

Genome-wide gene expression dynamics throughout brittle star regeneration were never characterised in any previous studies, as such most of our findings here are novel, whereas most of the previous knowledge comes from vertebrate systems.

To better emphasise what was previously known in the brittle star, we have included a supplementary heatmap figure that shows the expression of previously investigated genes (Figure S6D). We have also updated the main text accordingly: *“These are consistent with morphological timelines of regeneration in the brittle star and other animal systems (Bideau et al. 2021; Srivastava 2021; Czarkwiani et al. 2016), but importantly document the underlying genome-wide transcriptional programme of brittle star arm regeneration. We corroborate the expression pattern of previously characterised brittle star regeneration genes and report novel key candidates (Figure S6, Table S4).”* (line 385-390).

27

We also clarify that most of the previous knowledge we discuss comes from studies in vertebrates: “*The early activation of NF- κ B in the context of regeneration has been evidenced in vertebrates and hydra (Karra et al. 2015; Straughn et al. 2019; Wenger et al. 2014), and our findings suggest its implication in the brittle star regenerative response as well.*” (lines 396-398), and “*These TFs have not been previously investigated in the context of brittle star regeneration but are functionally well-characterised in vertebrates.*” (line 404-406). We highlight concordance with previous studies of brittle star regeneration regarding two signalling pathways: “*Notably, the VEGF and Akt pathways have been previously implicated in brittle star regeneration (Purushothaman et al. 2015).*”

With respect to morphological knowledge of arm regeneration in the brittle star, we prefer not to overcrowd the figure more than it already is, as there are already many published studies with very detailed schematics (as this was the main focus of those previous studies): overcrowding the figure might result in the opposite effect of overwhelming the unfamiliar readers. Instead, to guide the reader and have everything at the same place as recommended by the reviewer, we have included more detailed information in the associated legend: “*Early stages are sampled at 48 and 72 hpa (hours post-amputation), when wound healing followed by regenerative bud formation occurs. Subsequent stages are defined by morphological landmarks: Stage 3 corresponds to the appearance of the radial water canal and nerve (~6 days post-amputation, dpa), Stage 4 is the appearance of the first regenerated metameric units (~8 dpa), Stage 5 corresponds to advanced arm extension and differentiation onset (~9 dpa), 50% stages correspond to when 50% of the regenerated arm has differentiated (~2-3 weeks post-amputation) sampled at the distal (D, less differentiated) and proximal (P, more differentiated) ends (Dupont and Thorndyke 2006; Czarkwiani et al. 2016).*”

Proliferation as the most similar feature across these regenerative taxa:

In comparing the transcriptomic datasets across three species, the authors identify proliferation as the most conserved feature of regeneration across taxa at a transcriptomic level, and propose that a common theme of regeneration might be an ancient metazoan proliferation program. There are a couple of discussion points we think should be addressed here. First, if proliferation is an important component of regeneration in all of these taxa, is it surprising that the genes involved in regenerative proliferation are similar between species, or is it simply that proliferative machinery in general is highly conserved across metazoa? Are there signatures of these regenerative proliferation-related co-expression modules that are distinct from proliferation in these organisms in other contexts?

We argue that wound response and differentiation are also important and tightly regulated processes in the context of regeneration, however we find very limited gene expression conservation related to these phases. While the proliferative machinery is known to be ancient (as reflected here also by our gene age analysis), its conserved temporal gene expression dynamics in the context of regeneration had not been previously directly evaluated. It is challenging to confidently assess whether parts of these proliferation modules are specific to regeneration, but our comparison of gene expression dynamics in brittle star regeneration versus Parhyale development suggest that proliferation modules are globally also used during development (early embryogenesis, **Figure S9E**). Together, this allows us to propose scenarios for the evolution of animal regeneration that rely on a

28

homologous proliferation machinery: *"We revealed that the proliferative phase of regeneration displays the highest expression conservation across these animals, suggesting that regeneration deploys an ancient, evolutionarily conserved proliferation machinery. These results are consistent with two alternative scenarios for the evolution of animal regeneration: (i) convergence, with the independent evolution of wound response programmes able to recruit the ancestral proliferative machinery or (ii) homology, with an elevated divergence of wound response gene expression through diversifying selection, as typical for immune-related genes."* (lines 689-693).

The second point relates to the timepoints that are included in this study, and how these might allow for comparison of the wound-response/healing component of the regenerative program. Based on the work presented here and in Czarkwiani et al, 2016, it seems like the earliest time point sampled in this study (48 hpa) is nearing the end of the wound-healing phase and entering the proliferation phase. From other regenerative systems, it is known that many of the wound-induced signals that are required to launch regenerative responses occur very quickly following injury, often before any morphological changes are discernable in the organism. Has any attempt been made to look at, for example, a 6 or 12h time point? We do not necessarily suggest adding these times into the analysis if they do not exist, but rather addressing the limitations of the existing data to identify overlap in the earliest wound-induced networks of regeneration. In line with this, the expression profiles of genes in clusters A1 and A2, which are proposed to correspond to the wound healing process, seem to also have high expression in the control tissue - is this to be expected of genes involved in the wound response?

We did indeed attempt to generate RNA-seq data for earlier time points. However, all resulting data failed our quality checks and were not retained for analysis, highlighting that early regenerates are challenging biological material to work with. Sampling earlier regeneration time points in the brittle star is thus technically challenging, and it is correct that it constitutes an area of improvement of our study.

We were similarly intrigued by the elevated expression of A1 and A2 genes in the control sample. A1 genes are progressively turned down as regeneration proceeds and A2 genes are slightly turned up at 48 hpa and silenced by stage 3. One highly speculative hypothesis would be that immune processes are highly active in brittle stars (in relation to their mud-dwelling lifestyle and high exposure to microorganisms), and that their expression is finely tuned in wound healing/regeneration. Using two previously published adult arm transcriptomic datasets (Purushothaman et al. 2015; Delroisse et al. 2014), we verified that our set of manually curated A1/A2 immune genes were also expressed in these controls (also see **Figure S9D**). Our interpretations of these observations remain very speculative and we concur with the reviewer that sampling earlier time points in regeneration, if possible, would allow for more precise characterisation of immune and wound healing processes at the onset of regeneration.

We now outline this in the discussion: *"Denser temporal samplings of early regeneration are necessary to confirm the limited conservation that we observe here, but are currently technically challenging in the brittle star model."* (lines 708-710).

29

Because comparisons more broadly to regeneration are invoked, it might also be interesting to place these findings in the context of common themes in regeneration. What do these findings suggest about the evolution of regeneration, or where we should be looking if we want to better understand common regenerative themes? i.e., do these results point to each species independently evolving a wound response that can feed into a common proliferative process, which then again transitions to species-specific differentiation depending on what exactly the animal has to build? How would we expect this to relate to other types of regeneration, like whole body regeneration or different types of structural regeneration? These also often follow the general pattern of wound response, some degree of proliferation, and differentiation. If the argument is that appendage regeneration is not homologous between these species, yet we see the common proliferative response, then would we also expect to see this in other regenerative modes (would we expect this same proliferation program to be involved in the proliferation phase of whole body regenerators)?

We thank the reviewer for their insightful suggestions and comments. We have updated the discussion to reflect our vision relating to these questions, and believe this makes a valuable addition to the manuscript.

We have added: *"We revealed that the proliferative phase of regeneration displays the highest expression conservation across these animals, suggesting that regeneration deploys an ancient, evolutionarily conserved proliferation machinery. These results are consistent with two alternative scenarios for the evolution of animal regeneration: (i) convergence, with the independent evolution of wound response programmes able to recruit the ancestral proliferative machinery or (ii) homology, with an elevated divergence of wound response gene expression driven by diversifying selection, as typical for immune-related genes."* (lines 689-693) and: *"Alternatively, they could reflect genuine biological differences of (larval) whole body regeneration studied in (Cary et al. 2019) and the adult appendage regeneration we investigate here. We nevertheless expect that future investigations into diverse regenerating animals with comprehensive temporal sampling will overall confirm the strong conservation of proliferation gene expression dynamics and untangle the scenarios we propose for the evolution of animal regeneration."* (lines 705-708).

Motif Analyses:

These binding motif enrichment experiments identify potential regulators of some of the regenerative processes described in this work. However, a total of 6kb surrounding a TSS is likely to harbor many potential binding sites, especially if the regions have not been narrowed down to those that are accessible at a particular time (and thus more likely to actually be involved in transcription factor binding/downstream gene expression at that time). Is there any evidence that transcription factors that might bind to the identified motifs are expressed at times consistent with them regulating the sets of coexpressed genes that showed enrichment of those motifs? For example, the YY1 motif is enriched at loci corresponding to transcripts in the A5 and A6 clusters; is there a gene corresponding to the YY1 transcription factor expressed at or before the onset of those genes at Stage 3? This could help strengthen the hypothesis that these factors are involved in the regenerative processes proposed.

The reviewer is correct that our analysis is not based on the identification of accessible regions: we search for significantly overrepresented motifs around TSS of genes sharing

30

similar expression dynamics, and interpret these results as indirect evidence that these motifs might reflect binding sites of transcription factors that regulate specific co-expression gene modules.

As recommended by the reviewer, we have added gene expression heatmaps in Figure S6 in an attempt to solidify this evidence. Globally, these results show that genes encoding the identified key TFs (NFKB, YY1, NRF1, RORA, PRDM14, ZBTB12) are effectively expressed at the corresponding regeneration stages in the brittle star. We however note two exceptions: znf768, which seems to lack identifiable homologs outside of vertebrates and p53 whose expression pattern does not match the motif enrichment results (but activity of the p53 signalling pathway appears consistent with the motif analysis, see Figure S6C). One difficulty is that databases for TF binding motifs are heavily biased towards vertebrates, and we thus rely on the generally accepted assumption that orthologous TFs across species have similar binding motifs. We integrate these confirming results and note the caveats in the text and figure legend: “While we note that binding motif overrepresentation analyses are inherently biased towards more studied vertebrate systems, TF gene expression in the brittle star is globally consistent with reported motifs enrichment (Figure S6).” (lines 412-415).

Figure S6: F. Expression of key TF genes during regeneration, as identified by binding motifs overrepresentation analysis (Figure 4D). TF genes were identified by reciprocal blasts with mouse and swissprot blast hits; several copies were reported where blast results were ambiguous. TF genes with consistent expression and binding motifs overrepresentation are shown in red. No homolog for ZNF268 could be identified in brittle star and the expression of the identified p53 homolog does not match motif enrichment results (but p53 pathway activation is consistent with p53 motif enrichments, see C).

Minor comments:

Lines 135-138: Why were these specific gene groups chosen for inclusion into manually curated lists? Were they previously known from brittle star biology? Important generally in body plan evolution and/or regeneration? Key pathways/processes that similar genomic studies tend to assess?

31

Gene lists were globally selected to be relevant to the study of regeneration and brittle star biology. Specifically, TFs were included as the main effector of gene regulation, and immune, stemness, neuronal genes and members of key signalling pathways to investigate their importance in regeneration processes.

Lines 244-265 - In discussion about Hox colinearity and spatio-temporal expression, is there any knowledge about spatial expression in brittle stars, or more broadly other echinoderms?

Hox collinearity and spatio-temporal expression remain poorly documented in echinoderms (mostly studied in sea urchins): it appears that the Hox cluster has a limited role in embryogenesis but is likely important for later (larval) development. These later stages are the most difficult to obtain in laboratory culture but we expect them to become a major point of focus in future studies. Previous knowledge about *Hox* gene expression is discussed in the manuscript (see lines 247-273), we have clarified that only few of the Hox genes appear to be important for echinoderm embryogenesis.

Lines 293-322 ("Tandem duplications...") - are there other examples of tandem duplications found in the genome, or were these two groups used as a candidate case study based on previous echinoderm knowledge?

These were indeed chosen as candidate case studies based on previous work, we have clarified in the text: "*In echinoderms, two specific gene families have previously come into focus as relevant examples of lineage-specific evolution through tandem duplications: phb/pmar1 (larval skeleton) and luciferases (bioluminescence) (Delroisse et al. 2017; Mariétaz et al. 2023; Dylus et al. 2016).*" (line 299).

Line 375 - It could help set up the findings in this section to state at the beginning that the stages were chosen to correspond to the known regenerative phases from previous regeneration studies; then the clusters confirmed this knowledge and allowed for further investigation (this also helps establish for readers less familiar with brittle star regeneration what was previously known).

We have clarified this point : "*Representative stages were selected on the basis of well-established morphological landmarks of brittle star arm regeneration (Czarkwiani et al. 2016) (Methods, Figure 4A).*" (line 380). We also emphasise that genome-wide expression patterns in brittle star arm regeneration were previously uncharacterised: "*These results are consistent with morphological timelines of regeneration in the brittle star and other animal systems (Bideau et al. 2021; Srivastava 2021; Czarkwiani et al. 2016), but importantly capture the underlying genome-wide transcriptional programme of brittle star arm regeneration.*" (lines 387-388).

Line 416 - It seems surprising that axial specification would be occurring so late in regeneration, is this consistent with previous reports in brittle stars?

We thank the reviewer for catching this: it was an overstatement only based on one putative role of *tbx3* during sea urchin development. We have rephrased to avoid making such a specific statement about the timing of axial specification: "*This cluster includes two T-box TFs that are important for patterning in echinoderms (tbx3-1 and tbx3-2) and two TFs with*

32

key roles in neurogenesis (ngn1-like and hey1-like) (Gross et al. 2003; Slota and McClay 2018; Slota et al. 2019).” (line 450).

Line 469 - Being clearer in the main text about the criteria for considering a cluster to be co-expressed would help to understand the analyses in this section - something like “clusters were considered co-expressed between two species if there were more shared genes than expected by random chance,” then readers who want more details can read about those in the methods.

We have rephrased to integrate the reviewer’s suggestion: “We used pairwise comparisons and permutation tests to reveal conserved co-expression clusters across species. Specifically, co-expression clusters were defined as conserved between two species when they employed more shared genes than expected at random (Figure 5B, Methods).” (lines 484-485).

Line 671 - In the discussion, expanding the finding that proliferation seems conserved to talking about a common proliferative cell type seems like a stretch. This work doesn’t address the cellular context of the proliferation program in these organisms or look more deeply at what it would mean for them to be homologous cell types.

We concur with the reviewer and did not mean that we addressed the cellular context of regeneration, only that our results might align with a previously proposed hypothesis in the field (which has not been supported with significant evidence either). We have rephrased to better reflect this: “The conservation of proliferation further ties in with a current hypothesis in the field that animal regeneration may recruit a homologous proliferating cell type (Srivastava 2021; Lai and Aboobaker 2018), but this should also be further explored with single-cell sequencing techniques and additional comparative analyses.” (lines 705-714).

Reviewer #3 (Remarks to the Author):

This manuscript focuses on the genomics and transcriptomics of the brittle star *Amphiura filiformis* with a focus on adult arm regeneration. This represents the first sequenced brittle star genome. It is a comprehensive study that is clear, well-written, and provides a broad evolutionary context by putting the findings in the context of other echinoderms as well as other invertebrates and vertebrates. For the appendage regeneration expression section, a major focus of the paper, they first characterize gene expression during three phases of arm regeneration and then compare co-expression clusters with both the crustacean *Parhyale* and with axolotl providing useful context on what aspects of the regeneration program are conserved across these three animals. They finish with explant RNAseq experiments to compare gene expression during wound closure and regeneration in their study organism.

This study will provide important resources for the community to include brittle stars in comparative analyses. For a genome manuscript, it was quite a pleasure to read, and highlights key features of the genome. The figures complement the text nicely and are clearly presented.

We sincerely thank the reviewer for their encouraging comments and helpful suggestions to streamline the manuscript and improve its presentation. We respond to each suggestion below.

Below, I highlight a few areas that could use some attention and clarification in the manuscript:

1. In the section "Expanded gene families in echinoderms are enriched in regeneration-related Processes" the authors state that "recurrent duplications of "regeneration-related" genes may underlie the remarkable regenerative capacity of many echinoderm species. Notably, in *A. filiformis*, members of these expanded gene families (Figure 3C) are expressed during arm regeneration (Figure S6)." I would like the authors to clarify a few points in this section:

Figure 3A shows that there are 266 gene families which have expanded in *A. filiformis*. It would be helpful to know how many total "regeneration-related" genes/genes annotated with the GO term "regeneration" there are in the *A. filiformis* genome overall. Then, by comparison, how many of these "regeneration-related" genes are found in expanded gene families - only 7, correct? I appreciate Figure 3B but would like to know how many genes from each GO term shown are found in the genome versus how many are in expanded or contracted families. This could be done in a supplemental format or added to the figure.

To clarify, the Gene Ontology terms that we report in Figure 3B are the ones that are statistically significantly enriched in expanded or contracted families when compared to all considered gene families as background. More specifically, for the "regeneration" GO term: 27 families among the 125 analysable expanded in the brittle star (266 total expanded but only 125 with at least one GO annotation) are annotated with the 'regeneration' term, 222 in all 7295 considered families, the enrichment ratio is 4.8 and the adjusted p-value is 7.13×10^{-12} . The 7 families that we show on the Figure 3C are the most relevant regeneration

34

expanded families (> 2 brittle genes in the family annotated with the 'regeneration' GO, as specified in the legend).

For each GO term, the associated p-values, enrichment ratios and brittle star genes are available in **Table S3**, and we have added a column detailing the number of gene families in background and foreground that are annotated with a given GO term. We have clarified in the legend for Figure 3B, by adding the following: *"Complete GO enrichment test results are provided in Table S3, including p-values, enrichment ratios, background and foreground gene families and brittle star genes."*

From Figure 3C it looks like there are 7 gene families listed that are "regeneration-related" and expanded in *A. filiformis*, which differs from other echinoderms and which is quite interesting. It is not clear if the four gene families associated with coagulation and/or clotting in vertebrates are included under the GO term "regeneration" or if these were found independently?

The 4 gene families associated with coagulation and/or clotting in vertebrates are indeed amongst the 7 families annotated with the GO term "regeneration". We have clarified this point in the text: *"Additionally, genes within four of the seven "regeneration-related" expanded families (plasminogen, carboxypeptidase B, coagulation factor and ficolin) directly regulate coagulation and/or clotting in vertebrates (Pryzdial et al. 2022), [...]"* (line 349).

Same question about the gene family associated with keratan sulfate metabolism. Is this gene family annotated with the GO term "regeneration" or was it found independently from the previous 2017 study on this topic?

For those who are not familiar with keratan sulfate metabolism, it might help to define it and explain that keratan sulfate is a glycosaminoglycan in the text. It might also help to list the gene family name (UDP-glucuronosyltransferase) in the text when talking about the specific gene family that has expanded that is related to keratan sulfate/glycosaminoglycan metabolism. Right now, the reader has to deduce that this is the specific gene that is related to this section of the text by looking at Figure 3C.

We thank the reviewer for reporting that this was unclear. No, keratan sulfate metabolism is a GO term displayed in Figure 3B, not 3C (4th term starting from the end). Panel C only lists expanded gene families associated with the GO term "regeneration", as specified line 356: *"Finally, genes involved in keratan sulfate metabolism are over-represented in both expanded and contracted gene families in the brittle star (Figure 3B)." and legend for Figure 3C: "Gene copy number variation across echinoderms for regeneration gene families with significant expansion in A. filiformis [...]"*. To further clarify, we have put the term "keratan sulfate metabolism" in bold (as we did for regeneration) in Figure 3B and added a supplementary heatmap with the gene families in this pathway and their expression in brittle star arm regeneration (**Figure S6I,J**). The reviewer is correct that this result is independent from the 2017 study: it is one of the enriched terms that we find with our GO enrichment analysis of expanded and contracted gene families. We have rephrased to clarify this point.

35

2. For the section "Gene expression during brittle star arm regeneration recapitulates major regeneration Phases" both transcription factors and evidence for "binding motifs associated with several TF" enriched around the TSS of genes expressed during regeneration are

discussed. I see in the Methods how the transcription factor binding motif enrichments were identified with HOMER, but can you expand on how you identified transcription factors? In the Methods section under “Gene lists curation” I found this sentence “We generated a list of TFs based on the presence of DNA-binding PFAM domains.” Can this be described in more detail? How exactly did you run the PFAM search? Did you use hmmscan or InterProScan or another method? Were default parameters and e-value cutoffs used? How did you curate the results for DNA-binding domains?

We ran the PFAM domain search using PfamScan. More specifically, we used the `pfam_scan.pl` tool with default parameters, which applies PFAM-curated family-specific e-values thresholds. This information is now added in the Methods: “*PFAM domains were annotated on protein sequences using the `pfam_scan.pl` tool run against the PFAM-A HMMs database, with default parameters (i.e. PFAM-curated statistical thresholds).*”. We then compiled a list of PFAM domains that correspond to known DNA-binding domains and retained as Transcription Factors all genes whose protein contained at least one of these domains. To address the reviewer’s concern over the lack of details on this step, we now provided the compiled list of curated DNA-binding domains in **Table S2**.

3. Is there a difference between “Gene lists enrichment tests” and “Gene ontology enrichment tests”? Were both performed with “the enricher function from the ClusterProfiler R package” as described in the Methods? In the manuscript this is unclear. Please clarify.

This uses the same approach as for the GO, but is coded directly in python (hypergeometric tests with correction for multiple testing instead of using ClusterProfiler. We have clarified this in the associated Methods section, which is now labelled “**Gene ontology and gene list enrichment tests**”: “*Similarly, for gene list enrichment and depletion tests on the regeneration co-expression clusters (Figure 4, see previous section for gene lists curation), we used the same foreground and background genes definition as for the GO enrichment tests above, and performed hypergeometric tests with a correction for multiple testing with the Benjamini-Hochberg procedure, with the same statistical threshold as for the GO (FDR<0.05).*”

4. In the Explant experiments section, two of the five genes with the “drastically opposite expression patterns in the wound healing and regenerating segments” are discussed in the context of vertebrates/mice but it would be nice if the other three genes could be more clearly discussed regarding if their functions are known from vertebrates or not to support the final sentence of the section. Or perhaps you could add a bit to the Discussion on this part of the paper.

Among the three other genes, two are uncharacterized (*AFI33635* and *AFI18858*, which do not have any significant Blast hit in the swissprot metazoa database) and one is described in sea cucumber only (*AW-SPI*). We discuss putative functions for these genes in the text, in relation with their predicted PFAM domains, but no further information is known for these genes. We have clarified the final sentence of the section as follows: “*In summary, these five candidate genes might be tightly linked with the transition from wound healing to regeneration-induced cell proliferation, and some may have a conserved function in the brittle star and vertebrates (Agrin and Gdf8), while the three others lack identifiable homologs in vertebrates and remain to be further characterised.*”

36

5. In the Methods for the “Arm regeneration RNA-seq in brittle star severed arm experiments (explant)” the sampling is described well except for explaining how the “43 groups of 150-200 explants” relates to the final samples. How do these groups relate to the sampling done at two time points (3 days and 5 days) and three sections: proximal, medial and distal? In the end, I would like to know how many replicates were there for each time point and each section? So how many samples total were sequenced? I think this could be more clearly described and would give the reader more confidence to evaluate whether the experiment was designed well with enough replication.

We thank the reviewer for catching this inconsistency.

We have added a legend detailing the terminology for the explant replicates listed in **Table S1** and have made the number of replicates evident in the legend for Figure 6: “*Proximal, distal and medial (control) segments are sampled for RNA-seq at 3 and 5 days post-amputation (dpa), using 3-4 replicates each, see Table S1.*”.

We also corrected the corresponding Methods statement (line 882), as the 20 samples indeed correspond to 20 groups of 150-200 explants (the number of 43 groups was left uncorrected and included samples that were not retained in the manuscript because they failed quality checks). We have amended as follows: “**20 samples (each sample consisting of a batch of 150-200 explants)** were cultured in flow through aquaria at 16°C. Explants were sampled at 3 and 5 days post-amputation (dpa), sedated in 3.5% w/w MgCl₂ in artificial seawater for 15 minutes and then dissected into three sections: proximal, medial and distal (Figure 6A, Table S1).”

Decision Letter, first revision:

13th March 2024

Dear Ferdinand,

Your revised manuscript entitled "The brittle star genome illuminates the genetic basis of animal appendage regeneration" has now been seen by the same three reviewers, whose comments are attached. You will see that while Reviewers 2 and 3 are satisfied with the revisions, the first reviewer still has concerns which will need to be addressed before we can offer publication in Nature Ecology & Evolution. We are particularly concerned with points 1 and 2 of their comments. We will therefore need to see your responses to the criticisms raised along with a revised manuscript, before we can reach a final decision regarding publication.

We therefore invite you to revise your manuscript taking into account all reviewer and editor comments. Please highlight all changes in the manuscript text file in Microsoft Word format.

* If you have not done so already please begin to revise your manuscript so that it conforms to our Article format instructions at <http://www.nature.com/natecolevol/info/final-submission>. Refer also to any guidelines provided in this letter.

[REDACTED]

Note: This URL links to your confidential home page and associated information about manuscripts

38you may have submitted, or that you are reviewing for us. If you wish to forward this email to co-authors, please delete the link to your homepage.

Nature Ecology & Evolution is committed to improving transparency in authorship. As part of our efforts in this direction, we are now requesting that all authors identified as 'corresponding author' on published papers create and link their Open Researcher and Contributor Identifier (ORCID) with their account on the Manuscript Tracking System (MTS), prior to acceptance. ORCID helps the scientific community achieve unambiguous attribution of all scholarly contributions. You can create and link your ORCID from the home page of the MTS by clicking on 'Modify my Springer Nature account'. For more information please visit www.springernature.com/orcid.

[REDACTED]

Reviewers' comments:

Reviewer #1 (Remarks to the Author):

1. A notable aspect of this study is the exceptional rate of genomic rearrangement within the brittle star genome, making the accuracy of the genome assembly paramount. Although the revised Figure S1 and its accompanying k-mer spectrum suggest the effective removal of heterozygous haplotypes, they fall short of confirming the structural integrity and accuracy of the assembly—a critical element for the study's validity. The methodology for estimating the targeted genome size at 3 Gb remains unclear, as do the measures employed to ensure the final assembly size of 1.57 Gb does not exceed the anticipated 1.33 Gb through over-assembly. Given that 3D-DNA and YAHS are scaffolders rather than assemblers, which dependent on both the fragmentation degree and precision of input contigs and subsequent manual curation via Hi-C map analysis tools such as Juicebox, the authors are urged to: 1) reassemble the contigs using an alternative assembler like NextDenovo, which shows better benchmarking than flye for nanopore reads and highly heterozygous genomes (Sun, Li et al. 2021), and 2) reapply scaffolding with the NextDenovo contig assembly, generating Hi-C heatmaps for both flye and NextDenovo assemblies under various curation conditions to compare different chromosomal configurations (e.g., 19-21 chromosomes).

Moreover, for such extensively rearranged genome, mere reliance on standard metrics such as BUSCO and N50 is insufficient. To decisively exclude mis-assembly, the author should perform genome evaluation and correction using tools such as Inspector, QUAST, CRAQ, and conduct whole-genome

39resequencing with independent samples to confirm the consistency of these rearrangements within the population.

2. The lack of new histological or functional data leaves the biological significance of the Hox gene rearrangements unclear. The authors also acknowledge that the role of Hox genes across echinoderm development remains largely unexplored. It would be prudent, therefore, to condense the Hox section and incorporate it into the preceding genomic rearrangement section as a specific instance.

Additionally, the rearrangements within the Hox gene still raise concerns about the choice of sperm as the input material, due to the unique hybrid haploid genomes produced by meiotic recombination in each sperm cell. In human, it only needs 90 sperm cells to construct the recombination map which is consistent with population-wide data (Wang, Fan et al. 2012). Sequencing a vast number of sperm cells introduces risk of mis-assembly due to the combination of heterozygous reads.

Although the authors listed 9 papers and 1 blog of echinoderm genome studies using sperm or gonad, studies No.3,4,7 used multiple tissues including muscle and the contig N50s of study No.1,2,4,5,6,7 are 55kb, 9kb, 190kb, 219kb, 11kb/18kb, 128kb, which are far from the current standard of a high-quality genome (contig N50 >1Mb) (Wang and Wang 2023). To add more evidence, the author should present a read coverage plot for the entire Hox gene and its ± 2 kb flanking regions to verify sufficient supporting reads at the rearrangement junctions.

3. The manuscript includes a section on tandem gene duplications linked to the brittle star's larval skeleton and bioluminescence, yet these duplications lack functional validation and relevance to the paper's focus on regeneration. It is suggested to omit this section.

4. Another stated objective of the manuscript is to determine whether regeneration represents an ancestral trait or a product of convergent evolution, as suggested in the Lines 90-96. However, the absence of innovative insights into this question significantly detracts from the paper's novelty. The assertion that the brittle star serves as a novel model for regeneration because of its lower rate of gene complement and gene expression evolution, is sort of paradoxical given the discovery of high rates of inter-chromosomal rearrangement and alterations in the highly conserved Hox gene within this species. Could the observed high rate of rearrangement also account for the limited number of conserved genes identified across the three examined species? The authors should consider including at least one additional conserved echinoderm species to elucidate the regeneration findings.

5. The claim of limited number of regeneration dataset appears peculiar. According to Figure 1 of their cited paper (Srivastava 2021), echinoderm regeneration is classified under "whole-body regeneration," indicating that the regenerative capacity and mechanisms should not be determined barely based on the site of injury. Therefore, the inclusion of both structural and whole-body regenerating organisms for a comprehensive comparative analysis is warranted to elucidate the evolutionary trajectory of regeneration. Numerous animal clades, including amphioxus (Liang, Rathnayake et al. 2019), earthworm (Shao, Wang et al. 2020), aceol worm (Gehrke, Neverett et al. 2019, Hulett, Kimura et al. 2023), hydra (Petersen, Hoger et al. 2015, Murad, Macias-Munoz et al. 2021) and recently, ctenophore (Mitchell, Edgar et al. 2024), have been the subjects of time-series transcriptome studies during regeneration. Notably, some of these studies incorporate single-cell and spatio-temporal transcriptome data, offering a profound understanding of the complex interplay among the molecular and cellular processes and define cell differentiation trajectories, which is also absent in current study.

40References

- Gehrke, A. R., E. Neverett, Y. J. Luo, A. Brandt, L. Ricci, R. E. Hulett, A. Gompers, J. G. Ruby, D. S. Rokhsar, P. W. Reddien and M. Srivastava (2019). "Acoel genome reveals the regulatory landscape of whole-body regeneration." *Science* 363(6432).
- Hulett, R. E., J. O. Kimura, D. M. Bolanos, Y. J. Luo, C. Rivera-Lopez, L. Ricci and M. Srivastava (2023). "Acoel single-cell atlas reveals expression dynamics and heterogeneity of adult pluripotent stem cells." *Nat Commun* 14(1): 2612.
- Liang, Y., D. Rathnayake, S. Huang, A. Pathirana, Q. Xu and S. Zhang (2019). "BMP signaling is required for amphioxus tail regeneration." *Development* 146(4).
- Mitchell, D. G., A. Edgar, J. R. Mateu, J. F. Ryan and M. Q. Martindale (2024). "The ctenophore *Mnemiopsis leidyi* deploys a rapid injury response dating back to the last common animal ancestor." *Commun Biol* 7(1): 203.
- Murad, R., A. Macias-Munoz, A. Wong, X. Ma and A. Mortazavi (2021). "Coordinated Gene Expression and Chromatin Regulation during Hydra Head Regeneration." *Genome Biol Evol* 13(12).
- Petersen, H. O., S. K. Hoger, M. Looso, T. Lengfeld, A. Kuhn, U. Warnken, C. Nishimiya-Fujisawa, M. Schnolzer, M. Kruger, S. Ozbek, O. Simakov and T. W. Holstein (2015). "A Comprehensive Transcriptomic and Proteomic Analysis of Hydra Head Regeneration." *Mol Biol Evol* 32(8): 1928-1947.
- Shao, Y., X. B. Wang, J. J. Zhang, M. L. Li, S. S. Wu, X. Y. Ma, X. Wang, H. F. Zhao, Y. Li, H. H. Zhu, D. M. Irwin, D. P. Wang, G. J. Zhang, J. Ruan and D. D. Wu (2020). "Genome and single-cell RNA-sequencing of the earthworm *Eisenia andrei* identifies cellular mechanisms underlying regeneration." *Nat Commun* 11(1): 2656.
- Srivastava, M. (2021). "Beyond Casual Resemblance: Rigorous Frameworks for Comparing Regeneration Across Species." *Annu Rev Cell Dev Biol* 37: 415-440.
- Sun, J., R. Li, C. Chen, J. D. Sigwart and K. M. Kocot (2021). "Benchmarking Oxford Nanopore read assemblers for high-quality molluscan genomes." *Philos Trans R Soc Lond B Biol Sci* 376(1825): 20200160.
- Wang, J., H. C. Fan, B. Behr and S. R. Quake (2012). "Genome-wide single-cell analysis of recombination activity and de novo mutation rates in human sperm." *Cell* 150(2): 402-412.
- Wang, P. and F. Wang (2023). "A proposed metric set for evaluation of genome assembly quality." *Trends Genet* 39(3): 175-186.

Reviewer #2 (Remarks to the Author):

The authors have address all major comments. The re-writing is appropriately executed. This work would be a great addition to the field.

Reviewer #3 (Remarks to the Author):

In the response to the referees and in the revised manuscript, the authors have addressed all of my concerns and modified the text accordingly. They now provide additional details in the methods and main text to clarify some key points that were previously confusing.

41I feel that all of my concerns have been adequately addressed and have no further comments.

*****END*****

Author Rebuttal, first revision:Reviewers' comments:

Reviewer #1 (Remarks to the Author):

1. A notable aspect of this study is the exceptional rate of genomic rearrangement within the brittle star genome, making the accuracy of the genome assembly paramount. Although the revised Figure S1 and its accompanying k-mer spectrum suggest the effective removal of heterozygous haplotypes, they fall short of confirming the structural integrity and accuracy of the assembly—a critical element for the study's validity. The methodology for estimating the targeted genome size at 3 Gb remains unclear, as do the measures employed to ensure the final assembly size of 1.57 Gb does not exceed the anticipated 1.33 Gb through over-assembly. Given that 3D-DNA and YAHS are scaffolders rather than assemblers, which dependent on both the fragmentation degree and precision of input contigs and subsequent manual curation via Hi-C map analysis tools such as Juicebox, the authors are urged to: 1) reassemble the contigs using an alternative assembler like NextDenovo, which shows better benchmarking than flye for nanopore reads and highly heterozygous genomes (Sun, Li et al. 2021), and 2) reapply scaffolding with the NextDenovo contig assembly, generating Hi-C heatmaps for both flye and NextDenovo assemblies under various curation conditions to compare different chromosomal configurations (e.g., 19-21 chromosomes). Moreover, for such extensively rearranged genome, mere reliance on standard metrics such as BUSCO and N50 is insufficient. To decisively exclude mis-assembly, the author should perform genome evaluation and correction using tools such as Inspector, QUASt, CRAQ, and conduct whole-genome resequencing with independent samples to confirm the consistency of these rearrangements within the population.

New controls

In line with the reviewer's requests, we performed new controls, including additional metrics computed with Merqury and Inspector and attempts to re-assemble the genome with alternative assemblers such as Raven and NextDenovo. Results demonstrate again the quality of our assembly, suitability of our methodology and robustness of our results.

We however reiterate that the rearrangements that we report are the products of inter-chromosomal fusions, fissions and translocations (macrosyntenic rearrangements), and these are inferred from HiC scaffolding. We did not investigate small-scale local rearrangements. We underline that it is only the inference of small-scale rearrangements that could be questioned by the possible presence of local contig assembly errors.

We detail below results of these new analyses and hope that they will meet the reviewer's expectations:

1. Merqury

Merqury evaluated k-mer completeness at 97% before haplotig purge, 67% after, which is consistent with the high heterozygosity and drop of kmers from heterozygous regions (Fig. S1). The Base Accuracy QV was 31.6 (0.000683556 error rate). This represents a very low error rate and is in line with values from recently reported high-quality chromosome-scale sea star genomes, for instance: QV=31.7 for *P. pectinifera* (Jung et al., 2023) and QV=36.3 for *P. borealis* (Lee et al., 2022).

2. Inspector

Similarly, Inspector reported a read-to-contig mapping rate of 97% before haplotig purge and 74% after, which is consistent with the high heterozygosity and correct collapse of haplotypes (Read Depth=53.431 before collapse and 92.1423 after, in line with our ~100X coverage). The reported

14

ESS
: IS

structural quality value QV was 26.88, which corresponds to a low error rate of 0.002. For comparison, the Inspector QV value has been reported for only one other echinoderm genome (as this metric is not current practice): the high-quality chromosome-scale genome of the sea cucumber *Chiridota heheva* had an only slightly higher QV of 29.7695 (Pu et al., 2024). **We thus further validate here that our assembly is of high quality, even at the small-scale structural level, which is not the level at which our synteny analyses are performed.**

We have integrated these new controls in the manuscript.

As a side note, while such validation tools provide an interesting reference-free evaluation of assembly quality, their usage remain relatively rare (for instance, to date, only < 15 studies have reported Inspector QV values, none have reported CRAQ statistics) makes it currently difficult to compare them across species, and to evaluate the impact of factors such as polymorphism, repeat content and sequencing technology on the metrics that they calculate. This complicates the evaluation of the performance of these software packages, which overall remain poorly characterised.

3. Re-assembly

We performed re-assembly from scratch using alternative methods: Raven, Canu and the NextDenovo assembler proposed by the reviewer.

Both Canu and Raven achieved poor results, suggesting they were not suitable to the high heterozygosity and repeat content of *Amphiura filiformis*. Specifically, Canu did not manage to yield any assembly, while Raven yielded a suboptimal assembly with a N50 contig of 290kb. NextDenovo generated an assembly with similar quality metrics than the Flye assembly that we report in this manuscript: after purging haplotypes, the total assembly size was of 1.60 Gb and N50 contig of 2.94 Mb. In comparison, our assembly has a size of 1.57 Gb and a better N50 contig of 3.2 Mb. **Scaffolding with HiC-data using YAHS revealed the same chromosomes, with a few exceptions that are clear scaffolding errors of the NextDenovo assembly,** as shown in the Figure below:

Comparison of our Flye+YAHS assembly and a re-assembly with NextDenovo + YAHS. **A.** Chromosome comparisons, using genes as anchors. The majority of inferred chromosomes are identical (shown in colours), with a few differences (grey). **B.** Chromosome comparisons based on alignments with minimap, showing a strong contiguity of the two assemblies. **C.** Contact map of our Flye genome assembly, demonstrating accurate HiC scaffolding, as outlined in the previous response and presented in Fig. S1. **D.** Contact map of the NextDenovo + YAHS assembly, presenting obvious errors that would need to be manually curated (grey arrows) and that are responsible for the discrepancy between the two assemblies (grey chromosomes).

These new analyses thus unambiguously support the high-quality of our genome assembly, and provide clear, independent support to our conclusions regarding chromosome evolution in echinoderms.

Suitability of our methodology

We again underscore the suitability of the genome assembly approaches we used, and express why we slightly disagree over some of the points made by the reviewer.

- We are confused as to the reason why the reviewer has put forward the thought that the structural integrity and accuracy of the assembly is not demonstrated, while all provided state-of-the-art metrics suggest otherwise. The reviewer does not present any argument here to explain why the methods we used or the presented data cast a doubt over the assembly. High contig N50 and BUSCO statistics demonstrate at the same time contiguity, non-fragmentation, accuracy and completeness. These statistics remain all the more relevant for repeat-rich, macrosyntetically rearranged genomes. For instance, the presence of extensive structural errors would be reflected as duplicated or fragmented BUSCO. Accordingly, **the benchmark cited by the reviewer** (Sun, Li et al. 2021) **relied on N50 and BUSCO metrics** to rank optimal assembly strategies for the considered two molluscan genomes, see for instance the (cautious) conclusion regarding aforementioned inadequacy of Flye: "However, **neither of these two Flye assemblies had N50 values over 500 Kb.** [...] it may indicate that the *M. coruscus* genome is too heterozygous or repetitive for Flye to be an effective assembler. [...] Among all the *M. coruscus* assemblies generated in our benchmarking, the NextDenovo version exhibited the highest NG50 (3.40 Mb) and BUSCO scores." (Sun, Li et al. 2021). The extrapolation that Flye would similarly not perform adequately on our species based on observations made on this one specific molluscan genome thus does not appear valid: **our Flye assembly has a N50 contig of 3.2 Mb (68.8 Mb after scaffolding) and 96% Complete BUSCO.**
- We estimated an haploid genome size of 1.33 Gb with GenomeScope and subsequently provided 3 Gb as a rough estimate for the expected resulting diploid genome size to Flye. We outline that this parameter is only a starting point for the algorithm and is even optional: the resulting Flye assembly is data-driven and not constrained by this estimate (<https://github.com/fenderglass/Flye/blob/flye/docs/FAQ.md>). Our final primary assembly contains 20 chromosomes of total size 1.47 Gb (1.57 Gb when all scaffolds are counted), highlighting very consistent estimations of genome size between Flye assembly and GenomeScope. K-mer plots further demonstrate that haplotigs were correctly purged and that they are not responsible for slight differences between these two numbers. Finally, our new Nextdenovo assembly yielded a similar genome size, providing independent validation. (In contrast, in the benchmark cited by the reviewer, the GenomeScope estimate is 1.593 Gb and the resulting NextDenovo assembly of 2.07 Gb.)
- We outline that 3D-DNA and YAHS do not involve manual curation as the reviewer suggests, but are automated tools to scaffold the genome into chromosomes from HiC data. Subsequent manual curation can be performed in case the contact map shows inadequate scaffolding, which is not the case here (see contact map, **Fig. S1**). It is therefore not clear to us how we can manually curate the HiC-map to present alternative solutions with 19 or 21 chromosomes, when it is clearly not supported by the data. We also highlight that we find **a substantial difference in the number of inter-chromosomal rearrangements in *A. filiformis* and the second most rearranged echinoderm (26 vs 5)**. This means that even in the case of an unlikely missed fission or fusion, *A. filiformis* would still be overwhelmingly more rearranged than currently available echinoderm genomes.
- Again, as stated in the previous round of review, we would like to stress that our conclusions regarding the rearranged nature of the ophiuroid lineage are relying on the **chromosomal reconstruction (linkage or macrosynteny), which is based on HiC scaffolding of**

47

ISS
: IS

contigs. We do not investigate the evolution of local gene order (microsynteny), as the closest available genomes are too distant to infer breakpoint locations. The only exception is the specific case of the Hox cluster (see additional controls below).

- We feel that some of the tools suggested by the reviewers are not the best suited to apply in the present case. For instance, one of the newly-suggested tools, **QUAST, reports exactly the same statistics as the ones we provided** (except in reference-based mode, which does not apply here). Another one, CRAQ, was only recently published and has never been used outside of the scope of the 2 genomes presented in the publication and as such does not qualify as a state-of-the-art, easily interpretable metric at the moment. Lastly, **NextDenovo is a promising novel assembler but is not published**, and still lacks some validation from the community as opposed to well-established and widely-used assemblers such as Flye, Raven and Canu.
- Finally, we feel that the last recommendations of the reviewer are highly unrealistic: resequencing a whole population and investigating structural variation is a completely different study, one that would be highly costly and beyond any golden rule practice defined by leading large-scale sequencing projects.

2. The lack of new histological or functional data leaves the biological significance of the Hox gene rearrangements unclear. The authors also acknowledge that the role of Hox genes across echinoderm development remains largely unexplored. It would be prudent, therefore, to condense the Hox section and incorporate it into the preceding genomic rearrangement section as a specific instance. Additionally, the rearrangements within the Hox gene still raise concerns about the choice of sperm as the input material, due to the unique hybrid haploid genomes produced by meiotic recombination in each sperm cell. In human, it only needs 90 sperm cells to construct the recombination map which is consistent with population-wide data (Wang, Fan et al. 2012). Sequencing a vast number of sperm cells introduces risk of mis-assembly due to the combination of heterozygous reads.

Although the authors listed 9 papers and 1 blog of echinoderm genome studies using sperm or gonad, studies No.3,4,7 used multiple tissues including muscle and the contig N50s of study No.1,2,4,5,6,7 are 55kb, 9kb, 190kb, 219kb, 11kb/18kb, 128kb, which are far from the current standard of a high-quality genome (contig N50 >1Mb) (Wang and Wang 2023). To add more evidence, the author should present a read coverage plot for the entire Hox gene and its ± 2 kb flanking regions to verify sufficient supporting reads at the rearrangement junctions.

New controls

Following the reviewer's recommendation, we present below the read coverage plot for the entire Hox cluster, which consistently shows elevated nanopore read coverage at rearrangement locations:

Nanopore reads coverage at the Hox cluster locus. Nanopore reads were mapped to the final assembly and filtered to retain only reads with high mapping quality (MAPQ > 30). **A.** Coverage is represented as the raw number of reads and the scale ranges from 0 to 130 reads. Red boxes outline breakpoints positions, where read coverage strongly supports the gene adjacencies: Hox1-Hox4 (left) and Hox7-Hox9/10-Hox8-Hox11/13a (right) display ~97.5 reads coverage, which represent high support within the context of our ~100X nanopore sequencing. Note that the Hox cluster locus was assembled as a single contig (CONTIG_20_18). **B.** Sample of ~30 reads at the gene adjacency Hox1-Hox4, showing strong alignment support. **C.** Same as B, but for the gene adjacencies Hox7-Hox9/10-Hox8-Hox11/13a.

We have incorporated this control in the Methods: “We moreover validated correct assembly at the Hox cluster locus by inspection of nanopore reads coverage.”.

Relevance of the results and suitability of our approach

In addition, we further justify the relevance of this result and soundness of our approaches below.

- The discovery of a rearranged Hox cluster in the brittle star *Amphiura filiformis* is of fundamental importance to the field of echinoderm genomics, which has been marked 18 years ago by the discovery of a rearranged Hox cluster in sea urchins (Cameron et al. 2006). Since then, this discovery has fuelled many hypotheses regarding the functional significance of these rearrangements and about the role of Hox genes more broadly (Byrne, Martinez, and Morris 2016; Lacalli 2014; David and Mooi 2014; Duboule 2007). The discovery of independent reorganisation in brittle stars is crucial to this discussion, as it suggests echinoderm genomes are permissive to such rearrangements. We thus retain this as one of the major findings of our work, and present it as such.
- We reiterate that potential issues related to recombination in sperm cells are hugely overstated by the reviewer, and do not appear to be supported by any specific study. In the human genome, only around 1 or 2 crossing over events occur per chromosome per sperm cell (22.8 ± 0.4 SE [± 3.7 SD] crossing-over in each sperm cell in the study cited by the reviewer (Wang, Fan et al. 2012)). The average crossing over size is of 500kb, thus yielding

only subtle genetic differences from parental alleles (500kb*23 = 11.5Mb, which accounts for 0.3% of the haploid human genome). This means that, for a given locus, parental alleles will overwhelmingly outnumber recombined alleles. The argument of the reviewer that 90 sperm cells are enough to build the recombination map only means that events are rare and localised enough that recombination spots can be comprehensively pinpointed from a low number of cells. For these reasons, and the ones explained in the response to point 1 above, sequencing a vast number of sperm cells does not introduce any increased risk of mis-assembly and does not result in broadly more heterozygous reads than would somatic tissues.

- As a further validation of the use of sperm in this context, **Hox cluster organisation in sea urchins was initially determined by both genome sequencing and BAC mapping from three distinct sperm samples** (Cameron et al. 2006), **all yielding exactly the same results**. The same configuration was also corroborated years later by whole genome sequencing of other sea urchin genomes (for instance see (Davidson et al. 2020)). We reiterate that **DNA extraction from sperm is the recommended protocol for genome assembly in echinoderms**, especially for species such as sea urchins or brittle stars displaying high calcification. This is thoroughly **documented in “DNA Extraction Protocols for Whole-Genome Sequencing in Marine Organisms”** by (Panova et al. 2016), with even specific protocol optimisation for *Amphiura filiformis*: “Following the protocol routinely used for genomic DNA extraction from echinoderm sperm of many different species including that of the sea urchin *Strongylocentrotus purpuratus*, DNA was successfully extracted from *A. filiformis* sperm. [...] This protocol provided around 130 µg of genomic DNA at a concentration of 500–600 ng/µL and absorbance ratios 1.89 and 2.3 at 260/280 and 260/230 nm, respectively. The DNA had a high-molecular weight, as required by NGS service providers.”
- Out of the cited papers that used sperm as starting material, we did explicitly point out that 3,4 and 7 used other tissues, so this is not a misrepresentation of the data from our side. Low N50 contig statistics (which, again is not the case in our study) from these studies are **affected by the fact that they did not use long-read** (or low coverage long-reads only, to guide short reads), **not by the choice of starting material**. Accordingly, (Wang and Wang 2023) never suggested that a good genome assembly should have > 1Mb contig N50, only that “Assembly of reads generated by long-read sequencing platforms enables contig N50 to easily surpass 1 Mb or even 10 Mb for an assembly.” (Wang and Wang 2023). (We also note that, by the reviewer’s standards, our assembly would then be a “good genome” because our N50 contig = 3.2 Mb. This ties in with the lack of specific metrics that the reviewer requires us to provide to validate our assembly). Of note, several of these studies still achieved chromosome-level assembly, thanks to the robustness of HiC scaffolding.
- Finally, the reviewer cannot completely disregard echinoderm genomes assembled from sperm or gonadal tissue and based on long-read and HiC data (Number 8,9,10) and their importance for the community. Notably, these include the blog post from the Wellcome Sanger Institute, which describes a **high-quality sea star genome generated from sperm DNA available in the leading database Ensembl Metazoa and generated in the context of the Darwin Tree of Life project**.

3. The manuscript includes a section on tandem gene duplications linked to the brittle star's larval skeleton and bioluminescence, yet these duplications lack functional validation and relevance to the paper's focus on regeneration. It is suggested to omit this section.

50

ESS
: is

This section is important to the fields of echinoderm genomics and ecology, as these are historically widely studied gene families and fundamental examples to study the evolution of novelties (Coubris et al. 2024; Koga, Morino, and Wada 2014), for which we provide a necessary evolutionary context and avenues for future investigations.

4. Another stated objective of the manuscript is to determine whether regeneration represents an ancestral trait or a product of convergent evolution, as suggested in the Lines 90-96. However, the absence of innovative insights into this question significantly detracts from the paper's novelty. The assertion that the brittle star serves as a novel model for regeneration because of its lower rate of gene complement and gene expression evolution, is sort of paradoxical given the discovery of high rates of inter-chromosomal rearrangement and alterations in the highly conserved Hox gene within this species. Could the observed high rate of rearrangement also account for the limited number of conserved genes identified across the three examined species? The authors should consider including at least one additional conserved echinoderm species to elucidate the regeneration findings.

Gene duplication and gene expression evolution are largely independent from macrosynteny evolution, hence we are confused why the reviewer would suggest that our results are paradoxical. Our statements are also relative to axolotl (a vertebrate who went through two vertebrate-specific WGDs events and associated gene complement and gene sequence and expression evolution) and Parhyale (an ecdysozoan, a lineage characterised by extensive gene losses), as is explained in the manuscript. The Hox cluster organisation is irrelevant in this context. Again, the limited number of genes with conserved temporal expression dynamics concerns comparisons between the axolotl and Parhyale more than comparisons with the brittle star (as explained in detail in our previous response). The final point about adding "one additional conserved echinoderm species" is also confusing: it is unclear what a "conserved" echinoderm species means and there are no currently available comparable datasets in this group.

Here, we conduct the first comparative analysis of temporal gene expression dynamics across vertebrate and invertebrate species, using standard comparative methodologies from the evo-devo field and reveal that proliferation is the most conserved phase of regeneration. No previous study has conducted similar investigations before or has reported similar findings, we do not understand the reviewer's claim of "absence of innovative insights".

5. The claim of limited number of regeneration dataset appears peculiar. According to Figure 1 of their cited paper (Srivastava 2021), echinoderm regeneration is classified under "whole-body regeneration," indicating that the regenerative capacity and mechanisms should not be determined barely based on the site of injury. Therefore, the inclusion of both structural and whole-body regenerating organisms for a comprehensive comparative analysis is warranted to elucidate the evolutionary trajectory of regeneration. Numerous animal clades, including amphioxus (Liang, Rathnayake et al. 2019), earthworm (Shao, Wang et al. 2020), aceol worm (Gehrke, Neverett et al. 2019, Hulett, Kimura et al. 2023), hydra (Petersen, Hoyer et al. 2015, Murad, Macias-Munoz et al. 2021) and recently, ctenophore (Mitchell, Edgar et al. 2024), have been the subjects of time-series transcriptome studies during regeneration. Notably, some of these studies incorporate single-cell and spatio-temporal transcriptome data, offering a profound understanding of the complex interplay among the molecular and cellular processes and define cell differentiation trajectories, which is also absent in current study.

We are studying the **evolution** of the **appendage regeneration** gene expression programme, which is biologically different from whole-body regeneration and hence all suggested datasets are irrelevant to our analyses (as detailed in our previous response). **While a handful of sea star species can undergo (larval) whole-body regeneration, this is not the case for the brittle star *Amphiura***

51

ESS
: is

filiformis, thus it obviously is not and cannot be the focus of our analysis. We feel that the suggestion that we determined *Amphiura filiformis* regeneration ability "barely based on the site of injury" is diminishing towards the work of several of the authors of this paper who produced numerous studies on regeneration in *Amphiura filiformis* over the past ten years (Czarkwiani, Dylus, and Oliveri 2013; Hu et al. 2014; Purushothaman et al. 2015; Czarkwiani et al. 2016; Ferrario et al. 2018; Piovani et al. 2021; Czarkwiani et al. 2021). Lastly, the idea that cellular insights regarding whole-body regeneration available in other species would compromise the novelty of our comparative investigations is misconstrued, since the aims of the cited studies are evidently distinct from ours.

References

- Gehrke, A. R., E. Neverett, Y. J. Luo, A. Brandt, L. Ricci, R. E. Hulett, A. Gompers, J. G. Ruby, D. S. Rokhsar, P. W. Reddien and M. Srivastava (2019). "Acoel genome reveals the regulatory landscape of whole-body regeneration." *Science* 363(6432).
- Hulett, R. E., J. O. Kimura, D. M. Bolanos, Y. J. Luo, C. Rivera-Lopez, L. Ricci and M. Srivastava (2023). "Acoel single-cell atlas reveals expression dynamics and heterogeneity of adult pluripotent stem cells." *Nat Commun* 14(1): 2612.
- Liang, Y., D. Rathnayake, S. Huang, A. Pathirana, Q. Xu and S. Zhang (2019). "BMP signaling is required for amphioxus tail regeneration." *Development* 146(4).
- Mitchell, D. G., A. Edgar, J. R. Mateu, J. F. Ryan and M. Q. Martindale (2024). "The ctenophore *Mnemiopsis leidyi* deploys a rapid injury response dating back to the last common animal ancestor." *Commun Biol* 7(1): 203.
- Murad, R., A. Macias-Munoz, A. Wong, X. Ma and A. Mortazavi (2021). "Coordinated Gene Expression and Chromatin Regulation during Hydra Head Regeneration." *Genome Biol Evol* 13(12).
- Petersen, H. O., S. K. Hoger, M. Looso, T. Lengfeld, A. Kuhn, U. Warnken, C. Nishimiya-Fujisawa, M. Schnolzer, M. Kruger, S. Ozbek, O. Simakov and T. W. Holstein (2015). "A Comprehensive Transcriptomic and Proteomic Analysis of Hydra Head Regeneration." *Mol Biol Evol* 32(8): 1928-1947.
- Shao, Y., X. B. Wang, J. J. Zhang, M. L. Li, S. S. Wu, X. Y. Ma, X. Wang, H. F. Zhao, Y. Li, H. H. Zhu, D. M. Irwin, D. P. Wang, G. J. Zhang, J. Ruan and D. D. Wu (2020). "Genome and single-cell RNA-sequencing of the earthworm *Eisenia andrei* identifies cellular mechanisms underlying regeneration." *Nat Commun* 11(1): 2656.
- Srivastava, M. (2021). "Beyond Casual Resemblance: Rigorous Frameworks for Comparing Regeneration Across Species." *Annu Rev Cell Dev Biol* 37: 415-440.
- Sun, J., R. Li, C. Chen, J. D. Sigwart and K. M. Kocot (2021). "Benchmarking Oxford Nanopore read assemblers for high-quality molluscan genomes." *Philos Trans R Soc Lond B Biol Sci* 376(1825): 20200160.
- Wang, J., H. C. Fan, B. Behr and S. R. Quake (2012). "Genome-wide single-cell analysis of recombination activity and de novo mutation rates in human sperm." *Cell* 150(2): 402-412.
- Wang, P. and F. Wang (2023). "A proposed metric set for evaluation of genome assembly quality." *Trends Genet* 39(3): 175-186.

References

- Byrne, Maria, Pedro Martinez, and Valerie Morris. 2016. "Evolution of a Pentamer Body Plan Was Not Linked to Translocation of Anterior Hox Genes: The Echinoderm HOX Cluster Revisited." *Evolution & Development* 18 (2): 137–43.
- Cameron, R. Andrew, Lee Rowen, Ryan Nesbitt, Scott Bloom, Jonathan P. Rast, Kevin Berney, Cesar Arenas-Mena, et al. 2006. "Unusual Gene Order and Organization of the Sea Urchin Hox Cluster." *Journal of Experimental Zoology. Part B, Molecular and Developmental Evolution* 306 (1): 45–58.
- Coubris, Constance, Laurent Duchatelet, Sam Dupont, and Jérôme Mallefet. 2024. "A Brittle Star Is Born: Ontogeny of Luminous Capabilities in *Amphiura Filiformis*." *PloS One* 19 (3): e0298185.
- Czarkwiani, Anna, David V. Dylus, Luisana Carballo, and Paola Oliveri. 2021. "FGF Signalling Plays

- Similar Roles in Development and Regeneration of the Skeleton in the Brittle Star *Amphiura Filiformis*." *Development* 148 (10). <https://doi.org/10.1242/dev.180760>.
- Czarkwiani, Anna, David V. Dylus, and Paola Oliveri. 2013. "Expression of Skeletogenic Genes during Arm Regeneration in the Brittle Star *Amphiura Filiformis*." *Gene Expression Patterns: GEP* 13 (8): 464–72.
- Czarkwiani, Anna, Cinzia Ferrario, David Viktor Dylus, Michela Sugni, and Paola Oliveri. 2016. "Skeletal Regeneration in the Brittle Star *Amphiura Filiformis*." *Frontiers in Zoology* 13 (April): 18.
- David, Bruno, and Rich Mooi. 2014. "How Hox Genes Can Shed Light on the Place of Echinoderms among the Deuterostomes." *EvoDevo* 5 (June): 22.
- Davidson, Phillip L., Haobing Guo, Lingyu Wang, Alejandro Berrio, He Zhang, Yue Chang, Andrew L. Soborowski, David R. McClay, Guangyi Fan, and Gregory A. Wray. 2020. "Chromosomal-Level Genome Assembly of the Sea Urchin *Lytechinus Variegatus* Substantially Improves Functional Genomic Analyses." *Genome Biology and Evolution* 12 (7): 1080–86.
- Duboule, Denis. 2007. "The Rise and Fall of Hox Gene Clusters." *Development* 134 (14): 2549–60.
- Ferrario, Cinzia, Yousra Ben Khadra, Anna Czarkwiani, Anne Zakrzewski, Pedro Martinez, Graziano Colombo, Francesco Bonasoro, Maria Daniela Candia Carnevali, Paola Oliveri, and Michela Sugni. 2018. "Fundamental Aspects of Arm Repair Phase in Two Echinoderm Models." *Developmental Biology* 433 (2): 297–309.
- Hu, Marian Y., Isabel Casties, Meike Stumpp, Olga Ortega-Martinez, and Sam Dupont. 2014. "Energy Metabolism and Regeneration Are Impaired by Seawater Acidification in the Infaunal Brittlestar *Amphiura Filiformis*." *The Journal of Experimental Biology* 217 (Pt 13): 2411–21.
- Jung, Jaehoon, So Yun Jhang, Bongsang Kim, Bomim Koh, Chaeyoung Ban, Hyojung Seo, Taeseo Park, et al. 2023. "The First High-Quality Genome Assembly and Annotation of *Patiria Pectinifera*." *Scientific Data* 10 (1): 642.
- Koga, Hiroyuki, Yoshiaki Morino, and Hiroshi Wada. 2014. "The Echinoderm Larval Skeleton as a Possible Model System for Experimental Evolutionary Biology." *Genesis* 52 (3): 186–92.
- Lacalli, Thurston. 2014. "Echinoderm Conundrums: Hox Genes, Heterochrony, and an Excess of Mouths." *EvoDevo* 5 (1): 46.
- Lee, Yujung, Bongsang Kim, Jaehoon Jung, Bomim Koh, So Yun Jhang, Chaeyoung Ban, Won-Jae Chi, Soonok Kim, and Jaewoong Yu. 2022. "Chromosome-Level Genome Assembly of *Plazaster borealis* Sheds Light on the Morphogenesis of Multiarmed Starfish and Its Regenerative Capacity." *GigaScience* 11 (July). <https://doi.org/10.1093/gigascience/giac063>.
- Panova, Marina, Henrik Aronsson, R. Andrew Cameron, Peter Dahl, Anna Godhe, Ulrika Lind, Olga Ortega-Martinez, et al. 2016. "DNA Extraction Protocols for Whole-Genome Sequencing in Marine Organisms." In *Marine Genomics: Methods and Protocols*, edited by Sarah J. Bourlat, 13–44. New York, NY: Springer New York.
- Piovani, Laura, Anna Czarkwiani, Cinzia Ferrario, Michela Sugni, and Paola Oliveri. 2021. "Ultrastructural and Molecular Analysis of the Origin and Differentiation of Cells Mediating Brittle Star Skeletal Regeneration." *BMC Biology* 19 (1): 9.
- Pu, Yujin, Yang Zhou, Jun Liu, and Haibin Zhang. 2024. "A High-Quality Chromosomal Genome Assembly of the Sea Cucumber *Chiridota Heheva* and Its Hydrothermal Adaptation." *GigaScience* 13 (January). <https://doi.org/10.1093/gigascience/giad107>.
- Purushothaman, Sruthi, Sandeep Saxena, Vuppapalaty Meghah, Cherukuvada V. Brahmendra Swamy, Olga Ortega-Martinez, Sam Dupont, and Mohammed Idris. 2015. "Transcriptomic and Proteomic Analyses of *Amphiura Filiformis* Arm Tissue-Undergoing Regeneration." *Journal of Proteomics* 112 (January): 113–24.
- Sun, Jin, Runsheng Li, Chong Chen, Julia D. Sigwart, and Kevin M. Kocot. 2021. "Benchmarking Oxford Nanopore Read Assemblers for High-Quality Molluscan Genomes." *Philosophical Transactions of the Royal Society of London. Series B, Biological Sciences* 376 (1825): 20200160.
- Wang, Jianbin, H. Christina Fan, Barry Behr, and Stephen R. Quake. 2012. "Genome-Wide Single-Cell Analysis of Recombination Activity and de Novo Mutation Rates in Human Sperm." *Cell* 150 (2): 402–12.
- Wang, Peng, and Fei Wang. 2023. "A Proposed Metric Set for Evaluation of Genome Assembly Quality." *Trends in Genetics: TIG* 39 (3): 175–86.

Reviewer #2 (Remarks to the Author):

The authors have address all major comments. The re-writing is appropriately executed. This work would be a great addition to the field.

We sincerely thank the reviewer for their constructive and supportive feedback which resulted in a significantly improved manuscript.

Reviewer #3 (Remarks to the Author):

In the response to the referees and in the revised manuscript, the authors have addressed all of my concerns and modified the text accordingly. They now provide additional details in the methods and main text to clarify some key points that were previously confusing.

I feel that all of my concerns have been adequately addressed and have no further comments.

We sincerely thank the reviewer for their constructive and supportive feedback which resulted in a significantly improved manuscript.

Decision Letter, second revision:

23rd April 2024

Dear Ferdinand,

Thank you for submitting your revised manuscript "The brittle star genome illuminates the genetic basis of animal appendage regeneration" (NATECOLEVOL-23112660B). It has now been seen again by Reviewer #1 and their comments are below. The reviewer is not supportive of publication but they have no remaining specific technical issues and we'll be happy in principle to publish it in Nature Ecology & Evolution, pending minor revisions to comply with our editorial and formatting guidelines.

You will need to reduce the length of your manuscript to no more than 5000 words (this is already much longer than our typical length of 3500 and should be considered an absolute limit). Our preference is to remove the section "Tandem duplications of key genes likely contribute to brittle star larval skeleton and bioluminescence", which is not very insightful and its removal would help streamline the paper.

Please email us a copy of the file in an editable format (Microsoft Word or LaTeX)-- we can not proceed with PDFs at this stage.

[REDACTED]

Reviewer #1 (Remarks to the Author):

First, I appreciate the author's serious attention to the review comments and their diligent efforts to address each point raised. However, I am a little surprised that the responses appear overly defensive, seemingly more concerned with formally refuting the reviewer. It is important to remember that each responsible reviewer spends extra time and effort reading the manuscript, reviewing literatures, and providing their feedback, with no intention to impede the manuscript's publication but to enhance the robustness of the findings and the solidity of the conclusions. The essence of science lies in its falsifiability and any high-quality work must withstand scrutiny from various perspectives. This rigorous peer review system and the critical thinking maintain in pursuit of truth are crucial supports for the healthy development of the scientific field. This rigor also upholds Nature Ecology & Evolution's high standards of "publishing the best research from across ecology and evolutionary biology".

55For studies reliant on genomic data, the quality of the genome is paramount. The materials, methods, and results must undergo rigorous scrutiny. Advanced algorithms and sequencing technologies do not guarantee high-quality genomes, as shown by the unsatisfactory results from state-of-the-art tools like Raven, Canu, and NextDenovo used by the authors. This issue also exists in published articles, such as in the Wellcome Sanger Institute's 25 Genomes for 25 Years project, where the genome of the king scallop showed good N50 and BUSCO scores but was incorrectly annotated with over 60,000 genes (Kenny, McCarthy et al. 2020). This 'gene-rich' erroneous conclusion was widely cited in the field, causing considerable controversy. It was not until 2022 that another team reassembled the genome and corrected the gene count to 26,995 (Zeng, Liu et al. 2021), after which the revised genome was accepted by prominent scholars for comparative genomic studies in journals such as Nature (Martin-Zamora, Liang et al. 2023) and PNAS (Zhang, Yanez-Guerra et al. 2022). In this study, a significant number of genomic rearrangements and inconsistent chromosome numbers are reported. Given Nature Ecology & Evolution's reputation and broad influence, it is natural for reviewers to propose cautious genomic quality control suggestions to minimize risks and potential subsequent impacts.

The focus of the manuscript should remain on the brittle star's regeneration mechanism. Sections on Hox and tandem gene duplications lack functional experiments and are not significantly related to the main topic of regeneration, which impedes the narrative flow. Less is more, I sincerely suggest that reducing these sections would enhance the focus and quality of the overall manuscript. Of course, I would fully respect the editor's decision on this matter.

It is confusing that the author claims the novelty of this article lies in "conducting the first comparative analysis of temporal gene expression dynamics across vertebrate and invertebrate species, using standard comparative methodologies from the evo-devo field, and revealing that proliferation is the most conserved phase of regeneration. No previous study has conducted similar investigations before or has reported similar findings...". Firstly, the author needs to consider whether the universality of evolutionary patterns derived from comparisons among just three species is broad enough. Secondly, as proliferation represents an undifferentiated cellular stage in the regeneration process, finding similar gene repertoire across species seems not highly unexpected, and it has been reported and discussed as such in many studies (Somorjai, Somorjai et al. 2012, Ricci and Srivastava 2018, Daponte, Tylzanowski et al. 2021). If I understand correctly, Nature Ecology & Evolution is seeking articles that achieve conceptual breakthroughs in ecology and evolutionary biology. Unfortunately, I am not convinced what the author claims necessarily meet this requirement. If I have overlooked any key points, I hope the author could explain their contributions in more detail for a broader audience.

References

1. Daponte, V., P. Tylzanowski and A. Forlino. (2021) Appendage regeneration in vertebrates: What makes this possible? *Cells* 10(2): 242.
2. Kenny, N. J., S. A. McCarthy, O. Dudchenko, K. James, E. Betteridge, C. Corton, J. Dolucan, D. Mead, K. Oliver, A. D. Omer, S. Pelan, Y. Ryan, Y. Sims, J. Skelton, M. Smith, J. Torrance, D. Weisz, A. Wipat, E. L. Aiden, K. Howe and S. T. Williams. (2020) The gene-rich genome of the scallop *Pecten maximus*. *Gigascience* 9(5): gaaa037.
3. Martin-Zamora, F. M., Y. Liang, K. Guynes, A. M. Carrillo-Baltodano, B. E. Davies, R. D. Donnellan,

56Y. Tan, G. Moggioli, O. Seudre, M. Tran, K. Mortimer, N. M. Luscombe, A. Hejnal, F. Marletaz and J. M. Martin-Duran. (2023) Annelid functional genomics reveal the origins of bilaterian life cycles. *Nature* 615(7950): 105-110.

4. Ricci, L. and M. Srivastava. (2018) Wound-induced cell proliferation during animal regeneration. *Wiley Interdiscip Rev Dev Biol* 7(5): e321.

5. Somorjai, I. M., R. L. Somorjai, J. Garcia-Fernandez and H. Escriva. (2012) Vertebrate-like regeneration in the invertebrate chordate amphioxus. *Proc Natl Acad Sci USA* 109(2): 517-522.

6. Zeng, Q., J. Liu, C. Wang, H. Wang, L. Zhang, J. Hu, L. Bao and S. Wang. (2021) High-quality reannotation of the king scallop genome reveals no 'gene-rich' feature and evolution of toxin resistance. *Comput Struct Biotechnol J* 19: 4954-4960.

7. Zhang, Y., L. A. Yanez-Guerra, A. B. Tinoco, N. Escudero Castelan, M. Egertova and M. R. Elphick. (2022) Somatostatin-type and allatostatin-C-type neuropeptides are paralogous and have opposing myoregulatory roles in an echinoderm. *Proc Natl Acad Sci USA* 119(7): e2113589119.

Our ref: NATECOLEVOL-23112660B

26th April 2024

Dear Dr. Marlétaz,

Thank you for your patience as we've prepared the guidelines for final submission of your *Nature Ecology & Evolution* manuscript, "The brittle star genome illuminates the genetic basis of animal appendage regeneration" (NATECOLEVOL-23112660B). Please carefully follow the step-by-step instructions provided in the attached file, and add a response in each row of the table to indicate the changes that you have made. Please also check and comment on any additional marked-up edits we have proposed within the text. Ensuring that each point is addressed will help to ensure that your revised manuscript can be swiftly handed over to our production team.

****We would like to start working on your revised paper, with all of the requested files and forms, as soon as possible (preferably within two weeks). Please get in contact with us immediately if you anticipate it taking more than two weeks to submit these revised files.****

If you have not done so already, please alert us to any related manuscripts from your group that are under consideration or in press at other journals, or are being written up for submission to other

57journals (see: <https://www.nature.com/nature-research/editorial-policies/plagiarism#policy-on-duplicate-publication> for details).

In recognition of the time and expertise our reviewers provide to Nature Ecology & Evolution's editorial process, we would like to formally acknowledge their contribution to the external peer review of your manuscript entitled "The brittle star genome illuminates the genetic basis of animal appendage regeneration". For those reviewers who give their assent, we will be publishing their names alongside the published article.

Nature Ecology & Evolution offers a Transparent Peer Review option for new original research manuscripts submitted after December 1st, 2019. As part of this initiative, we encourage our authors to support increased transparency into the peer review process by agreeing to have the reviewer comments, author rebuttal letters, and editorial decision letters published as a Supplementary item. When you submit your final files please clearly state in your cover letter whether or not you would like to participate in this initiative. Please note that failure to state your preference will result in delays in accepting your manuscript for publication.

Cover suggestions

We welcome submissions of artwork for consideration for our cover. For more information, please see our guide for cover artwork.

Nature Ecology & Evolution has now transitioned to a unified Rights Collection system which will allow our Author Services team to quickly and easily collect the rights and permissions required to publish your work. Approximately 10 days after your paper is formally accepted, you will receive an email in providing you with a link to complete the grant of rights. If your paper is eligible for Open Access, our Author Services team will also be in touch regarding any additional information that may be required to arrange payment for your article.

Please note that *Nature Ecology & Evolution* is a Transformative Journal (TJ). Authors may publish their research with us through the traditional subscription access route or make their paper immediately open access through payment of an article-processing charge (APC). Authors will not be required to make a final decision about access to their article until it has been accepted. Find out more about Transformative Journals

Authors may need to take specific actions to achieve compliance with funder and institutional open access mandates. If your research is supported by a funder that requires immediate open access (e.g. according to Plan S principles) then you should select the gold OA route,

58and we will direct you to the compliant route where possible. For authors selecting the subscription publication route, the journal's standard licensing terms will need to be accepted, including <https://www.nature.com/nature-portfolio/editorial-policies/self-archiving-and-license-to-publish>. Those licensing terms will supersede any other terms that the author or any third party may assert apply to any version of the manuscript.

[REDACTED]

[REDACTED]

Reviewer #1:

Remarks to the Author:

First, I appreciate the author's serious attention to the review comments and their diligent efforts to address each point raised. However, I am a little surprised that the responses appear overly defensive, seemingly more concerned with formally refuting the reviewer. It is important to remember that each responsible reviewer spends extra time and effort reading the manuscript, reviewing literatures, and providing their feedback, with no intention to impede the manuscript's publication but to enhance the robustness of the findings and the solidity of the conclusions. The essence of science lies in its falsifiability and any high-quality work must withstand scrutiny from various perspectives. This rigorous peer review system and the critical thinking maintain in pursuit of truth are crucial supports for the healthy development of the scientific field. This rigor also upholds Nature Ecology & Evolution's high standards of "publishing the best research from across ecology and evolutionary biology".

For studies reliant on genomic data, the quality of the genome is paramount. The materials, methods, and results must undergo rigorous scrutiny. Advanced algorithms and sequencing technologies do not guarantee high-quality genomes, as shown by the unsatisfactory results from state-of-the-art tools like Raven, Canu, and NextDenovo used by the authors. This issue also exists in published articles, such as in the Wellcome Sanger Institute's 25 Genomes for 25 Years project, where the genome of the king scallop showed good N50 and BUSCO scores but was incorrectly annotated with over 60,000 genes (Kenny, McCarthy et al. 2020). This 'gene-rich' erroneous conclusion was widely cited in the field, causing considerable controversy. It was not until 2022 that another team reassembled the genome and corrected the gene count to 26,995 (Zeng, Liu et al. 2021), after which the revised genome was accepted by prominent scholars for comparative genomic studies in journals such as

59Nature (Martin-Zamora, Liang et al. 2023) and PNAS (Zhang, Yanez-Guerra et al. 2022). In this study, a significant number of genomic rearrangements and inconsistent chromosome numbers are reported. Given Nature Ecology & Evolution's reputation and broad influence, it is natural for reviewers to propose cautious genomic quality control suggestions to minimize risks and potential subsequent impacts.

The focus of the manuscript should remain on the brittle star's regeneration mechanism. Sections on Hox and tandem gene duplications lack functional experiments and are not significantly related to the main topic of regeneration, which impedes the narrative flow. Less is more, I sincerely suggest that reducing these sections would enhance the focus and quality of the overall manuscript. Of course, I would fully respect the editor's decision on this matter.

It is confusing that the author claims the novelty of this article lies in "conducting the first comparative analysis of temporal gene expression dynamics across vertebrate and invertebrate species, using standard comparative methodologies from the evo-devo field, and revealing that proliferation is the most conserved phase of regeneration. No previous study has conducted similar investigations before or has reported similar findings...". Firstly, the author needs to consider whether the universality of evolutionary patterns derived from comparisons among just three species is broad enough. Secondly, as proliferation represents an undifferentiated cellular stage in the regeneration process, finding similar gene repertoire across species seems not highly unexpected, and it has been reported and discussed as such in many studies (Somorjai, Somorjai et al. 2012, Ricci and Srivastava 2018, Daponte, Tylzanowski et al. 2021). If I understand correctly, Nature Ecology & Evolution is seeking articles that achieve conceptual breakthroughs in ecology and evolutionary biology. Unfortunately, I am not convinced what the author claims necessarily meet this requirement. If I have overlooked any key points, I hope the author could explain their contributions in more detail for a broader audience.

References

1. Daponte, V., P. Tylzanowski and A. Forlino. (2021) Appendage regeneration in vertebrates: What makes this possible? *Cells* 10(2): 242.
2. Kenny, N. J., S. A. McCarthy, O. Dudchenko, K. James, E. Betteridge, C. Corton, J. Dolucan, D. Mead, K. Oliver, A. D. Omer, S. Pelan, Y. Ryan, Y. Sims, J. Skelton, M. Smith, J. Torrance, D. Weisz, A. Wipat, E. L. Aiden, K. Howe and S. T. Williams. (2020) The gene-rich genome of the scallop *Pecten maximus*. *Gigascience* 9(5): giaa037.
3. Martin-Zamora, F. M., Y. Liang, K. Guynes, A. M. Carrillo-Baltodano, B. E. Davies, R. D. Donnellan, Y. Tan, G. Moggioni, O. Seudre, M. Tran, K. Mortimer, N. M. Luscombe, A. Hejnal, F. Marletaz and J. M. Martin-Duran. (2023) Annelid functional genomics reveal the origins of bilaterian life cycles. *Nature* 615(7950): 105-110.
4. Ricci, L. and M. Srivastava. (2018) Wound-induced cell proliferation during animal regeneration. *Wiley Interdiscip Rev Dev Biol* 7(5): e321.
5. Somorjai, I. M., R. L. Somorjai, J. Garcia-Fernandez and H. Escriva. (2012) Vertebrate-like regeneration in the invertebrate chordate amphioxus. *Proc Natl Acad Sci USA* 109(2): 517-522.
6. Zeng, Q., J. Liu, C. Wang, H. Wang, L. Zhang, J. Hu, L. Bao and S. Wang. (2021) High-quality reannotation of the king scallop genome reveals no 'gene-rich' feature and evolution of toxin resistance. *Comput Struct Biotechnol J* 19: 4954-4960.
7. Zhang, Y., L. A. Yanez-Guerra, A. B. Tinoco, N. Escudero Castelan, M. Egertova and M. R. Elphick.

60(2022) Somatostatin-type and allatostatin-C-type neuropeptides are paralogous and have opposing myoregulatory roles in an echinoderm. *Proc Natl Acad Sci USA* 119(7): e2113589119.

Final Decision Letter:

29th May 2024

Dear Dr Marlétaz,

I'm writing in the temporary absence of my colleague Vera Domingues. We are pleased to inform you that your Article entitled "The brittle star genome illuminates the genetic basis of animal appendage regeneration", has now been accepted for publication in *Nature Ecology & Evolution*.

Over the next few weeks, your paper will be copyedited to ensure that it conforms to *Nature Ecology and Evolution* style. Once your paper is typeset, you will receive an email with a link to choose the appropriate publishing options for your paper and our Author Services team will be in touch regarding any additional information that may be required

Due to the importance of these deadlines, we ask you please us know now whether you will be difficult to contact over the next month. If this is the case, we ask you provide us with the contact information (email, phone and fax) of someone who will be able to check the proofs on your behalf, and who will be available to address any last-minute problems . Once your paper has been scheduled for online publication, the Nature press office will be in touch to confirm the details.

Acceptance of your manuscript is conditional on all authors' agreement with our publication policies (see www.nature.com/authors/policies/index.html). In particular your manuscript must not be published elsewhere and there must be no announcement of the work to any media outlet until the publication date (the day on which it is uploaded onto our web site).

Please note that *Nature Ecology & Evolution* is a Transformative Journal (TJ). Authors may publish their research with us through the traditional subscription access route or make their paper immediately open access through payment of an article-processing charge (APC). Authors will not be required to make a final decision about access to their article until it has been accepted. Find out more about Transformative Journals

Authors may need to take specific actions to achieve compliance with funder and institutional open access mandates. If your research is supported by a funder that requires immediate open access (e.g. according to Plan S principles) then you should select the gold OA route, and we will direct you to the compliant route where possible. For authors selecting the subscription publication route, the journal's standard licensing terms will need to be accepted, including . All co-authors, authors' institutions and authors' funding agencies can order reprints using the form appropriate to their geographical region.

We welcome the submission of potential cover material (including a short caption of around 40 words) related to your manuscript; suggestions should be sent to Nature Ecology & Evolution as electronic files (the image should be 300 dpi at 210 x 297 mm in either TIFF or JPEG format). Please note that such pictures should be selected more for their aesthetic appeal than for their scientific content, and that colour images work better than black and white or grayscale images. Please do not try to design a cover with the Nature Ecology & Evolution logo etc., and please do not submit composites of images related to your work. I am sure you will understand that we cannot make any promise as to whether any of your suggestions might be selected for the cover of the journal.

You can generate the link yourself when you receive your article DOI by entering it here: <http://authors.springernature.com/share>.

Thank you for choosing NEE to publish your work; we look forward to seeing it published soon.

[REDACTED]

P.S. Click on the following link if you would like to recommend Nature Ecology & Evolution to your

62librarian <http://www.nature.com/subscriptions/recommend.html#forms>

** Visit the Springer Nature Editorial and Publishing website at www.springernature.com/editorial-and-publishing-jobs for more information about our career opportunities. If you have any questions please click here.**